# Multiplayer Federated Learning: Reaching Equilibrium with Less Communication

## Abstract

Traditional Federated Learning (FL) approaches assume collaborative clients with aligned objectives working towards a shared global model. However, in many real-world scenarios, clients act as rational players with individual objectives and strategic behaviors, a concept that existing FL frameworks are not equipped to adequately address. To bridge this gap, we introduce *Multiplayer Federated Learning (MpFL)*, a novel framework that models the clients in the FL environment as players in a game-theoretic context, aiming to reach an equilibrium. In this scenario, each player tries to optimize their own utility function, which may not align with the collective goal. Within MpFL, we propose *Per-Player Local Stochastic Gradient Descent* (PEARL-SGD), an algorithm in which each player/client performs local updates independently and periodically communicates with other players. We theoretically analyze PEARL-SGD and prove that it reaches a neighborhood of equilibrium with less communication in the stochastic setup than its non-local counterpart. Finally, we experimentally verify our theoretical findings.

## 1 Introduction

Federated Learning (FL) has emerged as a powerful collaborative learning paradigm where multiple clients jointly train a machine learning model without sharing their local data. In the classical FL setting, a central server coordinates multiple clients (e.g., mobile devices, edge devices) to collaboratively learn a shared global model without exchanging their own training data (Kairouz et al., 2021; Konečný et al., 2016; McMahan et al., 2017b; Li et al., 2020a). In this scenario, each client performs local computations on its private data and periodically communicates model updates to the server, which aggregates them to update the global model. This collaborative approach has been successfully applied in various domains, including natural language processing (Liu et al., 2021; Hard et al., 2018), computer vision (Liu et al., 2020a; Li et al., 2021), and healthcare (Antunes et al., 2022; Xu et al., 2021).

Despite their success, traditional FL frameworks rely on the key assumption that all participants are fully cooperative and share aligned objectives, collectively working towards optimizing the performance of a shared global model (e.g., minimizing the average of individual loss functions). This assumption overlooks situations where participants have individual objectives, or competitive interests that may not align with the collective goal. Diverse examples of such scenarios have been considered in the game theory literature, including Cournot competition in economics (Ahmed & Agiza, 1998), optical networks (Pan & Pavel, 2007), electricity markets (Saad et al., 2012), energy consumption control in smart grid (Ye & Hu, 2017), or mobile robot control (Kalyva & Psillakis, 2024). In the current era dominated by large-scale machine learning, relevant game-theoretic learning applications involving a large network of players could emerge.

To address these limitations of classical FL approaches, we propose a novel framework called *Multiplayer Federated Learning (MpFL)*, which models the FL process as a game among rational players with individual utility functions. In MpFL, each participant is considered a player who aims to optimize their own objective while interacting strategically with other players in the network via a central server. This game-theoretic perspective acknowledges that participants may act in their self-interest, have conflicting goals, or be unwilling to fully cooperate. By incorporating these dynamics, MpFL provides a more realistic and flexible foundation for FL in competitive and heterogeneous environments.

In the literature, there are multiple strategies that aim to incorporate a personalization approach into classical FL, including multi-task learning (Smith et al., 2017; Mills et al., 2021), transfer learning (Khodak et al., 2019), and mixing of the local and global models (Hanzely & Richtárik, 2020; Hanzely et al., 2020), to name a few. However, to the best of our knowledge, none of them is able to formulate the behaviour of the clients/players in a non-cooperative environment. This gap is precisely what MpFL aims to address.

In this work, we make the following main contributions:

- **Introducing Multiplayer Federated Learning (MpFL).** We develop a novel MpFL framework, which models the FL process as a game among rational players with individual utility functions. In MpFL, each client within the FL environment is viewed as a player of the game, and their local models are viewed as their actions. Each player constantly adjusts their model (action) to optimize their own objective function, and the MpFL framework aims for each player to reach to a Nash equilibrium by collaboratively training their model under the orchestration of a central server (e.g., service provider), while keeping the training data decentralized. That is, MpFL extends the scope of FL to scenarios where clients are allowed to have more general, diversified, possibly competing objectives.

- **Design and analysis of Per-Player Local SGD.** To handle the Multiplayer Federated Learning framework, we introduce *Per-Player Local SGD* (PEARL-SGD), a new algorithm inspired by the stochastic gradient descent ascent method in minimax optimization, that is able to handle the competitive nature of the players/clients. In PEARL-SGD, each player performs local SGD steps independently on their own actions/strategies (keeping the strategies of the other players fixed), and the updated actions/models are periodically communicated with the other players of the network via a central server.

- **Convergence guarantees for PEARL-SGD on heterogeneous data.** We provide tight convergence guarantees for PEARL-SGD, in both deterministic and stochastic regimes with heterogeneous data (see Table 1 for a summary of our results).

  - **Deterministic setting:** For the full-batch (deterministic) variant of PEARL-SGD, we prove that under suitable assumptions, PEARL-SGD converges linearly to an equilibrium for any communication period $\tau > 1$, provided that the constant step-size $\gamma$ is sufficiently small (see Theorem 3.3). In this setting, no communication gain is achieved via our analysis.

  - **Stochastic setting:** In its more general version, PEARL-SGD assumes that each player uses an unbiased estimator of its gradient in the update rule. For this setting, we provide two Theorems based on two different step-size choices:
    * *Constant step-size:* We show that under the same assumptions as in the deterministic case, PEARL-SGD converges linearly to a neighborhood of equilibrium (see Theorem 3.4). In Corollary 3.5, we show that with appropriate step-size depending on the total number of local SGD iterations $T$, PEARL-SGD achieves $\tilde{\mathcal{O}}(1/T)$ convergence rate with improved communication complexity when $T$ is sufficiently large.
    * *Decreasing step-size rule:* We prove that PEARL-SGD converges to an exact equilibrium (without neighborhood of convergence) with sublinear convergence (see Theorem 3.6). In this scenario, the asymptotic rate and communication complexity are essentially the same as in Corollary 3.5, but this result does not require the step-sizes to depend on $T$.

- **Numerical Evaluation:** We provide extensive numerical experiments verifying our theoretical results and show the benefits in terms of communications of PEARL-SGD over its non-local counterpart in the MpFL settings.

## 2 MULTIPLAYER FEDERATED LEARNING AND CLOSELY RELATED SETTINGS

In this section, we introduce the framework of Multiplayer Federated Learning and explain its main differences compared to the classical FL (Kairouz et al., 2021) and Federated Minimax Optimization (Deng & Mahdavi, 2021; Sharma et al., 2022; Zhang et al., 2023).

### 2.1 PROBLEM SETUP: MPFL

Multiplayer Federated Learning (MpFL) is a machine learning setting that combines the benefits of a game-theoretic formulation with classical federated learning. In this setting, the problem is

Table 1: Summary of theoretical results for PEARL-SGD. Theorem 3.3 considers the full-batch (deterministic) scenario. Theorem 3.4 and Theorem 3.6 both considers the general stochastic case. These results differ in the step-size choice; the former uses a constant step-size, while the latter uses decreasing step-sizes. In the *Convergence* column, "Linear" and "Sublinear" indicates the convergence rate, "Exact" refers to convergence to an equilibrium, and "Neighborhood" refers to convergence to a neighborhood of an equilibrium.

| *Theorem* | *Setting* | *Step-size* | *Convergence* |
|---|---|---|---|
| Theorem 3.3 | Deterministic | Constant | Linear+Exact |
| Theorem 3.4 | Stochastic | Constant | Linear+Neighborhood |
| Theorem 3.6 | Stochastic | Decreasing | Sublinear+Exact |

an $n$-player game in which multiple players/clients (e.g., mobile devices or whole organizations) communicate with each other via a central server (e.g., service provider) to reach an equilibrium. That is, reach a set of strategies—one for each player—such that no player can unilaterally deviate from their strategy to achieve a better payoff, given the strategies chosen by all other players.

In classical $n$-player games, communication between players was assumed to be cheap, easy, and straightforward, mainly because all players were in close proximity and had direct access to one another. This assumption made communication an insignificant concern in typical game theory analysis. However, with the advent of new large-scale machine learning applications, this is no longer the case. Nowadays, communication between players can be expensive and challenging, especially in distributed systems where the clients/players are geographically dispersed or operate under communication constraints. Addressing communication costs and designing communication-efficient algorithms for $n$-player games have become increasingly important, and this is precisely the challenge that Multiplayer Federated Learning aims to address.

Multiplayer games, where multiple players each minimize their own cost function that is affected by the actions of the others, are a long-studied, fundamental topic in mathematics and economics. (Nash Jr, 1950; Nash, 1951; Shapley, 1953; Schelling, 1980; Kreps et al., 1982; Harsanyi & Selten, 1988; Luce & Raiffa, 1989; Kreps, 1990; Von Neumann & Morgenstern, 2007). More recently, there has been increasing interest of the ML community in game-theoretic problems with motivating applications, including adversarial learning (Goodfellow et al., 2014; Daskalakis et al., 2018), multi-agent reinforcement learning (MARL) (Lanctot et al., 2017; Li et al., 2022; Sokota et al., 2023), and language models (Gemp et al., 2024; Jacob et al., 2024).

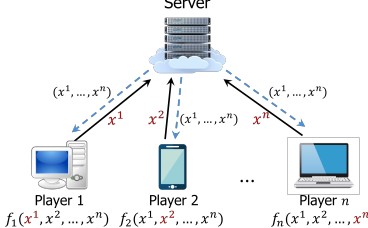

Figure 1: Illustration of MpFL for heterogeneous functions $f_i$. The goal is for each player to reach the equilibrium $(x_\star^1, \ldots, x_\star^n)$ (see (1)) with as little communication as possible.

**Equilibrium in $n$-player game.** Let $x^i \in \mathbb{R}^{d_i}$ denote the action of player $i \in [n]$ and let $\mathbf{x} = (x^1, \ldots, x^n) \in \mathbb{R}^D = \mathbb{R}^{d_1 + \cdots + d_n}$ be the joint action/strategy. Let $f_i(x^1, \ldots, x^n) \colon \mathbb{R}^{d_1 + \cdots + d_n} \to \mathbb{R}$ be the function of player $i$ and let $x^{-i} = (x^1, \ldots, x^{i-1}, x^{i+1}, \ldots, x^n) \in \mathbb{R}^{D - d_i}$ be the vector of all players' actions except that of player $i$. With this notation in place, the goal of the $n$-player game is to find an *equilibrium* $\mathbf{x}_\star = (x_\star^1, \ldots, x_\star^n) \in \mathbb{R}^D$ satisfying $f_i(x_\star^i; x_\star^{-i}) \leq f_i(x^i; x_\star^{-i})$ for each $x^i \in \mathbb{R}^{d_i}$, $i = 1, \ldots, n$, where $f_i(x^i; x^{-i}) = f_i(x^1, \ldots, x^n)$.

**MpFL: Multiplayer game in FL environment.** In the setting of interest of this paper, we focus on an $n$-player game in which multiple players/clients (e.g., mobile devices or whole organizations) communicate via a central server (e.g., service provider) to reach an equilibrium. In this setting, the classical clients of the federated learning environment are players of the $n$-player game, and each of them represents a client to the system (see Figure 1). Mathematically, the problem is formulated as

$$\underset{\mathbf{x}_\star = (x_\star^1, \ldots, x_\star^n)}{\text{find}} \quad f_i(x_\star^i; x_\star^{-i}) \leq f_i(x^i; x_\star^{-i}), \quad \forall x^i \in \mathbb{R}^{d_i}, \quad \text{for } i \in [n],$$

$$\text{where } f_i(x^1, \ldots, x^n) = \mathbb{E}_{\xi^i \sim \mathcal{D}_i} \left[ f_{i,\xi^i}(x^1, \ldots, x^n) \right]. \tag{1}$$

Here $\mathcal{D}_i$ denotes the data distribution of the $i$-th player, $f_{i,\xi^i}$ is the loss of the $i$-th player for a data point $\xi^i$ sampled from $\mathcal{D}_i$. In the FL environment, each client/player uses the strategies of all players to execute local updates and by keeping the other strategies fixed, and update their own value which later share with the master server that concatenates all new strategies and send them back to all players. Similar to the classical FL regime, our setting focuses on *heterogeneous data* (non-i.i.d) as we do not make any restrictive assumption on the data distribution $\mathcal{D}_i$ or the similarity between the functions of the players.

**Assumptions of multiplayer game.** Let us now present the main assumptions on the functions of the multiplayer game, which we later use to provide the convergence analysis for the proposed *Per-Player Local SGD*. Here, we denote the gradient of $f_i$ (function of player $i \in [n]$) with respect to $x^i$ by: $\nabla_{x^i} f_i(x^1, \ldots, x^n) = \nabla f_i(x^i; x^{-i})$. This convention allows us to remove the cumbersome subscript $x^i$ from the $\nabla$ notation; we only differentiate $f_i$ with respect to $x^i$ but never with $x^{-i}$.

In our work, we make two main assumptions on the functions $f_i$ of each player $i \in [n]$. We assume that the function is convex and smooth.

**Assumption 2.1 (*Convex (CVX)*).** For $i \in [n]$, for any $x^{-i} \in \mathbb{R}^{D-d_i}$, the local function $f_i(\cdot; x^{-i}) \colon \mathbb{R}^{d_i} \to \mathbb{R}$ is convex. That is, for any $x^i, y^i \in \mathbb{R}^{d_i}$ and $x^{-i} \in \mathbb{R}^{D-d_i}$,

$$f_i(y^i; x^{-i}) \geq f_i(x^i; x^{-i}) + \left\langle \nabla f_i(x^i; x^{-i}), y^i - x^i \right\rangle$$

**Assumption 2.2 (*Smoothness (SM)*).** For $i \in [n]$, for any $x^{-i} \in \mathbb{R}^{D-d_i}$, the local function $f_i(\cdot; x^{-i}) \colon \mathbb{R}^{d_i} \to \mathbb{R}$ is $L_i$-smooth. That is, for any $x^i, y^i \in \mathbb{R}^{d_i}$ and $x^{-i} \in \mathbb{R}^{D-d_i}$,

$$\left\| \nabla f_i(x^i; x^{-i}) - \nabla f_i(y^i; x^{-i}) \right\| \leq L_i \left\| x^i - y^i \right\|.$$

As we explained in the stochastic regime of MpFL we have $f_i(x^1, \ldots, x^n) = \mathbb{E}_{\xi^i \sim \mathcal{D}_i} \left[ f_{i,\xi^i}(x^1, \ldots, x^n) \right]$. In that scenario, to have convergence guarantees for PEARL-SGD, we need the following assumption of bounded variance. This is a common assumption in stochastic optimization literature which guarantees that the variance of the stochastic gradient oracle is bounded.

**Assumption 2.3 (*Bounded Variance (BV)*).** For each $i = 1, \ldots, n$, we assume

$$\mathbb{E}_{\xi^i \sim \mathcal{D}_i} \left[ \left\| \nabla f_{i,\xi^i}(x^i; x^{-i}) - \nabla f_i(x^i; x^{-i}) \right\|^2 \right] \leq \sigma_i^2, \quad \forall x^i \in \mathbb{R}^{d_i}, x^{-i} \in \mathbb{R}^{D-d_i}.$$

## 2.2 COMPARISON WITH CLOSELY RELATED FL FRAMEWORKS

Having presented the MpFL setting, let us now provide a short survey of related setups from classical FL and federated minimax optimization and compare them with our proposed MpFL. Additional list of related work is provided in Appendix A.

**Federated learning.** In its basic formulation, classical federated learning can be expressed as the minimization of the objective function (Kairouz et al., 2021),

$$\underset{x \in \mathbb{R}^d}{\text{minimize}} \quad f(x) = \frac{1}{n} \sum_{i=1}^n f_i(x) \quad \text{where} \quad f_i(x) = \mathbb{E}_{\xi^i \sim \mathcal{D}_i}[F_i(x, \xi^i)].$$

Here, $x \in \mathbb{R}^d$ represents the parameter for the global model, $f_i$ denotes the local objective function at client $i$, and $\mathcal{D}_i$ denotes the data distribution of client $i$. The local loss functions $F_i(x, \xi^i)$ are often the same across all clients, but the local data distribution $\mathcal{D}_i$ will often vary, capturing data heterogeneity. The foundational communication-efficient algorithm for this setup is FedAvg (Local SGD), proposed and massively popularized by McMahan et al. (2017a). Despite its simplicity, Local SGD has shown empirical success in terms of convergence speed and communication frequency, and many works have provided theoretical explanation for this performance (Stich, 2019; Dieuleveut & Patel, 2019; Stich & Karimireddy, 2020; Khaled et al., 2020).

In these works, clients work in a fully cooperative manner to find $x_\star = \operatorname{argmin}_{x \in \mathbb{R}^d} f(x)$, unlike our proposed MpFL where the clients who now serve as players of the game seek an equilibrium among possibly competing (non-cooperative) objectives.

**Federated minimax optimization.**    Federated minimax optimization was more recently proposed as a federated extension of minimax optimization problems, where the problem is formulated as:

$$\underset{x \in \mathbb{R}^{d_x}}{\text{minimize}} \ \underset{y \in \mathbb{R}^{d_y}}{\text{maximize}} \quad \mathcal{L}(x) = \frac{1}{n} \sum_{i=1}^{n} \mathcal{L}_i(x, y) \quad \text{where} \quad \mathcal{L}_i(x, y) = \mathbb{E}_{\xi^i \sim \mathcal{D}_i}[\phi_i(x, y, \xi)].$$

Here $n$ is the number of clients, and $\mathcal{L}_i(x, y)$ is the local loss function at client $i$ that depends on both $x$ and $y$, defined as $\mathcal{L}_i(x, y) = \mathbb{E}_{\xi \sim \mathcal{D}_i}[\phi_i(x, y, \xi)]$. Note that here each client has access to the information of both players $x$ and $y$. $\phi_i(x, y, \xi)$ denotes the loss for $\xi$, sampled from the local data distribution $\mathcal{D}_i$ at client $i$. Based on the properties of the model, the functions $\mathcal{L}_i(x, y)$ can be smooth/non-smooth, convex/non-convex with respect to player $x$, and concave/non-concave with respect to player $y$. The extension of Local SGD to these problems are Local Stochastic Gradient Descent-Ascent (SGDA) (Deng & Mahdavi, 2021; Sharma et al., 2022) or Local Stochastic Extragradient (SEG) (Beznosikov et al., 2020; 2022). More recently there were also approaches using primal-dual updates (Condat & Richtárik, 2022) and client-drift mitigation (Zhang et al., 2023).

While this line of work also studied federated learning in the context of (minimax) games, it is totally different from MpFL. The setup assumes that each FL client has access to both players of the minimax game, and they do not take the *multiplayer* aspect into account. In contrast, in MpFL, each client is a player of a large-scale multiplayer game who only has access to their objective $f_i$ and its gradient, and only updates their action $x^i$. We design the novel PEARL-SGD algorithm, suitable for the MpFL setting, a task not possible using the existing Local SGDA and Local SEG methods.

## 3  PEARL-SGD: Algorithm and Convergence Guarantees

In this section, we introduce and analyze Algorithm 1, named *Per-Player Local SGD* (PEARL-SGD), which is suitable for the MpFL setting we described in Section 2.

PEARL-SGD works by having the clients/players of the game run SGD independently in parallel for updating their strategy (keeping the strategies $x^{-i}$ of the other players fixed) and communicates the strategies of players only once in a while (via a central server). In more detail, in every round of PEARL-SGD, each player $i \in [n]$ runs $\tau$ iterations of SGD with respect to $f_i(\cdot, x^{-i})$, with $x^{-i}$ fixed to the information of the other players' actions obtained from the previous synchronization. Once each player completes $\tau$ SGD iterations (local updates), the server collects actions of all players, and distributes the concatenation of all updated strategies to all players (synchronization step).

Note that the synchronization step involves transferring $D = (d_1 + \cdots + d_n)$-dimensional vector (different from communication from classical FL where the dimension does not scale with $n$). This is a significant computational overhead, and it is required at every iteration for the non-local version of PEARL-SGD. We aim to reduce this overhead by communicating less frequently (with $\tau > 1$).

We emphasize that PEARL-SGD and its convergence hold without any assumption on players' data distributions $\mathcal{D}_i$, i.e., $f_i$'s can be very different among players and the setting is fully heterogeneous.

---

**Algorithm 1** Per-Player Local SGD (PEARL-SGD)

---

**Input:** Step-sizes $\gamma_k > 0$, Synchronization interval $\tau \geq 1$, Number of rounds $R \geq 1$
**Output:** $\mathbf{x}_{\tau R} \in \mathbb{R}^D$
  **for** $p = 0, \ldots, R - 1$ **do**
    Master server collects and distributes $\mathbf{x}_{\tau p} = (x_{\tau p}^1, \ldots, x_{\tau p}^n)$ to players $i = 1, \ldots, n$
    **for** $k = \tau p, \ldots, \tau(p + 1) - 1$ **do**
      **for** $i = 1, \ldots, n$ **do**
        Draw $\xi_k^i \sim \mathcal{D}_i$
        $g_k^i \leftarrow \nabla f_{i, \xi_k^i}(x_k^i; x_{\tau p}^{-i})$
        $x_{k+1}^i \leftarrow x_k^i - \gamma_k g_k^i$
      **end for**
    **end for**
  **end for**

---

**Assumptions on the joint gradient operator.**    We require some definitions and additional assumptions in order to carry out the theory. Define the joint gradient operator $\mathbb{F} \colon \mathbb{R}^D \to \mathbb{R}^D$ as

$$\mathbb{F}(\mathbf{x}) = \left( \nabla f_1(x^1; x^{-1}), \ldots, \nabla f_n(x^n; x^{-n}) \right).$$

**Assumption 3.1 (*Quasi-strong monotonicity (QSM)*).** There exists a unique equilibrium $\mathbf{x}_\star = (x_\star^1, \ldots, x_\star^n) \in \mathbb{R}^D$, for which $\mathbb{F}(\mathbf{x}_\star) = 0$, and $\mu > 0$ such that for any $\mathbf{x} \in \mathbb{R}^D$, $\langle \mathbb{F}(\mathbf{x}), \mathbf{x} - \mathbf{x}_\star \rangle \geq \mu \|\mathbf{x} - \mathbf{x}_\star\|^2$.

*(QSM)* is a concept extending the quasi-strong convexity (Gower et al., 2019) to the context of variational inequality problems (VIPs). This condition has been called with several different names in the literature, such as strong coherent VIPs (Song et al., 2020), VIPs with strong stability condition (Mertikopoulos & Zhou, 2019), or the strong Minty variational inequality (Diakonikolas et al., 2021). It is more general than strong monotonicity, and captures several non-monotone problems. Loizou et al. (2021) proposed this assumption for the analysis of SGDA, ensuring the convergence of SGDA dynamics in minimax games without the well-known issues of cycling or diverging (Mescheder et al., 2017; Daskalakis et al., 2018). Later, this also appeared in the study of Stochastic Extragradient (Gorbunov et al., 2022) and Stochastic Past Extragradient methods as well (Choudhury et al., 2024).

**Assumption 3.2 (*Star-cocoercivity (SCO)*).** $\mathbb{F}$ is $\frac{1}{\ell}$-star-cocoercive, i.e., there is $\ell > 0$ such that for any $\mathbf{x} \in \mathbb{R}^D$, $\langle \mathbb{F}(\mathbf{x}), \mathbf{x} - \mathbf{x}_\star \rangle \geq \frac{1}{\ell} \|\mathbb{F}(\mathbf{x})\|^2$.

Star-cocoercivity generalizes the class of coercive operators and, interestingly, can hold for non-Lipschitz operators (Loizou et al., 2021). This has also been taken as minimal assumption for SGDA analysis in prior work (Beznosikov et al., 2023).

Note that *(QSM)* and *(SCO)* together imply $\mu \|\mathbf{x} - \mathbf{x}_\star\| \leq \|\mathbb{F}(\mathbf{x})\| \leq \ell \|\mathbf{x} - \mathbf{x}_\star\|$ for any $\mathbf{x} \in \mathbb{R}^D$, which implies $\mu \leq \ell$. We call $\kappa = \ell/\mu \geq 1$ the *condition number* of the problem.

### 3.1 CONVERGENCE OF PEARL-SGD: DETERMINISTIC SETUP

First, we provide the convergence result for PEARL-SGD with constant step-size $\gamma_k \equiv \gamma$ in the full-batch (deterministic) scenario, where there is no noise in the gradient computation. While this is recovered as a special case of Theorem 3.4, we state it separately because the deterministic case provides several points of discussion which worth emphasis on their own.

**Theorem 3.3.** Assume *(CVX)*, *(SM)*, *(QSM)* and *(SCO)*, and let $L_{\max} = \max\{L_1, \ldots, L_n\}$. Let $0 < \gamma_k \equiv \gamma \leq \frac{1}{\ell\tau + 2(\tau-1)L_{\max}\sqrt{\kappa}}$, and $\kappa = \ell/\mu$ is the condition number. Then the Deterministic PEARL-SGD (Algorithm 1 with full-batch) converges with the rate

$$\|\mathbf{x}_{\tau R} - \mathbf{x}_\star\|^2 \leq (1 - \gamma\tau\mu\zeta)^R \|\mathbf{x}_0 - \mathbf{x}_\star\|^2$$

where $\zeta = 2 - \gamma\ell\tau - 2(\tau-1)\gamma L_{\max}\sqrt{\kappa/3} > 0$ (by the choice of $\gamma$).

Theorem 3.3 shows that deterministic PEARL-SGD converges linearly to an equilibrium. This distinguishes our result from the analyses of local gradient descent for finite sum minimization in heterogeneous data setups, where one has convergence to a neighborhood of optimum even when there is no noise (Khaled et al., 2020), unless further correction mechanism is used (Mishchenko et al., 2022). In addition, let us note that when $\tau = 1$, the step-size constraint and the convergence rate of Theorem 3.3 coincide with those from the analysis of the gradient descent-ascent (GDA) under *(QSM)* and *(SCO)* assumptions of (Loizou et al., 2021) showing the tightness of our analysis.

**Player drift and step-size constraint.** If $\gamma$ does not appropriately scale down with $\tau$, then at each round, players' actions (SGD iterates) converge to minimizers of local functions. Generally, this causes PEARL-SGD to quickly diverge away from the equilibrium. We call this phenomenon *player drift*, analogous to client drift of classical FL, enforcing the $\mathcal{O}(1/\tau)$ step-size.

### 3.2 CONVERGENCE OF PEARL-SGD: STOCHASTIC SETUP

We now discuss the convergence of PEARL-SGD with stochastic gradients. We first present the convergence of PEARL-SGD to a neighborhood of an equilibrium $\mathbf{x}_\star$ given constant step-sizes $\gamma_k \equiv \gamma$, and then discuss the communication complexity gain we achieve. Then we present the convergence result using a decreasing step-size selection, showing sublinear convergence to the

exact equilibrium $\mathbf{x}_\star$ rather than its neighborhood. While we defer the details of the proofs to Appendix B, we provide a proof outline for Theorem 3.4 in Section 3.3.

**Theorem 3.4.** Assume *(CVX)*, *(SM)*, *(BV)*, *(QSM)* and *(SCO)* hold. Let $0 < \gamma_k \equiv \gamma \le \frac{1}{\ell\tau + 2(\tau-1)L_{\max}\sqrt{\kappa}}$ and denote $q = L_{\max}/\sqrt{\ell\mu}$. Then PEARL-SGD exhibits the rate:

$$\mathbb{E}\left[\|\mathbf{x}_{\tau R} - \mathbf{x}_\star\|^2\right] \le (1 - \gamma\tau\mu\zeta)^R \|\mathbf{x}_0 - \mathbf{x}_\star\|^2 + \left(1 + (\tau-1)\left((4 + \sqrt{3}q)\gamma\tau L_{\max} + \frac{q}{2\tau}\right)\right)\frac{\gamma\sigma^2}{\mu\zeta}.$$

where $\sigma^2 = \sum_{i=1}^n \sigma_i^2$ and $\zeta = 2 - \gamma\ell\tau - 2(\tau-1)\gamma L_{\max}\sqrt{\kappa/3} > 0$ by the choice of $\gamma$.

When $\tau = 1$, with $\gamma \le 1/\ell$, the above rate becomes $\mathbb{E}\left[\|\mathbf{x}_R - \mathbf{x}_\star\|^2\right] \le (1 - \gamma\mu)^R \|\mathbf{x}_0 - \mathbf{x}_\star\|^2 + \gamma\sigma^2/\mu$, which is consistent with the usual analysis of the SGDA. Note that $\sigma^2$ is the sum of playerwise variances $\sigma_i^2 \ge \mathbb{E}_{\xi^i \sim \mathcal{D}_i}\left[\|\nabla f_{i,\xi^i}(x^i; x^{-i}) - \nabla f_i(x^i; x^{-i})\|^2\right]$, which represents the upper bound on the variance in estimating the joint gradient operator $\mathbb{F}(\cdot)$.

Based on Theorem 3.4, we show the convergence rate in terms of the total number of SGD iterations per player, $T = \tau R$, and discuss the communication complexity of PEARL-SGD.

**Corollary 3.5.** Under the assumptions of Theorem 3.4, let $q = L_{\max}/\sqrt{\ell\mu}$, $\gamma = \frac{1}{\mu\eta(1+2q)}$ and $T = \tau R = 2(1 + 2q)\eta\log\eta$, where $\eta > \kappa\tau$ is chosen so that $T$ is a multiple of $\tau$. Then

$$\mathbb{E}\left[\|\mathbf{x}_T - \mathbf{x}_\star\|^2\right] = \tilde{\mathcal{O}}\left(\frac{(1+q)^2 \|\mathbf{x}_0 - \mathbf{x}_\star\|^2}{T^2} + \frac{(1+q)\sigma^2}{\mu^2 T} + \frac{(1+q)\tau^2 L_{\max}\sigma^2}{\mu^3 T^2}\right)$$

where $\tilde{\mathcal{O}}$-notation drops polylogarithmic factors in $T$.

**Optimal $\tau$ and communication complexity.** In Corollary 3.5, the $\tilde{\mathcal{O}}\left((1+q)^2\|\mathbf{x}_0 - \mathbf{x}_\star\|^2/T^2\right)$ term decays fast (as $T$ grows) and the terms proportional to $\sigma^2$ become dominant. The order of convergence is no slower than the near-optimal $\tilde{\mathcal{O}}(1/T)$ rate, provided that $\tau^2 L_{\max}\sigma^2/\mu^3 T^2 = \mathcal{O}(\sigma^2/\mu^2 T) \iff \tau = \mathcal{O}\left(\sqrt{\mu T/L_{\max}}\right)$. Up to this factor we can gain theoretical improvement in the communication cost compared to fully communicating case $\tau = 1$ (provided that $T$ is sufficiently large), and the resulting communication complexity is $T/\tau = \Theta\left(\sqrt{TL_{\max}/\mu}\right) = \Theta\left(\sqrt{T}\right)$.

**Convergence to equilibrium via decreasing step-sizes.** We conclude the section with convergence result for PEARL-SGD using a decreasing step-size selection. While showing a similar convergence rate in terms of $T$ as in Corollary 3.5, Theorem 3.6 has the advantage of not requiring to fix $T$ in advance to determine the step-sizes.

**Theorem 3.6.** Under the assumptions of Theorem 3.4, let $q = L_{\max}/\sqrt{\ell\mu}$, and choose the step-sizes $\gamma_k = \begin{cases} \frac{1}{\ell\tau(1+2q)} & \text{if } p < 2(1+2q)\kappa \\ \frac{1}{\tau\mu}\frac{2p+1}{(p+1)^2} & \text{if } p \ge 2(1+2q)\kappa \end{cases}$ for $\tau p \le k \le \tau(p+1) - 1, p = 0, \ldots, R-1$. Then PEARL-SGD converges with the rate

$$\mathbb{E}\left[\|\mathbf{x}_T - \mathbf{x}_\star\|^2\right] \le \frac{4(1+2q)^2\kappa^2\tau^2 \|\mathbf{x}_0 - \mathbf{x}_\star\|^2}{eT^2} + \frac{4(1+q)\sigma^2}{\mu^2 T}$$
$$+ \frac{4(1+2q)^2\kappa\tau\sigma^2}{\mu^2 T^2}\left(1 + \frac{2\tau}{\sqrt{\kappa}}\right) + \frac{32(1+q)\tau^2 L_{\max}\sigma^2 \log T}{\mu^3 T^2}$$

where $T = \tau R$ is the total number of iterations.

## 3.3 PROOF OUTLINE

In this section, we provide a proof outline for Theorem 3.4. The key components of the proof are as follows: **(i)** a round of local SGD in PEARL-SGD behaves like a large single descent step with

respect to the joint gradient operator $\mathbb{F}$ except for *local error* terms caused by running multiple SGD steps locally (Lemma 3.7), and **(ii)** we bound these local error terms (Lemma 3.8).

**Lemma 3.7.** Assume *(SM)*, and let $L_{\max} = \max\{L_1, \ldots, L_n\}$. Let $0 \leq p \leq R - 1$ be a fixed round index in PEARL-SGD and suppose $\gamma_k \equiv \gamma > 0$ for $k = \tau p, \ldots, \tau(p+1) - 1$. Then for arbitrary $\alpha > 0$, we have

$$
\mathbb{E}\left[\left\|\mathbf{x}_{\tau(p+1)} - \mathbf{x}_\star\right\|^2 \,\Big|\, \mathbf{x}_{\tau p}\right] \leq (1 + (\tau - 1)\alpha\gamma) \left\|\mathbf{x}_{\tau p} - \mathbf{x}_\star\right\|^2 - 2\gamma\tau \left\langle \mathbb{F}(\mathbf{x}_{\tau p}), \mathbf{x}_{\tau p} - \mathbf{x}_\star \right\rangle
$$
$$
+ \frac{\gamma L_{\max}^2}{\alpha} \sum_{j=\tau p+1}^{\tau p + \tau - 1} \mathbb{E}\left[\left\|\mathbf{x}_{\tau p} - \mathbf{x}_j\right\|^2 \,\Big|\, \mathbf{x}_{\tau p}\right] + \mathbb{E}\left[\left\|\mathbf{x}_{\tau p} - \mathbf{x}_k\right\|^2 \,\Big|\, \mathbf{x}_{\tau p}\right].
$$

**Local error bound.** Lemma 3.7 shows that we need to bound the quantity

$$
\mathbb{E}\left[\left\|\mathbf{x}_{\tau p} - \mathbf{x}_k\right\|^2 \,\Big|\, \mathbf{x}_{\tau p}\right] = \sum_{i=1}^{n} \mathbb{E}\left[\left\|x_{\tau p}^i - x_k^i\right\|^2 \,\Big|\, \mathbf{x}_{\tau p}\right] \tag{2}
$$

for $k = \tau p + 1, \ldots, \tau(p+1)$. This is achieved by the following result.

**Lemma 3.8.** Suppose Assumptions *(CVX)*, *(SM)* and *(BV)* hold. For a fixed $i \in [n]$ and a fixed communication round $p$ in PEARL-SGD, suppose $\gamma_k \equiv \gamma$ for $k = \tau p, \ldots, \tau(p+1) - 1$, where $0 < \gamma \leq \frac{1}{L_i} \min\left\{1, \frac{1}{\tau-1}\right\}$. Then for $t = 0, \ldots, \tau$,

$$
\mathbb{E}\left[\left\|x_{\tau p}^i - x_{\tau p+t}^i\right\|^2 \,\Big|\, \mathbf{x}_{\tau p}\right] \leq \gamma^2 t^2 \left\|\nabla f(x_{\tau p}^i; x_{\tau p}^{-i})\right\|^2 + \gamma^2 t \left(1 + 2(t-1)(t+1)\gamma L_i\right)\sigma_i^2.
$$

Here we sketch the proof of Lemma 3.8 and clarify the role of Assumption *(CVX)*. By assuming each $f_i(\cdot; x_{\tau p}^{-i})$ is convex and $L_i$-smooth, we can prove Lemma 3.9, showing that the expectation of squared gradient norm is "almost" nonincreasing along the local SGD steps, except for some additional term due to stochasticity. Then, we rewrite each summand in (2) as

$$
\mathbb{E}\left[\left\|x_{\tau p}^i - x_k^i\right\|^2 \,\Big|\, \mathbf{x}_{\tau p}\right] = \mathbb{E}\left[\gamma^2 \left\|\sum_{j=\tau p}^{k-1} g_j^i\right\|^2 \,\Big|\, \mathbf{x}_{\tau p}\right] = \mathbb{E}\left[\gamma^2 \left\|\sum_{j=\tau p}^{k-1} \nabla f_{i,\xi_j^i}(x_j^i; x_{\tau p}^{-i})\right\|^2 \,\Big|\, \mathbf{x}_{\tau p}\right] \tag{3}
$$

and use Lemma 3.9 to bound (3).

**Lemma 3.9.** Under the assumptions of Lemma 3.8, for $j = \tau p + 1, \ldots, \tau(p+1)$,

$$
\mathbb{E}\left[\left\|\nabla f_i(x_j^i; x_{\tau p}^{-i})\right\|^2 \,\Big|\, \mathbf{x}_{\tau p}\right] \leq \left\|\nabla f_i(x_{\tau p}^i; x_{\tau p}^{-i})\right\|^2 + 2(j - \tau p)\gamma L_i \sigma_i^2.
$$

**Remark.** Given (3), it is tempting to apply Jensen's inequality to the rightmost quantity and then apply Lemma 3.9. However, this results in a bound that is looser than our Lemma 3.8. We need more careful arguments regarding the expectations, which we detail throughout Appendix B.

*Proof outline for Theorem 3.4.* We combine Lemmas 3.7 and 3.8, and then apply *(SCO)* to eliminate the $\|\mathbb{F}(\mathbf{x}_{\tau p})\|^2$ terms to obtain

$$
\mathbb{E}\left[\left\|\mathbf{x}_{\tau(p+1)} - \mathbf{x}_\star\right\|^2 \,\Big|\, \mathbf{x}_{\tau p}\right] \leq (1 + (\tau - 1)\alpha\gamma) \left\|\mathbf{x}_{\tau p} - \mathbf{x}_\star\right\|^2 + \text{(terms proportional to } \sigma^2)
$$
$$
- \underbrace{\left(2\gamma\tau - \gamma^2\tau^2\ell - \frac{\gamma^3 L_{\max}^2 \tau^2(\tau-1)\ell}{3\alpha}\right)}_{:=C} \left\langle \mathbb{F}(\mathbf{x}_{\tau p}), \mathbf{x}_{\tau p} - \mathbf{x}_\star \right\rangle \tag{4}
$$

Provided that $C \geq 0$, we can upper bound the second line of (4) by $-C\mu \left\|\mathbf{x}_{\tau p} - \mathbf{x}_\star\right\|^2$ using *(QSM)*. Then we choose $\alpha = \gamma\tau L_{\max}\sqrt{\frac{\ell\mu}{3}}$ which minimizes the resulting coefficient of $\left\|\mathbf{x}_{\tau p} - \mathbf{x}_\star\right\|^2$, and rewrite it in the form $1 - \gamma\tau\mu\zeta$. Finally, take expectation over $\mathbf{x}_{\tau p}$ in (4) and unroll the recursion. $\square$

## 4 NUMERICAL EXPERIMENTS

In this section, we conduct experiments to assess the empirical performance of PEARL-SGD and verify our theory. We show two setups: first, a minimax game (where $n = 2$) and second, a multi-player game with $n = 5$ players. Details of the experiments are provided in Appendix C.

### 4.1 QUADRATIC MINIMAX GAME

Consider the minimax game $\min_{u\in\mathbb{R}^d} \max_{v\in\mathbb{R}^d} \mathcal{L}(u,v) = \frac{1}{M}\sum_{m=1}^{M}\mathcal{L}_m(u,v)$ where $\mathcal{L}_m(u,v)$ is as below ($\mathbf{A}_m, \mathbf{B}_m, \mathbf{C}_m$ are matrices and $a_m, c_m$ are vectors). In this two-player zero-sum game, we have $n = 2$ with $f_1(x^1; x^2) = \mathcal{L}(x^1, x^2)$ and $f_2(x^2; x^1) = -\mathcal{L}(x^1, x^2)$.

$$\mathcal{L}_m(u,v) := \tfrac{1}{2}\langle u, \mathbf{A}_m u\rangle + \langle u, \mathbf{B}_m v\rangle - \tfrac{1}{2}\langle v, \mathbf{C}_m v\rangle + \langle a_m, u\rangle - \langle c_m, v\rangle.$$

**PEARL-SGD with tuned step-size.** In this experiment, we demonstrate the empirical performance of PEARL-SGD with varying values of $\tau$. For each $\tau \in \{1, 2, 4, 5, 8, 20\}$, we tune $\gamma$ by running PEARL-SGD with each $\gamma \in \{10^{-1}, 10^{-2}, \ldots, 10^{-6}\}$, and plot the best relative error $\|\mathbf{x}_{\tau p}-\mathbf{x}_\star\|^2/\|\mathbf{x}_0-\mathbf{x}_\star\|^2$ ($y$-axis) versus the communication round index $p$ ($x$-axis).

Figure 2a presents results from Deterministic PEARL-SGD. We observe that performance improves as $\tau$ is increased from 1 to 5, and then degrades. Figure 2b presents results under stochasticity, imposed by mini-batching from the finite sum. We repeat each experiment 5 times and plot the mean relative error with standard deviation (shaded region). Here we observe the lowest relative errors with large values of $\tau$, demonstrating the advantage of larger synchronization intervals in PEARL-SGD given stochastic gradients.

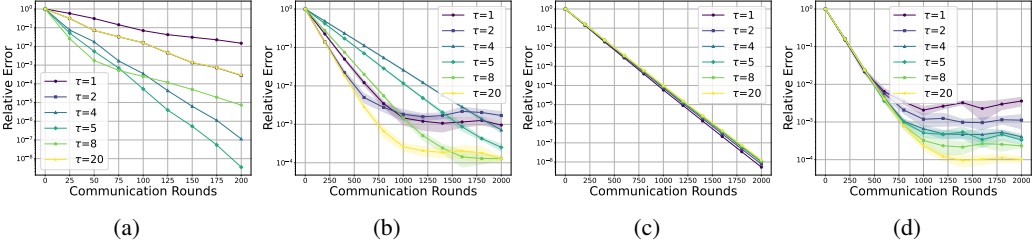

| (a) | (b) | (c) | (d) |

Figure 2: Performance of PEARL-SGD on quadratic minimax game, with different $\tau$. Figures 2a (deterministic) and 2b (stochastic) show the performance of PEARL-SGD with empirically tuned step-sizes, and Figures 2c (deterministic) and 2d (stochastic) show the performance with tight theoretical step-sizes.

**PEARL-SGD with theoretical step-size.** Figures 2c and 2d demonstrates the performance of PEARL-SGD using the theoretical step-size $\gamma = 1/(\ell\tau + 2(\tau-1)L_{\max}\sqrt{\kappa})$ from Theorems 3.3 and 3.4 with $\tau \in \{1, 2, 4, 5, 8, 20\}$. Figure 2c shows results from Deterministic PEARL-SGD, and Figure 2d shows the stochastic case. In the deterministic case, as $\gamma$ scales down with $\tau$, we observe similar linear convergence pattern for all values of $\tau$. On the other hand, in the stochastic case, we observe clear benefit of using larger $\tau$; it reaches smaller relative error within same number of communication rounds. This is consistent with Corollary 3.5, predicting reduced communication cost in the stochastic case.

**Performance of PEARL-SGD for different $(\gamma, \tau)$ pairs.** Figure 3 displays the heatmap of relative errors (log-scale) after 100 communication rounds of Deterministic PEARL-SGD on a quadratic minimax game. White and yellow regions indicate divergence/poor performance; darker regions indicate lower relative errors.

Figure 3 reveals a trend: for a fixed $\gamma$, PEARL-SGD's performance improves as $\tau$ increases up to certain threshold, after which it declines and finally diverges. Another key observation is that the dark region of the heatmap

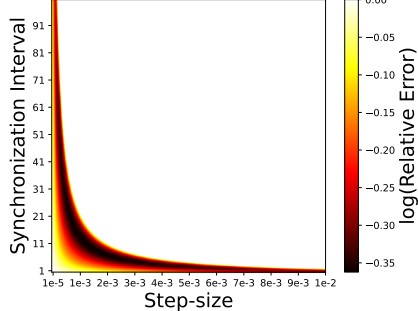

Figure 3: Heatmap of $\log$(Relative Errors).

(signifying the best performance) takes the shape of a hyperbola. This is consistent with our Theorem 3.3, showing the relationship $\gamma_\tau \propto 1/\tau$ where $\gamma_\tau$ is the optimal step-size choice given $\tau$ (providing fastest convergence).

## 4.2 $n$-PLAYER GAME

In this section, we analyze an $n$-player game where the local function for the $i$-th player is given by:

$$\min_{x^i \in \mathbb{R}^d} f_i(x^i; x^{-i}) := \frac{1}{M} \sum_{m=1}^{M} f_{i,m}(x^i; x^{-i}), \tag{5}$$

for $i = 1, \ldots, n$ (with $d_1 = \cdots = d_n = d$). Each $f_{i,m}$ takes the form:

$$f_{i,m}(x^i; x^{-i}) = \frac{1}{2} \langle x^i, \mathbf{A}_{i,m} x^i \rangle + \sum_{1 \le j \le n, j \ne i} \langle x^i, \mathbf{B}_{i,j,m} x^j \rangle + \langle a_{i,m}, x^i \rangle,$$

where $\mathbf{A}_{i,m}, \mathbf{B}_{i,j,m} \in \mathbb{R}^{d \times d}$ and $a_{i,m} \in \mathbb{R}^d$ for $m = 1, \ldots, M$. We set the number of players to $n = 5$ for the subsequent experiments.

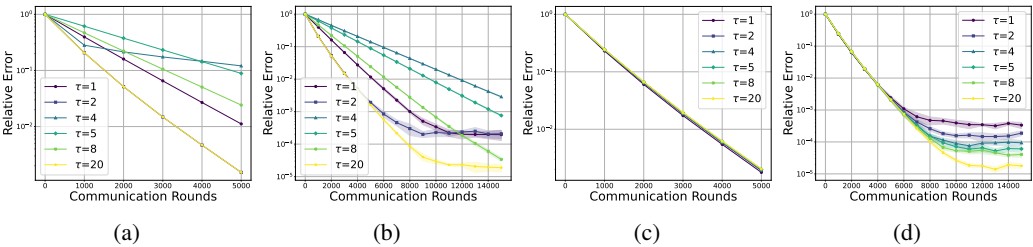

(a)  (b)  (c)  (d)

Figure 4: Performance of PEARL-SGD on the $n$-player game defined by (5), with different $\tau$. Figures 4a (deterministic) and 4b (stochastic) show the performance of PEARL-SGD with empirically tuned step-sizes, and Figures 4c (deterministic) and 4d (stochastic) show the performance with tight theoretical step-sizes.

PEARL-SGD **with tuned step-size.**    In this experiment we use tuned step-size for each choice of synchronization interval $\tau \in \{1, 2, 4, 5, 8, 20\}$. We use the same $\gamma$-grid $\{10^{-1}, 10^{-2}, \cdots, 10^{-6}\}$ as in Section 4.1 and proceed similarly. Figure 4a shows that the choices $\tau = 2$ and $\tau = 20$ outperform the fully communicating case $\tau = 1$ with step-size tuning. Figure 4b shows results from the stochastic setting, indicating that using larger values of $\tau$ could lead to higher accuracy levels, as in Section 4.1.

PEARL-SGD **with theoretical step-size.**    Again, we run PEARL-SGD with the theoretical step-size $\gamma = 1/(\ell\tau + 2(\tau-1)L_{\max}\sqrt{\kappa})$ of Theorems 3.3 and 3.4, for $\tau \in \{1, 2, 4, 5, 8, 20\}$. We set the cocoercivity parameter to $\ell = L^2/\mu$ according to Facchinei & Pang (2003), where $L$ and $\mu$ are explicitly computed Lipschitz constant and strong monotonicity parameters of $\mathbb{F}$. Figure 4c displays the results from Deterministic PEARL-SGD and similarly as in Section 4.1, we observe that all values of $\tau$ produce indistinguishable performance plots. On the other hand, Figure 4d demonstrates that in the general stochastic setting, PEARL-SGD with larger synchronization interval $\tau$ provides a clear benefit of achieving smaller relative error using the same number of communication rounds.

## 5 CONCLUSION

In this paper, we introduce Multiplayer Federated Learning (MpFL), a FL framework under setups where clients, strategically acting in their own interests, collaborate through a central server to train models (actions) with the goal of reaching an equilibrium. We propose the PEARL-SGD algorithm handling MpFL, and provide its tight convergence guarantees under heterogeneous setups where each player has distinct objectives and data distributions. We show that PEARL-SGD provides improved communication complexity, reducing the primary overhead in large-scale applications.

Our work offers a number of potential extensions by incorporating the ideas such as Extragradient (Korpelevich, 1976), asynchronous updates (Dean et al., 2012; Stich, 2019), gradient compression (Alistarh et al., 2017), gradient tracking (Nedic et al., 2017) and algorithmic correction for drifts (Karimireddy et al., 2020; Mishchenko et al., 2022). We anticipate that our initiation of the study of MpFL will lead to interesting future work including but not limited to these topics.

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

# Supplementary Material

We organize the Supplementary Material as follows: Section A provides an additional survey of related work. Section B presents the proofs of theoretical results omitted from the main text. Section C provides the details of the experiments omitted from the main paper. Section D provides an additional experiment on application involving the control of mobile robots. Section E provides detailed explanation and interpretation on the theoretical assumptions made in the paper.

## CONTENTS

## A  ADDITIONAL RELATED WORK AND DISCUSSION

**Heterogeneity and client drift.**   One fundamental challenge for theory of Local SGD (FedAvg) is heterogeneity, i.e., varying $f_i$'s due to differences in local data distributions (Konečný et al., 2016; Li et al., 2020b). Under such setup, Local SGD is prone to client drift (Zhao et al., 2018; Karimireddy et al., 2020) where local descent trajectories head toward distinct minima (of local objectives), and convergence theories require either additional assumptions (Wang et al., 2019; Yu et al., 2019; Haddadpour & Mahdavi, 2019; Li et al., 2020b) or technical analyses (Khaled et al., 2020; Koloskova et al., 2020) to control this drift. Some papers, based on theoretical insights, introduced/analyzed correction mechanisms for Local SGD to mitigate client drift (Karimireddy et al., 2020; Gorbunov et al., 2021; Mitra et al., 2021; Mishchenko et al., 2022; Hu & Huang, 2023; Grudzień et al., 2023). We note that the $n$-player game setup of MpFL is also fully heterogeneous as each player has distinct (possibly even conflicting) objective functions, and consequently, we have the analogous concept of *player drift*.

**Client drift vs. Player drift.**   Two two concepts of drifts are seemingly similar, but they are distinct concepts. Here we highlight the key differences between them. The client drift occurs in the classical FL (minimization) setup

$$\min_{x \in \mathbb{R}^d} \ \frac{1}{n} \sum_{i=1}^{n} f_i(x),$$

and indicates the phenomenon where each client $i$ converges to $x_\star^i = \mathrm{argmin}_{x \in \mathbb{R}^d} f_i(x)$ (if excessive number of local steps are performed using large step-sizes). Usually, this leads Local SGD to converge to the mean of $x_\star^i$'s, instead of $x_\star = \mathrm{argmin}_{x \in \mathbb{R}^d} \ f(x)$ (so the problem is the convergence to a biased—incorrect—solution). On the other hand, player drift occurs in the MpFL setup, and indicates each client $i$ converging to $x_\star^i(x_{\tau p}^{-i}) := \mathrm{argmin}_{x^i \in \mathbb{R}^{d_i}} f_i(x^i; x_{\tau p}^{-i})$ (in the extreme case). Note that there is a dependency on $x_{\tau p}^{-i}$, the strategy of other players. This could lead PEARL-SGD dynamics to even diverge away to the infinity, which can be checked with simple examples, e.g., a two-player quadratic minimax game $\min_{u \in \mathbb{R}} \max_{v \in \mathbb{R}} \ \frac{\mu}{2} u^2 + uv - \frac{\mu}{2} v^2$ with $\mu < 1$.

In short, there are three notable conceptual differences: **1)** the setup in which they occur, **2)** dependency of undesirable local solutions on other players' iterates, and **3)** dynamics of the algorithm (incorrect convergence vs. divergence).

**FL frameworks with individual models.**   There are several distinct contexts for FL frameworks where each client learns an individual model. In Personalized FL (Fallah et al., 2020; T. Dinh et al., 2020; Hanzely et al., 2020; Hanzely & Richtárik, 2020; Deng et al., 2020; Tan et al., 2023), clients aim to learn models tailored to each local distributions, while benefiting from collaborative learning. In Vertical FL (Yang et al., 2019; Liu et al., 2022; 2024) scenarios, multiple organizations hold distinct features from the common set of samples and they collaborate to train their each local model. In Federated Transfer Learning (Sharma et al., 2019; Liu et al., 2020b; Feng et al., 2022), the participating organizations similarly keep and train local models, but their datasets have heterogeneity over both sample and feature spaces with limited overlaps. Federated Multi-Task Learning (Smith et al., 2017; Marfoq et al., 2021; Mills et al., 2021) extends FL to cases where each client solves different, but related tasks.

**Distributed coordinate descent methods.**   In the "homogeneous" case of MpFL, where all players share the same objective $f$, our PEARL-SGD seems related to distributed coordinate descent methods (Richtárik & Takáč, 2016), where coordinates of the optimization variable are partitioned and distributed to multiple computers, working in parallel to minimize $f$. However, the main motivation of coordinate descent is to gain speedup via parallelization of gradient computation over nodes, and hence they focus on how the number of workers or the number of random coordinates chosen per iteration affects the convergence rate. On the other hand, PEARL-SGD aims to reduce the cost of communication among players, and we focus on how large $\tau$ (the communication period) can become without compromising the convergence rate.

## B OMITTED PROOFS FOR PER-PLAYER LOCAL SGD (PEARL-SGD)

### B.1 PROOF OF LEMMA 3.7

For $k = \tau p + 1, \ldots, \tau(p+1)$, we have

$$
\begin{aligned}
\|\mathbf{x}_k - \mathbf{x}_\star\|^2 &= \sum_{i=1}^{n} \left\| x_k^i - x_\star^i \right\|^2 \\
&= \sum_{i=1}^{n} \left\| x_k^i - x_\star^i - (x_{\tau p}^i - x_k^i) \right\|^2 \\
&= \sum_{i=1}^{n} \left[ \left\| x_{\tau p}^i - x_\star^i \right\|^2 - 2 \left\langle x_{\tau p}^i - x_\star^i, x_{\tau p}^i - x_k^i \right\rangle + \left\| x_{\tau p}^i - x_k^i \right\|^2 \right] \\
&= \|\mathbf{x}_{\tau p} - \mathbf{x}_\star\|^2 - 2\gamma \sum_{i=1}^{n} \sum_{j=\tau p}^{k-1} \left\langle x_{\tau p}^i - x_\star^i, g_j^i \right\rangle + \sum_{i=1}^{n} \left\| x_{\tau p}^i - x_k^i \right\|^2, \quad (6)
\end{aligned}
$$

where

$$
g_j^i = \nabla f_{i,\xi_j^i}(x_j^i; x_{\tau p}^{-i})
$$

for $j = \tau p, \ldots, k-1$ and $i = 1, \ldots, n$. Note that we have

$$
\mathbb{E}_{\xi_{\tau p}^i} \left[ - \left\langle x_{\tau p}^i - x_\star^i, g_{\tau p}^i \right\rangle \mid \mathbf{x}_{\tau p} \right] = - \left\langle x_{\tau p}^i - x_\star^i, \nabla f_i(x_{\tau p}^i; x_{\tau p}^{-i}) \right\rangle,
$$

while for the other indices $j = \tau p + 1, \ldots, k-1$, we have the upper bound

$$
\begin{aligned}
&\mathbb{E}_{\xi_j^i} \left[ - \left\langle x_{\tau p}^i - x_\star^i, g_j^i \right\rangle \mid x_j^i \right] \\
&= - \left\langle x_{\tau p}^i - x_\star^i, \nabla f_i(x_j^i; x_{\tau p}^{-i}) \right\rangle \\
&= - \left\langle x_{\tau p}^i - x_\star^i, \nabla f_i(x_{\tau p}^i; x_{\tau p}^{-i}) \right\rangle + \left\langle x_{\tau p}^i - x_\star^i, \nabla f_i(x_{\tau p}^i; x_{\tau p}^{-i}) - \nabla f_i(x_j^i; x_{\tau p}^{-i}) \right\rangle \\
&\leq - \left\langle x_{\tau p}^i - x_\star^i, \nabla f_i(x_{\tau p}^i; x_{\tau p}^{-i}) \right\rangle + \frac{\alpha}{2} \left\| x_{\tau p}^i - x_\star^i \right\|^2 + \frac{1}{2\alpha} \left\| \nabla f_i(x_{\tau p}^i; x_{\tau p}^{-i}) - \nabla f_i(x_j^i; x_{\tau p}^{-i}) \right\|^2 \\
&\leq - \left\langle x_{\tau p}^i - x_\star^i, \nabla f_i(x_{\tau p}^i; x_{\tau p}^{-i}) \right\rangle + \frac{\alpha}{2} \left\| x_{\tau p}^i - x_\star^i \right\|^2 + \frac{L_i^2}{2\alpha} \left\| x_{\tau p}^i - x_j^i \right\|^2
\end{aligned}
$$

where in the fourth line, we use Young's inequality with an arbitrary $\alpha > 0$ that we determine later. Take expectations of the both sides in (6) (conditioned on $\mathbf{x}_{\tau p}$), and apply the above bound with the tower rule to obtain

$$
\begin{aligned}
&\mathbb{E} \left[ \|\mathbf{x}_k - \mathbf{x}_\star\|^2 \,\Big|\, \mathbf{x}_{\tau p} \right] \\
&\leq \|\mathbf{x}_{\tau p} - \mathbf{x}_\star\|^2 - 2\gamma \sum_{i=1}^{n} \sum_{j=\tau p}^{k-1} \left\langle x_{\tau p}^i - x_\star^i, \nabla f_i(x_{\tau p}^i; x_{\tau p}^{-i}) \right\rangle + 2\gamma \sum_{i=1}^{n} \sum_{j=\tau p+1}^{k-1} \frac{\alpha}{2} \left\| x_{\tau p}^i - x_\star^i \right\|^2 \\
&\quad + 2\gamma \sum_{i=1}^{n} \sum_{j=\tau p+1}^{k-1} \mathbb{E} \left[ \frac{L_i^2}{2\alpha} \left\| x_{\tau p}^i - x_j^i \right\|^2 \,\Big|\, \mathbf{x}_{\tau p} \right] + \sum_{i=1}^{n} \mathbb{E} \left[ \left\| x_{\tau p}^i - x_k^i \right\|^2 \,\Big|\, \mathbf{x}_{\tau p} \right].
\end{aligned}
\quad (7)
$$

Now we apply the identities

$$
\sum_{i=1}^{n} \left\langle x_{\tau p}^i - x_\star^i, \nabla f_i(x_{\tau p}^i; x_{\tau p}^{-i}) \right\rangle = \left\langle \mathbf{x}_{\tau p} - \mathbf{x}_\star, \mathbb{F}(\mathbf{x}_{\tau p}) \right\rangle, \quad \sum_{i=1}^{n} \left\| x_{\tau p}^i - x_\star^i \right\|^2 = \|\mathbf{x}_{\tau p} - \mathbf{x}_\star\|^2
$$

$$
\sum_{i=1}^{n} \mathbb{E} \left[ \left\| x_{\tau p}^i - x_k^i \right\|^2 \,\Big|\, \mathbf{x}_{\tau p} \right] = \mathbb{E} \left[ \|\mathbf{x}_{\tau p} - \mathbf{x}_k\|^2 \,\Big|\, \mathbf{x}_{\tau p} \right]
$$

and the inequality

$$
\begin{aligned}
\sum_{i=1}^{n} \sum_{j=\tau p+1}^{k-1} \mathbb{E} \left[ \frac{L_i^2}{2\alpha} \left\| x_{\tau p}^i - x_j^i \right\|^2 \,\Big|\, \mathbf{x}_{\tau p} \right] &\leq \frac{L_{\max}^2}{2\alpha} \sum_{j=\tau p+1}^{k-1} \sum_{i=1}^{n} \mathbb{E} \left[ \left\| x_{\tau p}^i - x_j^i \right\|^2 \,\Big|\, \mathbf{x}_{\tau p} \right] \\
&= \frac{L_{\max}^2}{2\alpha} \sum_{j=\tau p+1}^{k-1} \mathbb{E} \left[ \|\mathbf{x}_{\tau p} - \mathbf{x}_j\|^2 \,\Big|\, \mathbf{x}_{\tau p} \right]
\end{aligned}
$$

to (7) and plug in $k = \tau(p + 1)$, which gives the desired result.

## B.2 SOME GENERAL ANALYSES OF SGD

In this section we present some general properties of stochastic gradient descent (SGD) for an $L$-smooth, convex function $f \colon \mathbb{R}^m \to \mathbb{R}$. Suppose that we have a stochastic oracle $\nabla f_\xi(\cdot)$ for the gradient operator $\nabla f(\cdot)$, satisfying

$$\mathbb{E}_\xi[\nabla f_\xi(x)] = \nabla f(x), \quad \mathbb{E}_\xi\left[\|\nabla f_\xi(x) - \nabla f(x)\|^2\right] \le \rho^2, \quad \forall x \in \mathbb{R}^m. \tag{8}$$

This setup and the subsequent results are the abstractions of intermediate results that we need for the proofs of Lemma 3.9 and Lemma 3.8. Specifically, we will later use the results of this section with

$$f(\cdot) = f_i(\cdot; x_{\tau p}^{-i}), \qquad \rho^2 = \sigma_i^2,$$

for each $i = 1, \ldots, n$. We make this abstraction to simplify notations and to more effectively convey the key intuitions underlying the analyses.

**Lemma B.1.** Let $f \colon \mathbb{R}^m \to \mathbb{R}$ be convex and $L$-smooth. Suppose that a stochastic gradient oracle $\nabla f_\xi(\cdot)$ satisfies (8). Let $y = x - \gamma \nabla f_\xi(x)$, where $0 < \gamma \le \frac{2}{L}$. Then we have

$$\mathbb{E}_\xi\left[\|\nabla f(y)\|^2\right] \le \|\nabla f(x)\|^2 + 2\gamma L \rho^2.$$

*Proof.* It is well-known that if $f$ is convex and $L$-smooth, then $\nabla f$ is $\frac{1}{L}$-cocoercive, i.e., for any $x, y \in \mathbb{R}^m$,

$$\langle x - y, \nabla f(x) - \nabla f(y) \rangle \ge \frac{1}{L} \|\nabla f(x) - \nabla f(y)\|^2.$$

By cocoercivity and the step-size condition $\gamma \le \frac{2}{L}$, we have

$$\frac{\gamma}{2} \|\nabla f(x) - \nabla f(y)\|^2$$

$$\le \frac{1}{L} \|\nabla f(x) - \nabla f(y)\|^2$$

$$\le \langle x - y, \nabla f(x) - \nabla f(y) \rangle$$

$$= \langle \gamma \nabla f_\xi(x), \nabla f(x) - \nabla f(y) \rangle$$

$$= \gamma \left( \langle \nabla f_\xi(x), \nabla f(x) \rangle - \langle \nabla f(x), \nabla f(y) \rangle + \langle \nabla f(x) - \nabla f_\xi(x), \nabla f(y) \rangle \right).$$

Taking expectation of the both sides, we obtain

$$\mathbb{E}_\xi\left[\frac{\gamma}{2} \|\nabla f(x) - \nabla f(y)\|^2\right]$$

$$\le \mathbb{E}_\xi\left[\gamma \langle \nabla f_\xi(x), \nabla f(x) \rangle - \gamma \langle \nabla f(x), \nabla f(y) \rangle + \gamma \langle \nabla f(x) - \nabla f_\xi(x), \nabla f(y) \rangle\right]$$

$$= \gamma \|\nabla f(x)\|^2 - \gamma \mathbb{E}_\xi\left[\langle \nabla f(x), \nabla f(y) \rangle\right] + \gamma \mathbb{E}_\xi\left[\langle \nabla f(x) - \nabla f_\xi(x), \nabla f(y) \rangle\right].$$

Cancelling out the terms and dividing both sides by $\frac{\gamma}{2}$, we then have

$$\mathbb{E}_\xi\left[\|\nabla f(y)\|^2\right] \le \|\nabla f(x)\|^2 + 2\mathbb{E}_\xi\left[\langle \nabla f(x) - \nabla f_\xi(x), \nabla f(y) \rangle\right]. \tag{9}$$

Now observe that

$$\mathbb{E}_\xi\left[\langle \nabla f(x) - \nabla f_\xi(x), \nabla f(y) \rangle\right] = \mathbb{E}_\xi\left[\langle \nabla f(x) - \nabla f_\xi(x), \nabla f(y) - \nabla f(x - \gamma \nabla f(x)) \rangle\right]$$

because $\nabla f(x - \gamma \nabla f(x))$ is a non-random quantity and $\mathbb{E}_\xi[\nabla f(x) - \nabla f_\xi(x)] = 0$. Then

$$\mathbb{E}_\xi\left[\langle \nabla f(x) - \nabla f_\xi(x), \nabla f(y) - \nabla f(x - \gamma \nabla f(x)) \rangle\right]$$

$$= \mathbb{E}_\xi\left[\langle \nabla f(x) - \nabla f_\xi(x), \nabla f(x - \gamma \nabla f_\xi(x)) - \nabla f(x - \gamma \nabla f(x)) \rangle\right]$$

$$\le \mathbb{E}_\xi\left[\|\nabla f(x) - \nabla f_\xi(x)\| \, \|\nabla f(x - \gamma \nabla f_\xi(x)) - \nabla f(x - \gamma \nabla f(x))\|\right]$$

$$\le \mathbb{E}_\xi\left[\|\nabla f(x) - \nabla f_\xi(x)\| \, L \, \|(x - \gamma \nabla f_\xi(x)) - (x - \gamma \nabla f(x))\|\right]$$

$$= \gamma L \mathbb{E}_\xi\left[\|\nabla f(x) - \nabla f_\xi(x)\|^2\right]$$

$$= \gamma L \rho^2,$$

and plugging this into (9) completes the proof.

$\square$

**Lemma B.2.** Let $f \colon \mathbb{R}^m \to \mathbb{R}$ be convex and $L$-smooth and let the stochastic gradient oracle $\nabla f_\xi(\cdot)$ satisfy (8). Let $x_0 \in \mathbb{R}^m$ be any initial point, $0 < \gamma \le \frac{2}{L}$, and $x_1, \dots, x_t$ be a sequence generated by the stochastic gradient descent algorithm

$$x_{s+1} = x_s - \gamma \nabla f_{\xi_s}(x_s)$$

for $s = 0, \dots, t-1$. Then we have

$$\mathbb{E}\left[\|\nabla f(x_s)\|^2\right] \le \|\nabla f(x_0)\|^2 + 2s\gamma L\rho^2$$

for $s = 0, \dots, t-1$.

*Proof.* Apply Lemma B.1 recursively and use the tower rule (law of total expectation).

$\square$

**Lemma B.3.** Let $f \colon \mathbb{R}^m \to \mathbb{R}$ be $L$-smooth and let $x_0, \dots, x_t$ be a sequence generated by stochastic gradient descent

$$x_{s+1} = x_s - \gamma \nabla f_{\xi_s}(x_s)$$

where the stochastic gradient oracle satisfies (8). Let $\hat{x}_0, \dots, \hat{x}_t$ be generated via *deterministic* gradient descent

$$\hat{x}_{s+1} = \hat{x}_s - \gamma \nabla f(\hat{x}_s)$$

where $\hat{x}_0 = x_0$. Then, provided that $0 < \gamma \le \frac{1}{L(t-1)}$, we have

$$\|x_t - \hat{x}_t\| \le 3\gamma \sum_{s=0}^{t-1} \|\nabla f_{\xi_s}(x_s) - \nabla f(x_s)\|.$$

**Remark.** Note that this result only assumes $L$-smoothness of $f$ (which is $L$-Lipschitz continuity of $\nabla f$) and does not require convexity.

*Proof.* When $t = 1$, we have $\|x_t - \hat{x}_t\| = \gamma \|\nabla f_{\xi_0}(x_0) - \nabla f(x_0)\|$ as $x_0 = \hat{x}_0$.

Now assume $t > 1$. Observe that

$$
\begin{aligned}
x_t - \hat{x}_t &= (x_{t-1} - \hat{x}_{t-1}) - \gamma\left(\nabla f_{\xi_{t-1}}(x_{t-1}) - \nabla f(\hat{x}_{t-1})\right) \\
&= (x_{t-1} - \hat{x}_{t-1}) - \gamma\left(\nabla f_{\xi_{t-1}}(x_{t-1}) - \nabla f(x_{t-1})\right) + \gamma\left(\nabla f(x_{t-1}) - \nabla f(\hat{x}_{t-1})\right)
\end{aligned}
$$

and therefore,

$$
\begin{aligned}
\|x_t - \hat{x}_t\| &\le \|x_{t-1} - \hat{x}_{t-1}\| + \gamma\left\|\nabla f_{\xi_{t-1}}(x_{t-1}) - \nabla f(x_{t-1})\right\| + \gamma\left\|\nabla f(x_{t-1}) - \nabla f(\hat{x}_{t-1})\right\| \\
&\le (1 + \gamma L)\|x_{t-1} - \hat{x}_{t-1}\| + \gamma\left\|\nabla f_{\xi_{t-1}}(x_{t-1}) - \nabla f(x_{t-1})\right\|
\end{aligned}
$$

where the last inequality uses the $L$-smoothness assumption. Now unrolling the recursion and using the fact $\|x_0 - \hat{x}_0\| = 0$ we obtain

$$
\begin{aligned}
\|x_t - \hat{x}_t\| &\le \sum_{s=0}^{t-1} \gamma(1 + \gamma L)^{t-s-1} \|\nabla f_{\xi_s}(x_s) - \nabla f(x_s)\| \\
&\le \gamma\left(1 + \frac{1}{t-1}\right)^{t-1} \sum_{s=0}^{t-1} \|\nabla f_{\xi_s}(x_s) - \nabla f(x_{s-1})\| \\
&\le 3\gamma \sum_{s=0}^{t-1} \|\nabla f_{\xi_s}(x_s) - \nabla f(x_s)\|.
\end{aligned}
$$

$\square$

**Lemma B.4.** Under the assumptions of Lemma B.3, we have

$$\mathbb{E}\left[\langle \nabla f_{\xi_0}(x_0) - \nabla f(x_0), \nabla f(x_t)\rangle\right] \le 3t\gamma L\rho^2.$$

*Proof.* Observe that because $\hat{x}_t$ as defined in Lemma B.3 is a non-random quantity and $\mathbb{E}[\nabla f_{\xi_0}(x_0) - \nabla f(x_0)] = 0$, we have

$$\begin{aligned}
&\mathbb{E}\left[\langle \nabla f_{\xi_0}(x_0) - \nabla f(x_0), \nabla f(x_t)\rangle\right] \\
&= \mathbb{E}\left[\langle \nabla f_{\xi_0}(x_0) - \nabla f(x_0), \nabla f(x_t) - \nabla f(\hat{x}_t)\rangle\right] \\
&\le \mathbb{E}\left[\|\nabla f_{\xi_0}(x_0) - \nabla f(x_0)\| \, \|\nabla f(x_t) - \nabla f(\hat{x}_t)\|\right] \\
&\le \mathbb{E}\left[\|\nabla f_{\xi_0}(x_0) - \nabla f(x_0)\| \, L \, \|x_t - \hat{x}_t\|\right] \\
&\le 3\gamma L \, \mathbb{E}\left[\|\nabla f_{\xi_0}(x_0) - \nabla f(x_0)\| \sum_{s=0}^{t-1} \|\nabla f_{\xi_s}(x_s) - \nabla f(x_s)\|\right] \\
&\le 3\gamma L \, \mathbb{E}\left[\sum_{s=0}^{t-1} \left(\frac{\|\nabla f_{\xi_0}(x_0) - \nabla f(x_0)\|^2}{2} + \frac{\|\nabla f_{\xi_s}(x_s) - \nabla f(x_s)\|^2}{2}\right)\right] \\
&\le 3t\gamma L\rho^2.
\end{aligned}$$

$\square$

**Lemma B.5.** Under the assumptions of Lemma B.3, we have

$$\mathbb{E}\left[\|x_0 - x_t\|^2\right] \le \gamma^2 \mathbb{E}\left[\left\|\sum_{s=0}^{t-1} \nabla f(x_s)\right\|^2\right] + \gamma^2 t\rho^2 + (t-1)t(t+1)\gamma^3 L\rho^2.$$

*Proof.* In the case $t = 1$, we have

$$\mathbb{E}\left[\|x_0 - x_1\|^2\right] = \gamma^2 \mathbb{E}_{\xi_0}\left[\|\nabla f_{\xi_0}(x_0)\|^2\right] \le \gamma^2\rho^2 + \gamma^2 \|\nabla f(x_0)\|^2,$$

which is the desired statement. Now we use induction on $t$. Suppose that the result holds for any initial point and $t$ steps of SGD. Consider a sequence $x_0, \ldots, x_{t+1}$ generated via SGD with initial point $x_0$ and step-size $\gamma > 0$. Observe that

$$\begin{aligned}
&\mathbb{E}\left[\|x_0 - x_{t+1}\|^2\right] \\
&= \gamma^2 \mathbb{E}\left[\left\|\sum_{s=0}^{t} \nabla f_{\xi_s}(x_s)\right\|^2\right] \\
&= \gamma^2 \mathbb{E}\left[\left\|\sum_{s=0}^{t-1} \nabla f_{\xi_s}(x_s)\right\|^2 + \mathbb{E}_{\xi_t}\left[2\left\langle \nabla f_{\xi_t}(x_t), \sum_{s=0}^{t-1} \nabla f_{\xi_s}(x_s)\right\rangle + \|\nabla f_{\xi_t}(x_t)\|^2 \,\bigg|\, x_t\right]\right] \\
&\le \gamma^2 \mathbb{E}\left[\left\|\sum_{s=0}^{t-1} \nabla f_{\xi_s}(x_s)\right\|^2 + 2\left\langle \nabla f(x_t), \sum_{s=0}^{t-1} \nabla f_{\xi_s}(x_s)\right\rangle + \|\nabla f(x_t)\|^2 + \rho^2\right] \qquad (10)
\end{aligned}$$

where the third line uses the tower rule. Now observe that for $s = 0, \ldots, t-1$,

$$\begin{aligned}
\mathbb{E}\left[\langle \nabla f(x_t), \nabla f_{\xi_s}(x_s)\rangle\right] &= \mathbb{E}\left[\langle \nabla f(x_t), \nabla f(x_s)\rangle\right] + \mathbb{E}\left[\langle \nabla f(x_t), \nabla f_{\xi_s}(x_s) - \nabla f(x_s)\rangle\right] \\
&= \mathbb{E}\left[\langle \nabla f(x_t), \nabla f(x_s)\rangle\right] + \mathbb{E}\left[\mathbb{E}\left[\langle \nabla f_{\xi_s}(x_s) - \nabla f(x_s), \nabla f(x_t)\rangle \,|\, x_s\right]\right] \\
&\le \mathbb{E}\left[\langle \nabla f(x_t), \nabla f(x_s)\rangle\right] + 3(t-s)\gamma L\rho^2
\end{aligned}$$

where the last inequality uses Lemma B.4 (with $x_s$ regarded as initial point of the stochastic gradient descent). Now we apply this inequality and the induction hypothesis to (10):

$$\mathbb{E}\left[\|x_0 - x_{t+1}\|^2\right]$$

$$\leq \gamma^2 \mathbb{E}\left[\left\|\sum_{s=0}^{t-1} \nabla f(x_s)\right\|^2 + t\rho^2 + (t-1)t(t+1)\gamma L\rho^2\right.$$

$$\left. + \sum_{s=0}^{t-1}\left(2\langle \nabla f(x_t), \nabla f(x_s)\rangle + 6(t-s)\gamma L\rho^2\right) + \|\nabla f(x_t)\|^2 + \rho^2\right]$$

$$= \gamma^2\left(t\rho^2 + (t-1)t(t+1)\gamma L\rho^2 + 3t(t+1)\gamma L\rho^2 + \rho^2\right)$$

$$+ \gamma^2 \mathbb{E}\left[\left\|\sum_{s=0}^{t-1} \nabla f(x_s)\right\|^2 + 2\left\langle \sum_{s=0}^{t-1} \nabla f(x_s), \nabla f(x_t)\right\rangle + \|\nabla f(x_t)\|^2\right]$$

$$= \gamma^2(t+1)\rho^2 + t(t+1)(t+2)\gamma^3 L\rho^2 + \gamma^2 \mathbb{E}_{\xi_0,\ldots,\xi_{t-1}}\left[\left\|\sum_{s=0}^{t} \nabla f(x_s)\right\|^2\right]$$

where for the first equality we use $\sum_{s=0}^{t-1} 6(t-s) = 3t(t+1)$. This completes the induction. $\qquad\square$

**Lemma B.6.** Let $f\colon \mathbb{R}^m \to \mathbb{R}$ be convex and $L$-smooth, and let $x_0 \in \mathbb{R}^m$ be any initial point. Let $x_1, \ldots, x_t$ be generated by stochastic gradient descent

$$x_{s+1} = x_s - \gamma \nabla f_{\xi_s}(x_s)$$

with $0 < \gamma \leq \frac{1}{L}\min\left\{1, \frac{1}{t-1}\right\}$. Then

$$\mathbb{E}\left[\|x_0 - x_t\|^2\right] \leq \gamma^2 t^2 \|\nabla f(x_0)\|^2 + \gamma^2 t(1 + 2(t-1)(t+1)\gamma L)\rho^2.$$

*Proof.* Lemma B.5 gives

$$\mathbb{E}\left[\|x_0 - x_t\|^2\right] \leq \gamma^2 \mathbb{E}\left[\left\|\sum_{s=0}^{t-1} \nabla f(x_s)\right\|^2\right] + \gamma^2 t\rho^2 + (t-1)t(t+1)\gamma^3 L\rho^2. \qquad(11)$$

Next, by Jensen's inequality and Lemma B.2,

$$\mathbb{E}\left[\left\|\sum_{s=0}^{t-1} \nabla f(x_s)\right\|^2\right] \leq t\sum_{s=0}^{t-1} \mathbb{E}\left[\|\nabla f(x_s)\|^2\right]$$

$$\leq t\sum_{s=0}^{t-1}\left(\|\nabla f(x_0)\|^2 + 2s\gamma L\rho^2\right)$$

$$\leq t^2 \|\nabla f(x_0)\|^2 + (t-1)t(t+1)\gamma L\rho^2$$

where the last inequality uses $\sum_{s=0}^{t-1} 2s = t(t-1) \leq (t-1)(t+1)$. Applying the above inequality to (11) we obtain the desired result. $\qquad\square$

### B.3 PROOFS OF LEMMAS 3.9 AND 3.8

*Proof of Lemma 3.9.* Observe that given $\mathbf{x}_{\tau p}$, the sequence $x_{\tau p}^i, \ldots, x_{\tau(p+1)}^i$ is a sequence generated via stochastic gradient descent

$$x_{j+1}^i = x_j^i - \gamma \nabla f_{i,\xi_j^i}(x_j^i; x_{\tau p}^{-i})$$

for the $L_i$-smooth convex function $f_i(\cdot; x^{-i}_{\tau p})$, with $x^i_{\tau p}$ as initial point, using the stochastic oracle $\nabla f_{i,\xi^i}(\cdot)$ satisfying *(BV)* (unbiased estimator of $\nabla f(\cdot)$ with uniformly bounded variance $\leq \sigma^2_i$). Therefore, we can apply Lemma B.2 with

$$f(\cdot) = f_i(\cdot; x^{-i}_{\tau p}), \qquad \rho^2 = \sigma^2_i, \qquad x_0 = x^i_{\tau p}, \qquad x_s = x^i_j$$

and this immediately proves the desired statement. (Note that $s$ is replaced with $j - \tau p$ because $x^i_j$ is obtained by $j - \tau p$ steps of SGD from $x^i_{\tau p}$.)

$\square$

*Proof of Lemma 3.8.* This is a direct consequence of Lemma B.6 with same choice of $f, \rho^2, x_0$ as in the proof of Lemma 3.9 and $x_t = x^i_k$.

$\square$

### B.4 REMAINING DETAILS IN PROOF OF THEOREM 3.4

Note that the step-size condition of Lemma 3.8 is satisfied by our step-size selection, as $\gamma < \frac{2}{\ell\tau + 2(\tau-1)L_{\max}\sqrt{\kappa}} \leq \frac{1}{L_{\max}(\tau-1)}$ (because $\kappa \geq 1$). Now combine Lemmas 3.7 and 3.8 to obtain

$$\mathbb{E}\left[\left\|\mathbf{x}_{\tau(p+1)} - \mathbf{x}_\star\right\|^2 \,\Big|\, \mathbf{x}_{\tau p}\right]$$

$$\leq \left\|\mathbf{x}_{\tau p} - \mathbf{x}_\star\right\|^2 - 2\gamma\tau \left\langle \mathbf{x}_{\tau p} - \mathbf{x}_\star, \mathbb{F}(\mathbf{x}_{\tau p})\right\rangle + \alpha\gamma(\tau - 1)\left\|\mathbf{x}_{\tau p} - \mathbf{x}_\star\right\|^2$$

$$+ \sum_{j=\tau p+1}^{\tau(p+1)-1} \sum_{i=1}^n \frac{\gamma L^2_i}{\alpha}\left(\gamma^2(j - \tau p)^2 \left\|\nabla f(x^i_{\tau p}; x^{-i}_{\tau p})\right\|^2 + \gamma^2(j - \tau p)\left(1 + 2(j - \tau p - 1)(j - \tau p + 1)\gamma L_i\right)\sigma^2_i\right)$$

$$+ \sum_{i=1}^n \left(\gamma^2(k - \tau p)^2 \left\|\nabla f(x^i_{\tau p}; x^{-i}_{\tau p})\right\|^2 + \gamma^2(k - \tau p)\left(1 + 2(k - \tau p - 1)(k - \tau p + 1)\gamma L_i\right)\sigma^2_i\right)$$

$$\leq \left(1 + \alpha\gamma(\tau - 1)\right)\left\|\mathbf{x}_{\tau p} - \mathbf{x}_\star\right\|^2 - 2\gamma\tau\left\langle\mathbf{x}_{\tau p} - \mathbf{x}_\star, \mathbb{F}(\mathbf{x}_{\tau p})\right\rangle + \left(\gamma^2\tau^2 + \frac{\gamma^3 L^2_{\max}\tau^2(\tau-1)}{3\alpha}\right)\left\|\mathbb{F}(\mathbf{x}_{\tau p})\right\|^2$$

$$+ \gamma^2\tau\left(1 + (\tau - 1)\gamma L_{\max}\left(2(\tau + 1) + \frac{L_{\max}}{2\alpha} + \frac{\gamma L^2_{\max}}{2\alpha}(\tau+1)^2\right)\right)\sigma^2$$

$$\tag{12}$$

where for the last inequality, we replace all occurrences of $L_i$'s by $L_{\max} = \max\{L_1, \ldots, L_n\}$ and use the identities

$$\sigma^2 = \sum_{i=1}^n \sigma^2_i, \quad \left\|\mathbb{F}(\mathbf{x}_{\tau p})\right\|^2 = \sum_{i=1}^n \left\|\nabla f_i(x^i_{\tau p}; x^{-i}_{\tau p})\right\|^2$$

to eliminate the summations $\sum_{i=1}^n$ and use the following elementary summation results:

$$\sum_{j=\tau p+1}^{\tau(p+1)-1} (j - \tau p)^2 = \frac{(\tau-1)\tau(2\tau-1)}{6} \leq \frac{(\tau-1)\tau^2}{3}$$

$$\sum_{j=\tau p+1}^{\tau(p+1)-1} (j - \tau p) = \frac{(\tau-1)\tau}{2}$$

and

$$\sum_{j=\tau p+1}^{\tau(p+1)-1} (j - \tau p - 1)(j - \tau p)(j - \tau p + 1) = \frac{(\tau-2)(\tau-1)\tau(\tau+1)}{2} \leq \frac{(\tau-1)\tau(\tau+1)^2}{2}.$$

Now in (12), we use the Assumption *(SCO)* to bound

$$- 2\gamma\tau \langle \mathbf{x}_{\tau p} - \mathbf{x}_\star, \mathbb{F}(\mathbf{x}_{\tau p}) \rangle + \left( \gamma^2\tau^2 + \frac{\gamma^3 L_{\max}^2 \tau^2 (\tau - 1)}{3\alpha} \right) \|\mathbb{F}(\mathbf{x}_{\tau p})\|^2$$

$$\leq - \left( 2\gamma\tau - \ell \left( \gamma^2\tau^2 + \frac{\gamma^3 L_{\max}^2 \tau^2 (\tau - 1)}{3\alpha} \right) \right) \langle \mathbf{x}_{\tau p} - \mathbf{x}_\star, \mathbb{F}(\mathbf{x}_{\tau p}) \rangle$$

$$= -\gamma\tau \left( 2 - \gamma\ell\tau - \frac{\gamma^2 \ell L_{\max}^2 \tau(\tau - 1)}{3\alpha} \right) \langle \mathbf{x}_{\tau p} - \mathbf{x}_\star, \mathbb{F}(\mathbf{x}_{\tau p}) \rangle . \qquad (13)$$

Provided that

$$2 - \gamma\ell\tau - \frac{\gamma^2 \ell L_{\max}^2 \tau(\tau - 1)}{3\alpha} \geq 0, \qquad (14)$$

we can again upper bound (13) using the Assumption *(QSM)*:

$$- \gamma\tau \left( 2 - \gamma\ell\tau - \frac{\gamma^2 \ell L_{\max}^2 \tau(\tau - 1)}{3\alpha} \right) \langle \mathbf{x}_{\tau p} - \mathbf{x}_\star, \mathbb{F}(\mathbf{x}_{\tau p}) \rangle$$

$$\leq -\gamma\tau \left( 2 - \gamma\ell\tau - \frac{\gamma^2 \ell L_{\max}^2 \tau(\tau - 1)}{3\alpha} \right) \mu \|\mathbf{x}_{\tau p} - \mathbf{x}_\star\|^2 .$$

We plug this into (12) and rearrange the terms to obtain

$$\mathbb{E} \left[ \|\mathbf{x}_{\tau(p+1)} - \mathbf{x}_\star\|^2 \,\Big|\, \mathbf{x}_{\tau p} \right]$$

$$\leq \left( 1 + \alpha\gamma(\tau - 1) - \gamma\tau \left( 2 - \gamma\tau\ell - \frac{\gamma^2 \ell L_{\max}^2 \tau(\tau - 1)}{3\alpha} \right) \mu \right) \|\mathbf{x}_{\tau p} - \mathbf{x}_\star\|^2$$

$$+ \gamma^2\tau \left( 1 + (\tau - 1)\gamma L_{\max} \left( 2(\tau + 1) + \frac{L_{\max}}{2\alpha} + \frac{\gamma L_{\max}^2}{2\alpha}(\tau + 1)^2 \right) \right) \sigma^2 . \qquad (15)$$

Now, we optimize the coefficient of the $\|\mathbf{x}_{\tau p} - \mathbf{x}_\star\|^2$ term in (15) by taking

$$\alpha = \operatorname*{argmin}_{\alpha > 0} \alpha\gamma(\tau - 1) + \frac{\gamma^3 \ell L_{\max}^2 \tau^2 (\tau - 1)\mu}{3\alpha} = \gamma\tau L_{\max} \sqrt{\frac{\ell\mu}{3}} .$$

With this choice of $\alpha$, the bound (15) becomes

$$\mathbb{E} \left[ \|\mathbf{x}_{\tau(p+1)} - \mathbf{x}_\star\|^2 \,\Big|\, \mathbf{x}_{\tau p} \right]$$

$$\leq \left( 1 - \gamma\tau\mu \left( 2 - \gamma\ell\tau - 2(\tau - 1)\gamma L_{\max} \sqrt{\frac{\ell}{3\mu}} \right) \right) \|\mathbf{x}_{\tau p} - \mathbf{x}_\star\|^2$$

$$+ \gamma^2\tau \left( 1 + (\tau - 1)\gamma L_{\max} \left( 2(\tau + 1) + \frac{1}{2\gamma\tau\sqrt{\ell\mu/3}} + \frac{L_{\max}(\tau + 1)^2}{2\tau\sqrt{\ell\mu/3}} \right) \right) \sigma^2$$

$$\leq (1 - \gamma\tau\mu\zeta) \|\mathbf{x}_{\tau p} - \mathbf{x}_\star\|^2 + \gamma^2\tau\sigma^2 \left( 1 + (\tau - 1) \left( 4\gamma\tau L_{\max} + \frac{L_{\max}}{2\tau\sqrt{\ell\mu/3}} + \frac{\gamma\tau L_{\max}^2}{\sqrt{\ell\mu/3}} \right) \right)$$

$$\qquad\qquad (16)$$

where for the last inequality, we use $\tau + 1 \leq 2\tau$ and make the substitution

$$\zeta = 2 - \gamma\ell\tau - 2(\tau - 1)\gamma L_{\max} \sqrt{\frac{\ell}{3\mu}} = 2 - \gamma\ell\tau - 2(\tau - 1)\gamma L_{\max} \sqrt{\kappa/3} .$$

Note that with our choice $\alpha = \gamma\tau L_{\max}\sqrt{\frac{\ell\mu}{3}}$ and $0 < \gamma < \frac{2}{\ell\tau + 2(\tau-1)L_{\max}\sqrt{\kappa}}$, the condition (14) is satisfied because

$$2 - \gamma\ell\tau - \frac{\gamma^2 \ell L_{\max}^2 \tau(\tau - 1)}{3\alpha} \geq 2 - \gamma\ell\tau - \frac{\gamma^2 \ell L_{\max}^2 \tau(\tau - 1)}{3\alpha}$$

$$= 2 - \gamma\ell\tau - (\tau - 1)\gamma L_{\max} \sqrt{\frac{\ell}{3\mu}}$$

$$\geq 2 - \gamma \left( \ell\tau + (\tau - 1)L_{\max}\sqrt{\kappa} \right) > 0 .$$

Finally, unrolling the recursion (16) using the following simple lemma, with $a_p = \mathbb{E}\left[\|\mathbf{x}_{\tau p} - \mathbf{x}_\star\|^2\right]$, $A = \tau\mu\zeta$ and

$$B = \tau\sigma^2 \left(1 + (\tau - 1)\left(4\gamma\tau L_{\max} + \frac{L_{\max}}{2\tau\sqrt{\ell\mu/3}} + \frac{\gamma\tau L_{\max}^2}{\sqrt{\ell\mu/3}}\right)\right)$$

gives the desired rate. (Note that $\gamma A = \gamma\tau\mu\zeta \leq \gamma\tau\mu(2 - \gamma\ell\tau) \leq \gamma\ell\tau(2 - \gamma\ell\tau) \leq 1$.)

**Lemma B.7.** Let $\gamma, A, B > 0$ with $\gamma A \leq 1$. If a sequence $a_0, \ldots, a_R \in \mathbb{R}$ satisfies

$$a_{p+1} \leq (1 - \gamma A)a_p + \gamma^2 B$$

for $p = 0, \ldots, R - 1$, then $a_R \leq (1 - \gamma A)^R a_0 + \frac{\gamma B}{A}$.

*Proof of Lemma B.7.* As there is nothing to prove if $\gamma A = 1$, suppose $\gamma A < 1$. Recursively applying the given inequality we have

$$a_R \leq (1 - \gamma A)a_{R-1} + \gamma^2 B \leq \cdots \leq (1 - \gamma A)^R a_0 + \gamma^2 B \sum_{p=0}^{R-1}(1 - \gamma A)^p.$$

Now apply the bound $\sum_{p=0}^{R-1}(1 - \gamma A)^p \leq \sum_{p=0}^{\infty}(1 - \gamma A)^p = \frac{1}{1 - (1 - \gamma A)} = \frac{1}{\gamma A}$ to the above inequality. $\qquad\square$

### B.5 PROOF OF COROLLARY 3.5

First, because $\eta > \kappa\tau$, we have

$$\gamma < \frac{1}{\mu\kappa\tau\left(1 + \frac{2L_{\max}}{\sqrt{\ell\mu}}\right)} = \frac{1}{\ell\tau\left(1 + \frac{2L_{\max}}{\sqrt{\ell\mu}}\right)} \leq \frac{1}{\ell\tau + 2(\tau - 1)L_{\max}\sqrt{\frac{\ell}{\mu}}} = \frac{1}{\ell\tau + 2(\tau - 1)L_{\max}\sqrt{\kappa}}.$$

Hence we can apply Theorem 3.4. Now observe that $\zeta > 2 - \gamma\left(\ell\tau + 2(\tau - 1)L_{\max}\sqrt{\kappa}\right) > 1$, and $(1 - u)^R \leq e^{-uR}$ for $u < 1$, so

$$(1 - \gamma\tau\mu\zeta)^R \leq e^{-\gamma\mu\zeta\tau R} \leq e^{-\gamma\mu T} = e^{-2\log\eta} = \frac{1}{\eta^2} = \frac{4(\log\eta)^2(1 + 2q)^2}{T^2} = \tilde{\mathcal{O}}\left(\frac{(1 + q)^2}{T^2}\right)$$

where we use $T = 2(1 + 2q)\eta\log\eta$ and remove the factor $\log\eta < \log T$ within the $\tilde{\mathcal{O}}$ notation. Next, for the terms proportional to $\sigma^2$, we have

$$\left(1 + (\tau - 1)\left(4\gamma\tau L_{\max} + \frac{L_{\max}}{2\tau\sqrt{\ell\mu/3}} + \frac{\gamma\tau L_{\max}^2}{\sqrt{\ell\mu/3}}\right)\right)\frac{\gamma\sigma^2}{\mu\zeta}$$

$$\leq \frac{\gamma\sigma^2}{\mu}\left(1 + \tau\left(4\gamma\tau L_{\max} + \frac{\sqrt{3}q}{2\tau} + \sqrt{3}\gamma\tau L_{\max}q\right)\right)$$

$$\leq \frac{\gamma\sigma^2}{\mu}\left(1 + \frac{\sqrt{3}q}{2}\right) + \frac{\gamma^2\tau^2 L_{\max}\sigma^2}{\mu}(4 + \sqrt{3}q)$$

$$= \frac{\sigma^2(1 + \sqrt{3}q/2)}{\mu^2\eta(1 + 2q)} + \frac{\tau^2 L_{\max}\sigma^2(4 + \sqrt{3}q)}{\mu^3\eta^2(1 + 2q)^2}$$

$$= \tilde{\mathcal{O}}\left(\frac{(1 + q)\sigma^2}{\mu^2 T} + \frac{(1 + q)\tau^2 L_{\max}\sigma^2}{\mu^3 T^2}\right).$$

Combining these with Theorem 3.4 we arrive at the desired conclusion.

### B.6 PROOF OF THEOREM 3.6

Note that we use constant step-size $\gamma_k \equiv \gamma_{\tau p}$ within each communication round $p$, i.e., for $\tau p \leq k \leq \tau(p+1) - 1$, so we can apply the bound (16) from the proof of Theorem 3.4, provided that

$$\gamma_{\tau p} \leq \frac{1}{\ell \tau + 2(\tau - 1) L_{\max} \sqrt{\kappa}}.$$

This clearly holds true when $p < 2(1+2q)\kappa - 1$, and when $p \geq 2(1+2q)\kappa - 1$ then

$$\gamma_{\tau p} = \frac{1}{\tau \mu} \frac{2p+1}{(p+1)^2} < \frac{1}{\tau \mu} \frac{2}{p+1} \leq \frac{1}{\tau \mu} \frac{1}{(1+2q)\kappa} = \frac{1}{\ell \tau + 2\tau L_{\max} \sqrt{\kappa}}$$

so we see that the step-size condition is satisfied. Furthermore we have

$$\zeta_{\tau p} = 2 - \gamma_{\tau p} \ell \tau - 2(\tau - 1) \gamma_{\tau p} L_{\max} \sqrt{\kappa/3} > 1,$$

so (16), with $q = \frac{L_{\max}}{\sqrt{\ell \mu}}$ and taking expectation with respect to $\mathbf{x}_{\tau p}$, gives

$$\mathbb{E}\left[\left\|\mathbf{x}_{\tau(p+1)} - \mathbf{x}_\star\right\|^2\right] \leq (1 - \gamma_{\tau p} \tau \mu \zeta_{\tau p}) \mathbb{E}\left[\left\|\mathbf{x}_{\tau p} - \mathbf{x}_\star\right\|^2\right]$$

$$+ \gamma_{\tau p}^2 \tau \sigma^2 \left(1 + (\tau - 1)\left(\gamma_{\tau p} \tau L_{\max}(4 + \sqrt{3}q) + \frac{\sqrt{3}}{2\tau q}\right)\right)$$

$$\leq (1 - \gamma_{\tau p} \tau \mu) \mathbb{E}\left[\left\|\mathbf{x}_{\tau p} - \mathbf{x}_\star\right\|^2\right] + (1+q)\gamma_{\tau p}^2 \tau \sigma^2 + 4(1+q)\gamma_{\tau p}^3 \tau^2 (\tau - 1) L_{\max} \sigma^2.$$

$$(17)$$

For $p \geq 2(1+2q)\kappa - 1$, plugging in $\gamma_{\tau p} = \frac{1}{\tau \mu} \frac{2p+1}{(p+1)^2}$ we obtain

$$\mathbb{E}\left[\left\|\mathbf{x}_{\tau(p+1)} - \mathbf{x}_\star\right\|^2\right] \leq \frac{p^2}{(p+1)^2} \mathbb{E}\left[\left\|\mathbf{x}_{\tau p} - \mathbf{x}_\star\right\|^2\right] + \frac{(2p+1)^2 \sigma^2 (1+q)}{\tau \mu^2 (p+1)^4}\left(1 + \frac{4(\tau - 1) L_{\max}(2p+1)}{\mu(p+1)^2}\right).$$

Multiplying $\tau^2(p+1)^2$ to both sides and upper-bounding $\frac{2p+1}{p+1} \leq 2$, we obtain

$$(\tau(p+1))^2 \mathbb{E}\left[\left\|\mathbf{x}_{\tau(p+1)} - \mathbf{x}_\star\right\|^2\right] \leq (\tau p)^2 \mathbb{E}\left[\left\|\mathbf{x}_{\tau p} - \mathbf{x}_\star\right\|^2\right] + \frac{4(1+q)\tau \sigma^2}{\mu^2}\left(1 + \frac{8(\tau - 1) L_{\max}}{\mu(p+1)}\right).$$

Let $p_0 = \lceil 2(1+2q)\kappa - 1 \rceil$. Chaining the above inequality for $p = p_0, \ldots, R-1$ gives

$$(\tau R)^2 \mathbb{E}\left[\left\|\mathbf{x}_{\tau R} - \mathbf{x}_\star\right\|^2\right]$$

$$\leq (\tau p_0)^2 \mathbb{E}\left[\left\|\mathbf{x}_{\tau p_0} - \mathbf{x}_\star\right\|^2\right] + \frac{4(1+q)\tau(R - p_0)\sigma^2}{\mu^2} + \frac{32(1+q)\tau(\tau - 1) L_{\max} \sigma^2}{\mu^3} \sum_{p=p_0}^{R-1} \frac{1}{p+1}$$

$$\leq (\tau p_0)^2 \mathbb{E}\left[\left\|\mathbf{x}_{\tau p_0} - \mathbf{x}_\star\right\|^2\right] + \frac{4(1+q)\tau(R - p_0)\sigma^2}{\mu^2} + \frac{32(1+q)\tau^2 L_{\max} \sigma^2 \log(R/p_0)}{\mu^3}$$

where we use $\sum_{p=p_0}^{R-1} \frac{1}{p+1} \leq \int_{p_0}^R \frac{dp}{p} = \log \frac{R}{p_0}$. Now substitute $T = \tau R$ using the upper bounds $\tau(R - p_0) \leq \tau R = T$ and $\log(R/p_0) \leq \log T$, we can write

$$T^2 \mathbb{E}\left[\left\|\mathbf{x}_T - \mathbf{x}_\star\right\|^2\right] \leq (\tau p_0)^2 \mathbb{E}\left[\left\|\mathbf{x}_{\tau p_0} - \mathbf{x}_\star\right\|^2\right] + \frac{4(1+q)T\sigma^2}{\mu^2} + \frac{32(1+q)\tau^2 L_{\max} \sigma^2 \log T}{\mu^3}.$$

$$(18)$$

As $\gamma_k$ is constantly $\gamma_0 = \frac{1}{\ell \tau(1+2q)}$ over rounds $p = 0, \ldots, p_0 - 1$, we can directly apply Theorem 3.4 with $R = p_0$ and similar simplification of the $\sigma^2$-terms as in (17) to bound

$$\mathbb{E}\left[\left\|\mathbf{x}_{\tau p_0} - \mathbf{x}_\star\right\|^2\right] \leq \left(1 - \frac{\mu}{\ell(1+2q)}\right)^{p_0} \left\|\mathbf{x}_0 - \mathbf{x}_\star\right\|^2 + \frac{(1+q)\gamma_0 \sigma^2}{\mu}\left(1 + 4\gamma_0 \tau(\tau - 1) L_{\max}\right)$$

$$\leq \left(1 - \frac{1}{\kappa(1+2q)}\right)^{\kappa(1+2q)} \left\|\mathbf{x}_0 - \mathbf{x}_\star\right\|^2 + \frac{\sigma^2}{\ell \mu \tau}\left(1 + \frac{4(\tau - 1) L_{\max}}{\ell(1+2q)}\right)$$

$$\leq \frac{\left\|\mathbf{x}_0 - \mathbf{x}_\star\right\|^2}{e} + \frac{\sigma^2}{\ell \mu \tau}\left(1 + \frac{2\tau}{\sqrt{\kappa}}\right),$$

where the second line uses $p_0 \geq 2(1 + 2q)\kappa - 1 \geq \kappa(1 + 2q)$, and the third line uses the bound $\left(1 - \frac{1}{t}\right)^t \leq \frac{1}{e}$ for $t > 1$ and $\frac{4(\tau - 1)L_{\max}}{\ell(1+2q)} \leq \frac{4q\tau\sqrt{\ell\mu}}{\ell(1+2q)} \leq 2\tau\sqrt{\frac{\mu}{\ell}} = \frac{2\tau}{\sqrt{\kappa}}$. Now plugging this into (18) and dividing both sides by $T^2$ we obtain

$$\mathbb{E}\left[\|\mathbf{x}_T - \mathbf{x}_\star\|^2\right]$$

$$\leq \frac{p_0^2\tau^2\|\mathbf{x}_0 - \mathbf{x}_\star\|^2}{eT^2} + \frac{\tau p_0^2\sigma^2}{\ell\mu T^2}\left(1 + \frac{2\tau}{\sqrt{\kappa}}\right) + \frac{4(1+q)\sigma^2}{\mu^2 T} + \frac{32(1+q)\tau^2 L_{\max}\sigma^2\log T}{\mu^3 T^2}$$

$$\leq \frac{4(1+2q)^2\kappa^2\tau^2\|\mathbf{x}_0 - \mathbf{x}_\star\|^2}{eT^2} + \frac{4(1+q)\sigma^2}{\mu^2 T} + \frac{4(1+2q)^2\kappa\tau\sigma^2}{\mu^2 T^2}\left(1 + \frac{2\tau}{\sqrt{\kappa}}\right) + \frac{32(1+q)\tau^2 L_{\max}\sigma^2\log T}{\mu^3 T^2}.$$

which is the desired result.

# C  DETAILS OF NUMERICAL EXPERIMENTS

## C.1  QUADRATIC MINIMAX GAME

**Data Generation.**   Here, we generate the matrices $\mathbf{A}_m, \mathbf{B}_m, \mathbf{C}_m \in \mathbb{R}^{d \times d}$ and vectors $a_m, c_m \in \mathbb{R}^d$ to ensure the quadratic game $f(x^1, x^2)$ is strongly convex-strongly concave and smooth. $\mathbf{A}_m$, $\mathbf{B}_m, \mathbf{C}_m$ are generated such that they are positive semi-definite and their eigenvalues lie in the interval $[\mu_A, L_A]$, $[0, L_B]$ and $[\mu_C, L_C]$ respectively (Loizou et al., 2021). In all our experiments, we generate the data with dimension $d = 10$ and for $M = 100$, where $M$ represents the number of samples. To implement the PEARL-SGD, we consider two computational nodes, one corresponding to the $x^1$ variable and the other to the $x^2$ variable.

## C.2  $n$-PLAYER GAME

**Data Generation.**   In this setup, we use $d = 10$ and $M = 100$ in all our experiments. The matrices $\mathbf{A}_{i,m}$ are generated randomly with their eigenvalues in the range $[\mu_\mathbf{A}, L_\mathbf{A}]$ (here we choose $\mu_\mathbf{A}, L_\mathbf{A} > 0$). Similarly, for $1 \le i < j \le n$, we generate the matrices $\mathbf{B}_{i,j,m}$ randomly such that their eigenvalues lie in the interval $[0, L_\mathbf{B}]$. However, for $1 \le j < i \le n$, we set $\mathbf{B}_{j,i,m} = -\mathbf{B}_{i,j,m}^\top$. This data generation procedure ensures that the operator corresponding to the objective function (5) satisfies the *(QSM)* assumption. We provide a proof below:

The operator corresponding to the $n$-Player Game (5) satisfies the quasi-strong monotonicity *(QSM)* assumption. We have

$$f_i(x^1, \ldots, x^n) = \frac{1}{2} \left\langle x^i, \mathbf{A}_i x^i \right\rangle + \left\langle a_i, x^i \right\rangle + \sum_{j \neq i} \left\langle x^i, \mathbf{B}_{i,j} x^j \right\rangle$$

for $i = 1, \ldots, n$. Then taking the partial gradient of $f_i$ with respect to $x^i$ we get

$$\nabla f_i(x^i; x^{-i}) = \mathbf{A}_i x^i + a_i + \sum_{j \neq i} \mathbf{B}_{i,j} x^j.$$

Therefore,

$$\nabla f_i(x^i; x^{-i}) - \nabla f_i(x_\star^i; x_\star^{-i}) = \left( \mathbf{A}_i x^i + a_i + \sum_{j \neq i} \mathbf{B}_{i,j} x^j \right) - \left( \mathbf{A}_i x_\star^i + a_i + \sum_{j \neq i} \mathbf{B}_{i,j} x_\star^j \right)$$

$$= \mathbf{A}_i(x^i - x_\star^i) + \sum_{j \neq i} \mathbf{B}_{i,j}(x^j - x_\star^j)$$

and

$$\left\langle \mathbb{F}(\mathbf{x}) - \mathbb{F}(\mathbf{x}_\star), \mathbf{x} - \mathbf{x}_\star \right\rangle = \sum_{i=1}^n \left\langle \nabla f_i(x^i; x^{-i}) - \nabla f_i(x_\star^i; x_\star^{-i}), x^i - x_\star^i \right\rangle$$

$$= \sum_{i=1}^n \left\langle x^i - x_\star^i, \mathbf{A}_i(x^i - x_\star^i) \right\rangle + \sum_{i=1}^n \sum_{j \neq i} \left\langle x^i - x_\star^i, \mathbf{B}_{i,j}(x^j - x_\star^j) \right\rangle$$

If $\mathbf{B}_{j,i} = -\mathbf{B}_{i,j}^\intercal$ for all $i \neq j$ then the double summation vanishes because for any $i \neq j$,

$$\left\langle x^i - x_\star^i, \mathbf{B}_{i,j}(x^j - x_\star^j) \right\rangle + \left\langle x^j - x_\star^j, \mathbf{B}_{j,i}(x^i - x_\star^i) \right\rangle = 0.$$

Then, provided that each $\mathbf{A}_i \succeq \mu I$ we see that $\mathbb{F}$ is $\mu$-QSM. (Actually it is $\mu$-strongly monotone.)

# D ADDITIONAL EXPERIMENT: MOBILE ROBOT CONTROL

Here, we consider a distributed control problem of mobile robots from (Kalyva & Psillakis, 2024). This is a multi-agent system where each robot has its own objective, depending on the positions $x^i \in \mathbb{R}^d$ (corresponding to action/strategy in our formulation of multiplayer game) of each $i$-th robot. Specifically, the objective function of the robot $i$ is:

$$f_i(\mathbf{x}) = J_{i1}(x^i) + J_{i2}(x^i; x^{-i}),$$

where $J_{i1}(x^i) = \frac{c_i}{2}\|x^i - x_{\text{anc}}^i\|^2$ represents the cost penalizing the distance of agent $i$ from some anchor point $x_{\text{anc}}^i \in \mathbb{R}^d$, and $J_{i2}(x^i; x^{-i}) = \frac{d_i}{2}\sum_{j=1}^{N}\|x^i - x^j - h_{ij}\|^2$ is the cost associated with the relative displacement between the robots' positions. The control problem finds an equilibrium of the $n$-player game, which is the concatenation of all robots' position vectors, ensuring that each robot stays close to $x_{\text{anc}}^i$ while maintaining designated displacement from other robots. We follow the choice of parameter values $c_i, d_i, x_{\text{anc}}^i, h_{ij}$ from (Kalyva & Psillakis, 2024): $n = 5$, $d = 1$, $c_i = 10 + i/6$, $d_i = i/6$,

$$\left(x_{\text{anc}}^1, x_{\text{anc}}^2, x_{\text{anc}}^3, x_{\text{anc}}^4, x_{\text{anc}}^5\right) = (1, -4, 8, -9, 13)$$

and

$$(h_{ij})_{\substack{1 \le i \le 5 \\ 1 \le j \le 5}} = \begin{pmatrix} 0 & 5 & -7 & 9 & -8 \\ -5 & 0 & -6 & 2 & -9 \\ 7 & 6 & 0 & 7 & -4 \\ -9 & -2 & -7 & 0 & -2 \\ 8 & 9 & 4 & 2 & 0 \end{pmatrix}.$$

We add Gaussian noise to the gradients to simulate stochasticity. In this setup, all our theoretical assumptions are satisfied.

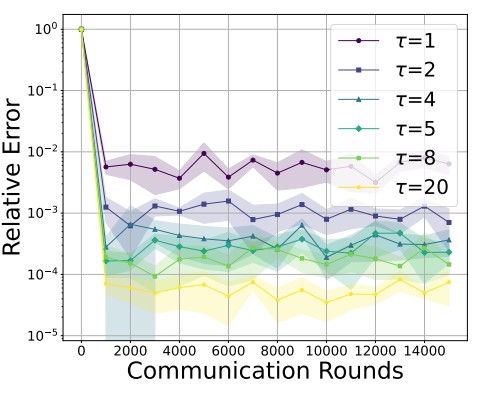

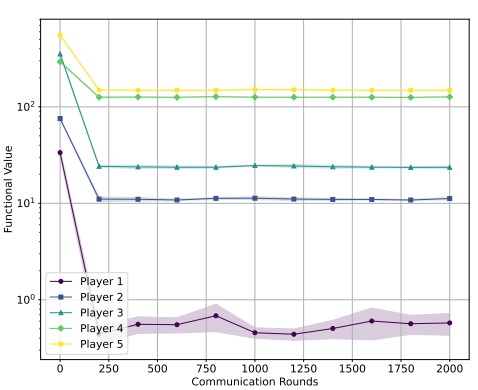

(a) Plot of relative errors $\|\mathbf{x}_{\tau p} - \mathbf{x}_\star\|^2 / \|\mathbf{x}_0 - \mathbf{x}_\star\|^2$

(b) Plot of local objectives $f_i$ ($\tau = 5$)

Figure 5: Performance of PEARL-SGD on the distributed mobile robot control problem.

We implement PEARL-SGD with synchronization intervals $\tau \in \{1, 2, 4, 5, 8, 20\}$ and the theoretical step-size $\gamma = \frac{1}{\ell\tau + L_{\max}(\tau-1)\sqrt{\kappa}}$. Figure 5a shows that with larger values of $\tau$, PEARL-SGD achieves better accuracy (in terms of distance to $\mathbf{x}_\star$) within a given number of communication rounds. This highlights the potential benefit of using local update steps in solving real-data problems formulated as multiplayer games. Figure 5b displays how the local objective values $f_i$ behave under PEARL-SGD, in the case $\tau = 5$.

# E    DISCUSSION ON THEORETICAL ASSUMPTIONS

## E.1    POSSIBLE SIMPLIFICATION OF ASSUMPTIONS: ASSUMING COCOERCIVITY OF $\mathbb{F}$

In fact, the convergence of PEARL-SGD can still be proved even if the three assumptions *(CVX)*, *(SM)* and *(SCO)* are replaced with the single assumption that $\mathbb{F}\colon \mathbb{R}^D \to \mathbb{R}^D$ is $\frac{1}{\ell}$-cocoercive, i.e.,

$$\langle \mathbb{F}(\mathbf{x}) - \mathbb{F}(\mathbf{y}), \mathbf{x} - \mathbf{y} \rangle \geq \frac{1}{\ell} \|\mathbf{x} - \mathbf{y}\|^2, \quad \forall \mathbf{x}, \mathbf{y} \in \mathbb{R}^D. \qquad \textbf{(COCO)}$$

In the subsequent paragraphs, we explain in detail why this is the case. However, we emphasize here that if we derived all convergence theory using *(COCO)* in place of *(CVX)*, *(SM)* and *(SCO)* and did not distinguish the role of $L_i$'s (the local Lipschitzness parameters from *(SM)*) from that of $\ell$, then the resulting convergence rates would have become much more pessimistic (worse) in many cases. Therefore, in our work, we choose to use the current set of assumptions. It allows us to more clearly present the tight dependency of convergence rates to $L_i$'s. Also note that assuming *(CVX)*, *(SM)* and *(SCO)* is strictly more general than assuming *(COCO)*, as we illustrate in Appendix E.2.

**(*COCO*) implies (*CVX*), (*SM*) and (*SCO*).**    Trivially, *(COCO)* implies *(SCO)*. Furthermore, if $\mathbb{F}$ is $\frac{1}{\ell}$-cocoercive, then $\mathbb{F}$ is monotone:

$$\langle \mathbb{F}(\mathbf{x}) - \mathbb{F}(\mathbf{y}), \mathbf{x} - \mathbf{y} \rangle \geq 0, \quad \forall \mathbf{x}, \mathbf{y} \in \mathbb{R}^D, \qquad (19)$$

and $\ell$-Lipschitz continuous:

$$\|\mathbb{F}(\mathbf{x}) - \mathbb{F}(\mathbf{y})\| \leq \ell \|\mathbf{x} - \mathbf{y}\|, \quad \forall \mathbf{x}, \mathbf{y} \in \mathbb{R}^D. \qquad (20)$$

In particular, for each $i = 1, \ldots, n$, we can take

$$\mathbf{x} = (x^1, \ldots, x^{i-1}, x^i, x^{i+1}, \ldots, x^n), \quad \mathbf{y} = (x^1, \ldots, x^{i-1}, y^i, x^{i+1}, \ldots, x^n) \qquad (21)$$

in (19), which gives

$$\langle \nabla f_i(x^i; x^{-i}) - \nabla f_i(y^i; x^{-i}), x^i - y^i \rangle \geq 0$$

for any $x^i, y^i \in \mathbb{R}^{d_i}$ and $x^{-i} \in \mathbb{R}^{D-d_i}$. That is, the gradient of $f_i(\cdot; x^{-i})\colon \mathbb{R}^{d_i} \to \mathbb{R}$ is a monotone operator on $\mathbb{R}^{d_i}$, and this implies that $f_i(\cdot; x^{-i})$ is convex, i.e., *(CVX)* holds. Similarly, plugging the choice (21) into (20) we obtain

$$\|\nabla f_i(x^i; x^{-i}) - \nabla f_i(y^i; x^{-i})\| \leq \ell \|x^i - y^i\|,$$

showing that *(SM)* holds, with $L_i = \ell$. Therefore, all theorems from the main paper hold under the assumptions *(QSM)*, *(COCO)*, and *(BV)*, with $\ell$ in place of $L_{\max}$ in step-size restrictions and convergence rates.

**What do we lose by replacing $L_{\max}$ with $\ell$?**    The previous discussion shows that we can assume *(COCO)* and replace all occurrences of $L_{\max}$ with $\ell$ within the theory. In this case, however, the step-size conditions in Theorems 3.3 and 3.4 become

$$\gamma \leq \frac{1}{\ell(\tau + 2(\tau - 1)\sqrt{\kappa})} = \mathcal{O}\left(\frac{1}{\ell \tau \sqrt{\kappa}}\right), \qquad (22)$$

and the $\sqrt{\kappa}$ factor in the denominator is undesirable as it significantly restricts the range of step-size one can use if $\kappa$ is large. Furthermore, in Corollary 3.5 and Theorem 3.6, the factor $q$ becomes $\sqrt{\frac{\ell}{\mu}} = \sqrt{\kappa}$, causing the constant factors in the convergence bounds to potentially become large.

In the following, we demonstrate the commonality of the parameter regime $L_{\max} \ll \ell$, showing why it is beneficial to keep the dependency on $L_{\max}$ tight as we do. First, let $\mathbb{F}$ be a generic $\mu$-strongly monotone and $M$-Lipschitz continuous operator. Then the tight (smallest) cocoercivity parameter one can guarantee on $\mathbb{F}$ is $\ell = M^2/\mu$ (Facchinei & Pang, 2003) (tightness can be shown using, e.g., the scaled relative graph theory in Ryu et al. (2022), Ryu & Yin (2022, Chapter 13)). On the other hand, we have

$$L_{\max} \leq \max_{i=1,\ldots,n} \sup_{\substack{\mathbf{x}=(x^i, x^{-i}), \mathbf{y}=(y^i, x^{-i}) \\ x^i \neq y^i}} \frac{\|\mathbb{F}(\mathbf{x}) - \mathbb{F}(\mathbf{y})\|}{\|\mathbf{x} - \mathbf{y}\|} \leq \sup_{\mathbf{x} \neq \mathbf{y}} \frac{\|\mathbb{F}(\mathbf{x}) - \mathbb{F}(\mathbf{y})\|}{\|\mathbf{x} - \mathbf{y}\|} = M,$$

i.e., $M$ is an upper bound on $L_{\max}$ (better than $\ell$). Therefore, $\ell$ is at least $\frac{\ell}{M} = \frac{\ell}{\sqrt{\ell\mu}} = \sqrt{\kappa}$ times larger than $L_{\max}$, and the largest step-size allowed in Theorems 3.3 and 3.4 is

$$\frac{1}{\ell\tau + 2(\tau-1)L_{\max}\sqrt{\kappa}} = \Omega\left(\frac{1}{\ell\tau}\right)$$

which is in contrast with (22) where we used $\ell$ in place of $L_{\max}$ and obtained $\sqrt{\kappa}$ times smaller step-size range. Additionally, note that in this case $q = \frac{L_{\max}}{\sqrt{\ell\mu}} = \frac{L_{\max}}{M} \leq 1$ in Corollary 3.5 and Theorem 3.6, so we can avoid the $\kappa$-dependent factors appearing in the convergence results.

We present yet another major problem class for which $L_{\max} \ll \ell$. Consider a two-player matrix game, regularized by adding strongly convex (resp. strongly concave) quadratic terms in $x$ (resp. $y$):

$$\underset{u\in\mathbb{R}^m}{\text{minimize}}\ \underset{v\in\mathbb{R}^m}{\text{maximize}}\ \mathcal{L}(u,v) = \frac{\mu}{2}\|u\|^2 + g^\intercal u + u^\intercal \mathbf{B}v - h^\intercal v - \frac{\mu}{2}\|v\|^2 \tag{23}$$

where $\mathbf{B} \in \mathbb{R}^{m\times m}, g, h \in \mathbb{R}^m$. In our $n$-player game notation, the first and second players respectively use the objective function $f_1(x^1; x^2) = \mathcal{L}(x^1, x^2)$ and $f_2(x^2; x^1) = -\mathcal{L}(x^1, x^2)$. In this case, the operator $\mathbb{F}$ is $\mu$-strongly monotone with $\mu$ and $M$-Lipschitz continuous with parameter $M \geq \sqrt{\|\mathbf{B}\|_2^2 + \mu^2} \geq \|\mathbf{B}\|_2$. Note that the cocoercivity parameter $\ell$ is at least $M$ (and at most $M^2/\mu$). On the other hand,

$$\nabla f_1(x^1; x^2) = \mu x^1 + g + \mathbf{B}x^2, \quad \nabla f_2(x^2; x^1) = \mu x^2 + h - \mathbf{B}^\intercal x^1,$$

so the Lipschitz constant for $\nabla f_1$ with $x^2$ fixed (resp. $\nabla f_2$ with $x^1$ fixed) is $\mu$, i.e., $L_{\max} = \mu$. Therefore, we have $L_{\max} \ll \ell$ in this scenario, as strength of regularization $\mu$ is usually small compared to the smoothness parameter $M$. The same principle applies to the $n$-player analogue of this setup we use in Section 4.2, where each player has the objective function

$$f_i(x^i; x^{-i}) = \frac{1}{2}\left\langle x^i, \mathbf{A}_i x^i \right\rangle + \left\langle a_i, x^i \right\rangle + \sum_{\substack{1\leq j\leq n \\ j\neq i}} \left\langle x^i, \mathbf{B}_{i,j} x^j \right\rangle$$

with $\mathbf{B}_{j,i} = -\mathbf{B}_{i,j}^\intercal$. If the quadratic terms are the small regularization terms introduced to induce convergence, so that $\mathbf{A}_i = \mu\mathbf{I}$ with $\mu \ll \|\mathbf{B}_{i,j}\|_2$, then we have $L_{\max} = \mu \ll \max_{i\neq j}\|\mathbf{B}_{i,j}\|_2 \leq \ell$.

### E.2 EXAMPLE OF NON-COCOERCIVE $\mathbb{F}$ SATISFYING *(CVX)*, *(SM)*, *(QSM)* AND *(SCO)*

Consider the two-player game where two players have the objectives

$$f_1(u;v) = \frac{u^2}{2}\varphi(v)$$

$$f_2(v;u) = \frac{v^2}{2}\varphi(u)$$

where $\varphi\colon \mathbb{R} \to \mathbb{R}$ is defined by

$$\varphi(t) = \left(\mu + (\ell-\mu)\sin^2 t\right).$$

Here $0 < \mu < \ell$, and we use the notation $\mathbf{x} = (u,v) \in \mathbb{R} \times \mathbb{R}$ instead of $\mathbf{x} = (x^1, x^2)$ for better readability. Note that because $\varphi$ satisfies

$$0 < \mu \leq \varphi(t) \leq \ell, \quad \forall t \in \mathbb{R},$$

$f_1(\cdot, v)\colon \mathbb{R} \to \mathbb{R}$ is convex (quadratic) for any $v \in \mathbb{R}$, and so is $f_2(u, \cdot)$ for any $u \in \mathbb{R}$. Therefore, this game satisfies *(CVX)*. For any $\mathbf{x} = (u, v)$, we have

$$\mathbb{F}(\mathbf{x}) = (\nabla_u f_1(u;v), \nabla_v f_2(v;u)) = (u\varphi(v), v\varphi(u)).$$

Therefore, the unique equilibrium of the game is $\mathbf{x}_\star = (u_\star, v_\star) = (0,0)$. Additionally, observe that

$$\nabla_{uu}f_1(u;v) = \varphi(v) \in [\mu, \ell], \quad \nabla_{vv}f_2(v;u) = \varphi(u) \in [\mu, \ell].$$

In particular, the both second derivatives are bounded, so *(SM)* is satisfied. Next, we have

$$\langle \mathbb{F}(\mathbf{x}), \mathbf{x} - \mathbf{x}_\star \rangle = u^2\varphi(v) + v^2\varphi(u) \geq \mu(u^2+v^2) = \mu\|\mathbf{x} - \mathbf{x}_\star\|^2,$$

i.e., $\mathbb{F}$ satisfies *(QSM)*. Finally, we have

$$\|\mathbb{F}(\mathbf{x})\|^2 = u^2\varphi(v)^2 + v^2\varphi(u)^2 \leq \max\{\varphi(v), \varphi(u)\}\left(u^2\varphi(v) + v^2\varphi(u)\right) \leq \ell\langle\mathbb{F}(\mathbf{x}), \mathbf{x} - \mathbf{x}_\star\rangle,$$

showing that $\mathbb{F}$ satisfies *(SCO)*.

On the other hand, $\mathbb{F}$ is not cocoercive with respect to any parameter; in fact, it is not even Lipschitz continuous. This is because the cross-derivatives

$$\nabla_{uv}f_1(u; v) = (\ell - \mu)u\sin(2v), \quad \nabla_{vu}f_2(u; v) = (\ell - \mu)v\sin(2u)$$

are unbounded over $\mathbb{R} \times \mathbb{R}$.

Note that while we provided a two-player example for simplicity, one can easily use the essentially same ideas to construct a non-cocoercive $n$-player game satisfying our assumptions with any $n > 2$. For example, we can choose $f_i(x^i; x^{-i}) = \frac{(x^i)^2}{2}\varphi(x^{i+1})$ where we identify $x^{n+1} = x^1$.

