# OpenReview forum: "Multiplayer Federated Learning: Reaching Equilibrium with Less Communications"
_ICLR.cc/2025/Conference — Submitted to ICLR 2025_

### Official Review · Reviewer_xjkH · 2024-11-04

**Soundness:** 3
**Presentation:** 3
**Contribution:** 2
**Rating:** 6
**Confidence:** 3

**Summary:**

The paper introduces a new setting, called Multiplayer Federated Learning (MpFL), in which each client does not share the same objective. More precisely, unlike the standard FL setting where the common objective is to minimize the average of all local losses, in MpFL, each client optimizes its own local loss, which depends on both the client’s model and the models of other clients. Given this setup, the goal is to converge to a Nash equilibrium and the authors propose PEARL-SGD, where each client computes the gradient of its local loss on its own model parameters and apply a (stochastic) gradient descend. The server collects and communicates the updated model parameters of all the clients without aggregating them. The authors analyse PEARL-(S)GD, studying cases with constant or decreasing learning rates, and demonstrate that it converges (sub)linearly to a Nash equilibrium (or a neighborhood of it) depending on the learning rate and the method chosen (SGD vs. GD). All theoretical results are derived under the assumptions of convex and smooth local loss functions. The authors validate their theory through two experiments simulating a 2-player and an n-player game.

**Strengths:**

- The paper is well-written and easy to read.
- The authors introduce MpFL, a new setting that may open up opportunities for future research.
- Convergence guarantee of PEARL-SGD to a Nash equilibrium for convex and smooth local losses.

**Weaknesses:**

- The problem is technically interesting, but I am not convinced by the real-world motivations presented in the paper—specifically, the idea that banks (or pharmaceutical companies) would collaborate and learn together as competitors while maintaining their competitive edge. If these problems are indeed real, the proposed solution may not be well-suited.
First, I find it unlikely that competing companies would be willing to share learning and potentially benefit a competitor, especially without guarantees that the collaboration would be mutually beneficial or at least better than learning alone. Secondly, given that this FL setting is now a multiplayer game where the clients are not from the same entity, expecting competitors to follow the same PEARL-SGD algorithm seems unrealistic. Any client might want to deviate from PEARL-SGD to take advantage over the competitors, or, if the collaboration does not give good results for one of the clients (since Nash equilibriums are not necessarly good for all), the client can simply sabotage the learning. While I find the MpFL setting very intriguing, PEARL-SGD seems incomplete without considering potential adversarial behavior from clients (or perhaps I am missing real-world cases where it cannot be the case?). At present, I struggle to see how this setting or algorithm aligns with a practical problem. Could the authors comment on this?

- The paper addresses only the case of convex functions (I consider this as a "weakness", but my primary concern lies with the first point).

**Questions:**

As said previously my main concerns lies in the motivation of the paper. I am willing to change my score if the authors could address this concern.

---

> ### Author Response · Authors · 2024-11-20
>
> We thank the reviewer for viewing our work as interesting and acknowledging the potential of MpFL. Below, we respond to each point mentioned by the reviewer.
>
> > The problem is technically interesting, but I am not convinced by the real-world motivations ... specifically, the idea that banks (or pharmaceutical companies) would collaborate and learn together as competitors while maintaining their competitive edge. If these problems are indeed real, the proposed solution may not be well-suited.
>
> > First, I find it unlikely that competing companies would be willing to share learning and potentially benefit a competitor, especially without guarantees that the collaboration would be mutually beneficial or at least better than learning alone.
>
> Indeed, as the reviewer points out, Nash equilibrium may not always be good. **However, this is a matter of how the setup is designed; if the objective functions $f_i$ are appropriately chosen, each player can be incentivized to collaborate with others to find Nash equilibrium, even if they are in competing relationships.**
>
> The reviewer’s first question asks what such situations are specifically where participating in the game will benefit all (potentially competing) players.
>
> **Let us first highlight the generality of MpFL.**
> n-player games exist in several scenarios (please see the applications mentioned in our paper), where the goal is to reach an equilibrium. In all of these scenarios, most existing ML approaches assume that everything is centralized (they do not count how expensive communication between players is). In our work via the MpFL setting, we precisely handle this problem, providing for the first time an algorithm (PEARL - SGD) that reaches an equilibrium with less communication between players. We believe this is a more practical scenario than existing approaches for finding equilibrium n-player games, where communications occur in every iteration. **As such, MpFL can be useful for any application that involves formulating the problem as an n-player game.**
>
> Having said the above, we acknowledge that our assumptions and setting are mainly theoretical (the focus of our work), and in our experiment, we have not directly formulated/implemented large-scale applications. We want our work to establish the first convergence guarantees in this setting and open the door for further evaluations (theoretical and practical).
>
> On the application side let us share here our visions on what we think our setting and algorithm would be particularly useful in the future.
> - There is a large body of literature on distributed game theory (for reaching Nash equilibrium), targeting applications in economy, energy consumption control, optical networks, robotics, etc. [2,3,4,5,6] where Nash equilibria are regarded as optimized states. While these works are relatively classical (small-scale) and communication complexity has not been considered a significant concern, we believe that with the progress of n-player games in ML scenarios, relevant large-scale applications will emerge (large number of players), and communication-efficient algorithms like our PEARL-SGD can be particularly valuable in these scenarios.
> - Recent works on language models have reported that the equilibrium of a certain two-player game corresponds to more consistent, accurate output to factual questions [1]. This provides empirical evidence that output consistency can be associated with game-theoretic equilibrium. We believe this idea could be extended and utilized for the joint training of multiple language models.
> - We speculate that in the future, the same idea (as we mentioned in our paper) could be incorporated into banks (or pharmaceutical companies), which would collaborate and learn together as competitors while maintaining their competitive edge. In our opinion, this is potentially the more far-fetched application for which we do not have a clear indicator at this stage. We can easily rephrase this part to better reflect our comment above.

---

> ### Author Response · Authors · 2024-11-20
> **Response continued**
>
> > Secondly, given that this FL setting is now a multiplayer game where the clients are not from the same entity, expecting competitors to follow the same PEARL-SGD algorithm seems unrealistic. Any client might want to deviate from PEARL-SGD to take advantage over the competitors, or, if the collaboration does not give good results for one of the clients (since Nash equilibriums are not necessarly good for all), the client can simply sabotage the learning.
>
> > While I find the MpFL setting very intriguing, PEARL-SGD seems incomplete without considering potential adversarial behavior from clients (or perhaps I am missing real-world cases where it cannot be the case?). At present, I struggle to see how this setting or algorithm aligns with a practical problem. Could the authors comment on this?
>
> We find the reviewer’s second question on the possibility of players’ adversarial behavior thoughtful and interesting. However, we emphasize the following two points.
>
> **First, players are unlikely to have a strong motivation to deviate from collaborative learning if the multiplayer game and its equilibrium are designed properly.** One significant example of this is the well-known _Personalized Federated Learning (PFL)_ setup [7], where each client seeks a local model that is better suited for their own local dataset; under suitable reformulation, PFL can be viewed as an instance of MpFL.
>
> **Second, the problem of defiant clients can be raised to other FL or PFL setups as well.** Even in these well-established setups, there is no guarantee that every iteration of an FL algorithm (or communication round) provides monotonic performance improvement for each client, and a client may refuse to comply at any time. A proper resolution to this question will already constitute a separate paper (see, e.g., [8]). **While we will be excited to see such exploration in future work, at the current stage where we are introducing the foundational concept, we believe having idealized assumptions should not be considered to be a critical flaw.**
>
> We hope that our response has resolved the reviewer’s concerns.  **If we have successfully addressed your concerns, please consider raising your mark. If you believe this is not the case, please let us know so that we have a chance to respond.**
>
> [1] A. P. Jacob, Y. Shen, G. Farina and J. Andreas. The Consensus Game: Language Model Generation via Equilibrium Search. ICLR, 2024.
>
> [2] Z. Li and Zhengtao Ding. Distributed Nash Equilibrium Searching via Fixed-Time Consensus-Based Algorithms. American Control Conference, 2019.
>
> [3] M. Ye and Guoqiang Hu. Distributed Nash Equilibrium Seeking by A Consensus Based Approach. IEEE Transactions on Automatic Control, 2017.
>
> [4] M. Ye. Game Design and Analysis for Price-Based Demand Response: An Aggregate Game Approach. IEEE Transactions on Cybernetics, 2017.
>
> [5] F. Salehisadaghiani and L. Pavel. Distributed Nash equilibrium seeking: A gossip-based algorithm. Automatica, 2016.
>
> [6] D. Kalyva and H. E. Psillakis. Distributed control of a mobile robot multi-agent system for Nash equilibrium seeking with sampled neighbor information. Automatica, 2024.
>
> [7] A. Fallah, A. Mokhtari and A. Ozdaglar. Personalized Federated Learning: A Meta-Learning Approach. NeurIPS, 2020.
>
> [8] K. Donahue and J. Kleinberg. Model-sharing Games: Analyzing Federated Learning Under Voluntary Participation. AAAI, 2021.

---

> ### Author Response · Authors · 2024-11-26
> **Request for response**
>
> Dear Reviewer xjkH,
>
> We would like to provide a reminder that the pdf update deadline is approaching, and ask the reviewer whether they will have any changes to the original evaluation given our rebuttal.
> In our rebuttal, we provided clarifications on **(1) The motivation of MpFL with the discussion on applications, and (2) The problem of defiant clients (players).**
> We also update our pdf file to reflect the reviewer’s comment. We replace the previous discussion with far-fetched examples (banks/pharmaceutical companies) with a more general vision on game-theoretic applications. Please see the parts highlighted in red.
>
> Please let us know if you have any other questions.
>
> If we have properly successfully addressed your concerns, please consider raising the rating beyond the borderline scores to provide support for our work.
>
> Thank you,
>
> The authors.

---

> > ### Comment · Reviewer_xjkH · 2024-11-26
> >
> > I thank the authors for their comments and changes in the revised version of the paper.
> >
> > I am satisfied by the authors’ response. I will increase my score to 6.

---

### Official Review · Reviewer_Bqbe · 2024-11-05

**Soundness:** 3
**Presentation:** 3
**Contribution:** 2
**Rating:** 6
**Confidence:** 4

**Summary:**

This paper studies the federated learning in a game-theoretic context by modeling the multiple agents as multiplayers trying to reach an equilibrium point. Theoretical proof and experiments are proved for the proposed local stochastic gradient based algorithm.

**Strengths:**

The motivation that in many practical FL settings agents may have individual objectives that do not align with the collective goal is interesting and worth studying if well defined.

**Weaknesses:**

1. The paper states that each agent only has access to its own local objective function while in the algorithm it has been shown that the master server collects and distributed all actions to all agents. Also, from the definition of the local utility function in equation (5), all other agents actions are required to compute the local gradient. Therefore, the statement is contradicting.
2. The experiment part is limited. Only the relative error is shown, which is not easy for readers to tell the benefits of the proposed method in the multiplayers environment. For example, can this algorithm make all agents satisfy their learning results? This can be shown by comparing the proposed method with FL using the local losses and the overall losses. It would also be better if some real datasets can be tested.
3. Writing needs to be polished. A thorough review of the abbreviation usage throughout the paper is needed, ensuring each term is defined only once and then used consistently thereafter. For example, Multiplayer Federated Learning (MpFL) appears 3 times in section 1 and 2 times in section 2. This would help improve the overall readability and professionalism of the paper.

**Questions:**

1. How the privacy of each agent's objective function is maintained if the master server is distributing all actions?
2. Can you include comparisons of individual agent performance using local losses to show that all agents achieve their desired learning objectives or can you provide a metric or visualization that shows how well each agent's objectives are satisfied.
3. Consider adding experiments with real-world datasets to demonstrate practical applicability.

---

> ### Author Response · Authors · 2024-11-20
>
> We thank the reviewer for the time and effort spent evaluating our work. Thanks for finding our paper well-written and easy to read, understanding the importance of MpFL, the new setting we proposed in our work, and for highlighting that it has the potential to open up opportunities for future research.
>
> Below, we provide responses to each point mentioned by the reviewer.
>
> > 1. The paper states that each agent only has access to its own local objective function while in the algorithm it has been shown that the master server collects and distributed all actions to all agents … Therefore, the statement is contradicting.
>
> Thanks for the question. **However, the statements are not contradicting.** Accessing the local objective function $f_i$ and accessing the local action $x^i$ are conceptually different things. We stated that the $i$-th player can only compute $f_i$ and its gradient, not that the players should have no access to other players’ actions $x^{-i}$.
>
> > Question 1. How the privacy of each agent's objective function is maintained if the master server is distributing all actions?
>
> As we mentioned before, $f_i$ is not directly accessed by players other than $i$. This is unrelated to having (occasional) access to $x^i$.
>
>
> > 2. The experiment part is limited. Only the relative error is shown, which is not easy for readers to tell the benefits of the proposed method in the multiplayers environment. For example, can this algorithm make all agents satisfy their learning results? This can be shown by comparing the proposed method with FL using the local losses and the overall losses. It would also be better if some real datasets can be tested.
>
> **We emphasize that in MpFL, there is generally no “overall loss”. The goal of MpFL is finding a game-theoretic equilibrium $\mathbf{x}_\star$ defined by equation (1) in the paper (lines 153-154), and this is not necessarily associated with minimizing a specific quantity.** Plotting each local loss value ($f_i$) could be visually appealing, but it would not indicate convergence to $\mathbf{x}_\star$ on its own.
>
> We have run an additional experiment on a robot control setup (presented in the general response), which has been previously considered in distributed game theory literature. If the reviewer believes that this enhances the paper, we will update the manuscript with it.
> ​
> > ​Question 2. Can you include comparisons of individual agent performance using local losses to show that all agents achieve their desired learning objectives or can you provide a metric or visualization that shows how well each agent's objectives are satisfied.
>
> Plotting each local loss value ($f_i$) would not indicate convergence to the equilibrium $\mathbf{x_\star}$, which is the goal of MpFL. The value of each $f_i$ is not necessarily globally minimized at the solution $\mathbf{x_\star}$. This is why we use the (scaled) distance to $\mathbf{x}_\star$ as our performance metric.
>
> **Additionally, running (classical) FL algorithms using local losses does not make sense in the first place.** In the MpFL setup, even the dimension $d_i$ of each player’s action $x^i$ could be all different, while in most FL settings all local models $x^i$ share the same dimensionality. **Majority of existing FL algorithms, which are variants of Local SGD, use averaging of local models $x^i$, but this is not possible in our setup because the players’ actions generally have different dimensions.**
>
> > 3. Writing needs to be polished. A thorough review of the abbreviation usage throughout the paper is needed …
>
> We will carefully review and revise the usage of abbreviations. Thank you for pointing this out.
>
> > Question 3. Consider adding experiments with real-world datasets to demonstrate practical applicability.
>
> Again, please refer to our additional experiment on a robot control setup (presented in the general response), which has been previously considered in distributed game theory literature.
>
> Thanks again for the review and time invested in evaluating our work. **If we have successfully addressed your concerns, please consider raising your mark. If you believe this is not the case, please let us know so that we have a chance to respond.**

---

> ### Author Response · Authors · 2024-11-26
> **Request for response**
>
> Dear Reviewer Bqbe,
>
> We would like to provide a reminder that the pdf update deadline is approaching, and ask the reviewer whether our second rebuttal would change their original evaluation.
> In our second rebuttal, **(1) we demonstrate through a numerical experiment what the plots of $f_i$ could show us, and (2) provide clarification on why Local SGD is not a valid MpFL algorithm.**
>
> We also update our pdf file to reflect the reviewer’s suggestion (please see Appendix D of the updated pdf).
>
> If we have properly successfully addressed your concerns, please consider raising the rating beyond the borderline scores to provide support for our work.
>
> Thank you,
>
> The authors.

---

### Official Review · Reviewer_pJvm · 2024-11-06

**Soundness:** 3
**Presentation:** 2
**Contribution:** 2
**Rating:** 5
**Confidence:** 3

**Summary:**

- This paper introduce a framework, referred to as Multiplayer Federated Learning (MpFL), that models clients as players in a game-theoretic setting and aim to reach an equilibrium.
- Within MpFL, the authors propose an algorithm where each client updates local parameters independently and communicate with other clients periodically.
- The algorithm under different step-sizes are analyzed theoretically that the proposed algorithm needs fewer communications than its non-local counter part, which is verified empirically over simulated data.

**Strengths:**

- The introduction is organized and logical
- The writing is generally easy to follow

**Weaknesses:**

- Unclear definition of MpFL:
	- Despite the introduction and discussion in Section 2.1, there is no rigorous definition of what MpFL is, and how that is different from FL.
	- The definition of FL should be in the preliminaries before introducing MpFL. Instead, it is delayed to Section 2.2.
	- Suggestions
		- Provide a concise, formal definition of MpFL, perhaps in a dedicated subsection or as a numbered definition. This would help clarify exactly what elements constitute the MpFL framework and appreciate the value of such formulation.
		- Discuss how the motivation lead to this formulation. Why communicating only once in a while? How does this falls into the federated learning framework?
		- Discuss the problem formulation itself. Does the problem exists a solution? Is it unique? Does the solution depends on the order of the optimizations of different players?
- Grammatical errors, typos
	- L22: less communications -> "less communication" or "fewer communications"

**Questions:**

- Why is "communicating through a central server" important in MpFL? Instead of concatenate the strategies of all players via a central server in the aggregation round, can we just assume each player can observe other players' strategies?
-  L167: What does the additional subscript $\xi_{i}$ in $f_{i, \xi_{i}}$ mean? Please explicitly define the notation $f_{i, \xi_{i}}$ and explain its relationship to $f_i$.

---

> ### Author Response · Authors · 2024-11-20
>
> We thank the reviewer for the time and effort spent evaluating our work. Below, we provide responses to each point mentioned by the reviewer.
>
> > - Unclear definition of MpFL:
> >   - Despite the introduction and discussion in Section 2.1, there is no rigorous definition of what MpFL is, and how that is different from FL.
> >     - Provide a concise, formal definition of MpFL, perhaps in a dedicated subsection or as a numbered definition ...
>
>
> We thank the reviewer for their comment on improving the presentation of our work.
>
> **However, let us highlight that in Section 2.1, we provide all the details of the problem setup and precise formulation of what we call MpFL.**
> Precisely, in lines 123-126, we provide a high-level conceptual definition of the MpFL. To restate it here with a slight rewording: **MpFL is a setup where multiple players/clients (e.g., mobile devices or whole organizations) play an n-player game with the goal of reaching equilibrium, and they communicate with each other via a central server (e.g., service provider).**
>
> The setting is simple as an idea but, to the best of our knowledge, underexplored in the literature (we are not aware of any paper focusing on this). As mentioned in Section 2.1, MpFL is the setting where we have an n-player game with the players being clients in a distributed network. In this scenario (which is more realistic than the scenario of prior works focusing on n-player games and assuming all the information is centralized), communication-efficient algorithms are required to reach an equilibrium point. Such methods currently do not exist in the literature, and this is what our paper focuses on – proposing and analyzing the PEARL-SGD method that precisely achieves that (reaching equilibrium with less communication compared to its non-local counterpart).
>
> Also through lines 144-168 and Figure 1, we provide all the precise technical information needed for understanding the framework.
> Let us provide the following list of bullet points (restatement of what Section 2.1 already explains), for a clearer demonstration of the characteristics and details of the MpFL setup:
>
> - Each player $i$ has an objective function $f_i(x^i; x^{-i})$ (depending both on their own action $x^i$ and on others’ actions $x^{-i}$), representing their interest.
> - Each $f_i$ is given as the expectation of $f_{i, \xi^i}$ where $\xi^i$ represents the distribution of $i$-th player’s local data.
> - The goal of the MpFL is to reach an equilibrium (defined by equation (1) in the paper).
> - Players other than the $i$-th player do not have access and cannot compute $f_i$ and its gradient.
> - Player $i$ can only update one’s own action $x^i$.
> - The updated values of $(x^1, … ,x^n)$ are shared among players via communication steps. In a communication step, the master node (server/service provider) collects the up-to-date values of $x^1, \dots , x^n$, concatenates them into $\mathbf{x}$, and sends $\mathbf{x}$ to each player.
> - Reducing the communication complexity (number of communications required to find an approximate equilibrium) is the primary concern of the setup.
>
> > The definition of FL should be in the preliminaries before introducing MpFL. Instead, it is delayed to Section 2.2.
>
> This is a personal preference. In our viewpoint, first starting from the multiplayer game and then explaining its federated learning aspect (MpFL) should be prioritized, and the concept of MpFL can be established clearly by itself, without necessarily focusing on classical FL first. However, as we believe it is also important to clarify connections with prior works, in Section 2.2, we provide detailed definitions of FL and Federated Min-max formulations, and how they are different from our approach.

---

> ### Author Response · Authors · 2024-11-20
> **Response continued**
>
> > Discuss how the motivation lead to this formulation. Why communicating only once in a while? How does this falls into the federated learning framework?
>
> Our lines 127-134 explain why we consider the problem of reducing communications important. Specifically, **we are interested in scenarios of modern large-scale, collaborative machine learning among multiple players that are clients within a network with limited communication bandwidth or geographically dispersed organizations** (for which the communication step is the most significant computational bottleneck).
>
> The newly proposed MpFL setup has similarities with the classical cross-device federated learning as it shares important characteristics of FL (see also our discussion of the connections of the two literatures): **1) The clients could potentially be a very large number of mobile or IoT devices (not small size like a classical distributed setting) 2) clients/players have heterogeneous datasets (which are kept private by each player/client), and 3) communication is the primary bottleneck.**
>
> > Discuss the problem formulation itself. Does the problem exists a solution? Is it unique? Does the solution depends on the order of the optimizations of different players?
>
> The existence of a solution is a part of theoretical assumptions, and the assumption (QSM) implies its uniqueness. The notion of a solution (equilibrium) is not affected by the order of players, as we do not assume any particular hierarchical structure in the game. The solution is simply characterized as zero of the joint gradient operator $\mathbf{F}$. We will more clearly explain these points in the revision.
>
> > Grammatical errors, typos
> L22: less communications -> "less communication" or "fewer communications"
>
> We remove the plural and use “less communication”. Thank you.
>
> > Why is "communicating through a central server" important in MpFL? Instead of concatenate the strategies of all players via a central server in the aggregation round, can we just assume each player can observe other players' strategies?
>
> **Having a trustable service provider (master node) taking charge of communication steps is the standard setup considered in most FL works, and we have naturally taken this as a starting point.** We agree that it will be an interesting future direction to study the decentralized setup where there is no master node and the players communicate directly with each other (this is a direction we are currently exploring). This scenario is more complicated as the network topology will be an important factor in the convergence analysis. **At the stage of introducing the new concept of MpFL, starting with the distributed scenario would be a strong enough first step that could open many doors for future exploration.**
>
> > L167: What does the additional subscript $\xi^i$ in $f_{i,xi^i}$ mean? Please explicitly define the notation $f_{i,xi^i}$  and its relationship to $f_i$.
>
> We have defined this in lines 165-167. It represents a dataset of the $i$-th player, following the distribution $\mathcal{D}_i$.
>
> Thanks again for the review and time invested in evaluating our work. We believe that all weaknesses the reviewer mentioned are related to clarifying our setting and algorithm, and improving the presentation of the paper.
> **If we have successfully addressed your concerns, please consider raising your mark. If you believe this is not the case, please let us know so that we have a chance to respond.**

---

> ### Author Response · Authors · 2024-11-26
> **Request for response**
>
> Dear Reviewer pJvm,
>
> We would like to provide a reminder that the pdf update deadline is approaching, and ask the reviewer whether they will have any changes to the original evaluation given our rebuttal.
> In our rebuttal, we provided clarifications on **(1) The problem definition of MpFL, (2) Order of presentation, (3) Motivation of MpFL and why it is considered as FL, (4) Assumption on the solution, (5) Existence of a central server and (6) Definition of $f_{i,\xi^i}$.**
> We also update our pdf file to reflect the changes the reviewer suggested (please see the parts highlighted in red).
>
> Please let us know if you have any other questions.
>
> If we have properly successfully addressed your concerns, please consider raising the rating beyond the borderline scores to provide support for our work.
>
> Thank you,
>
> The authors.

---

> > ### Comment · Reviewer_pJvm · 2024-11-28
> >
> > I thank the authors for the detailed response which partially addressed my concerns. I decided to keep my ratings because the following points are not addressed in the response and revision:
> > 1. On the problem definition of MpFL. I agree that the authors have provided details and descriptions for the setup, but a definition should be mathematical like the ones the authors provided for related works (L204 & L220).
> > 2. Definition of $f_{i,\xi_i}$: L165-167 reads *"Similar to the classical FL regime, our setting focuses on heterogeneous data (non-i.i.d) as we do not make any restrictive assumption on the data distribution Di or the similarity between the functions of the players"* which I don't think is a definition of $f_{i,\xi_i}$. I have not seen the definition updated in the pdfdiff either.

---

> > > ### Author Response · Authors · 2024-11-28
> > >
> > > We thank the reviewer for clarifying the concern once again. We now see that the mathematical definitions of MpFL and $f_{i,\xi^i}$ the reviewer mentioned are something that are clearly analogous to the problem formulation of FL and Federated Minimax Optimization (in lines 204, 220).
> > >
> > > We accordingly restructured the presentation **(lines 154-168)**, with an explanation that $f_{i,\xi^i}$ is the loss of the $i$-th player for a data point $\xi^i$ sampled from the data distribution $\mathcal{D}_i$ **(line 162-163)** and explicit statement via **equation (1) (lines 160-161)** of how MpFL is mathematically formulated.
> > >
> > > We kindly ask the reviewer to check if this change aligns with what reviewer considers as a clear definition.

---

> > > ### Author Response · Authors · 2024-12-02
> > > **Request for reconsideration**
> > >
> > > Dear Reviewer pJvm,
> > >
> > > We would like to provide a kind reminder that we have reflected the reviewer's follow-up comments on mathematical definition.
> > > We made an update on the pdf manuscript, in **lines 154-168**, featuring clear definition that $f_{i,\xi^i}$ is the loss of the $i$-th player for a data point $\xi^i$ sampled from the data distribution $\mathcal{D}_i$ **(line 162-163)** and **equation (1)** defining the mathematical problem which MpFL aims to solve.
> > >
> > > We request the reviewer to let us know if this resolves the remaining concern.
> > > **We believe that we have addressed all points from the reviewer's initial review.**
> > >
> > > Thank you,
> > >
> > > Authors.

---

### Official Review · Reviewer_8Mga · 2024-11-09

**Soundness:** 3
**Presentation:** 2
**Contribution:** 2
**Rating:** 5
**Confidence:** 3

**Summary:**

The paper introduces Multiplayer Federated Learning (MpFL), a framework designed to address the limitations of traditional Federated Learning (FL), which assumes collaborative clients with aligned goals. However, in many real-world settings, clients act as rational players with individual objectives, making a game-theoretic approach more appropriate. MpFL models clients as players aiming to reach an equilibrium, where each optimizes their own utility function, which may diverge from the collective goal. The authors propose a new algorithm, Per-Player Local SGD (PEARL-SGD), in which clients perform local updates independently and periodically communicate with one another. Theoretical analysis, according to the authors, shows that PEARL-SGD reaches equilibrium with fewer communication rounds compared to non-local methods, and numerical experiments are reported to support these findings.

**Strengths:**

1) This paper is generally well-written and accessible, even for those not deeply versed in game theory. Despite my limited background in that area, I was able to follow the main contributions and narrative of the work with ease. The core idea, as well as the supporting results, are conveyed in a clear and structured manner.

2) The introduction provides a strong foundation for the paper, presenting the key contributions in well-organized bullet points that make the novel aspects easy to identify. The inclusion of a more detailed literature review in the appendix is especially appreciated, as it gives interested readers a broader context. Table 1 is highly informative and supports quick understanding, and Figure 1, with its clear diagram, adds helpful visual clarity.

3) The assumptions, lemmas, theorems, and corollaries throughout the paper are presented in a straightforward and precise way. Their formulation is both rigorous and accessible, making it easy to understand the logical flow.

4) Including a draft of the proofs in the main body of the paper is a thoughtful choice. This addition provides readers with helpful intuition, aiding in a deeper understanding of the theoretical claims.

5) The experimental setup is particularly effective, as it clearly demonstrates the theoretical results. I also appreciate the inclusion of results both with theoretically-derived and tuned parameters, allowing for a more comprehensive view of the findings.

6) The conclusion is well-written and provides a concise yet informative summary, reinforcing the contributions and implications of the study.

7) Overall, I find the topic both timely and compelling. This paper appears to be one of the pioneering efforts to explore the integration of game theory into Federated Learning, making it a significant contribution to the field.

**Weaknesses:**

1) The extended introduction is comprehensive, yet it would benefit from including some key recent contributions on client drift reduction. Specifically, the following papers could enrich the discussion, as they address related topics and offer relevant insights:

   - Hu, Zhengmian, and Heng Huang. "Tighter analysis for proxskip." International Conference on Machine Learning. PMLR, 2023.
   - Grudzień, Michał, Grigory Malinovsky, and Peter Richtárik. "Can 5th generation local training methods support client sampling? Yes!" International Conference on Artificial Intelligence and Statistics. PMLR, 2023.
   - Condat, Laurent, and Peter Richtárik. "RandProx: Primal-dual optimization algorithms with randomized proximal updates." arXiv preprint arXiv:2207.12891 (2022).
   - Zhang, Siqi, and Nicolas Loizou. "ProxSkip for Stochastic Variational Inequalities: A Federated Learning Algorithm for Provable Communication Acceleration." OPT 2022: Optimization for Machine Learning (NeurIPS 2022 Workshop).

   The last two references are particularly relevant, as they deal with settings related to minimax problems. It would be beneficial to consider and discuss these works for a more thorough literature grounding.

2) While Table 1 is quite informative, it would be even more helpful if it included a direct comparison with previous approaches. Although some comparative discussion is in the text, incorporating this within the table itself could provide readers with a more immediate visual reference.

3) The assumptions are clearly presented; however, the problem statement itself could be more rigorously defined. A more precise formulation would enhance the clarity and accessibility of the paper.

4) In Algorithm 1, the aggregation step is somewhat underexplained. Furthermore, it’s not emphasized that the server needs to send all client models to each participant. This is a significant concern in Federated Learning, where communication costs are a key bottleneck. It would be beneficial to expand the discussion on this issue, considering its practical implications.

5) In Theorems 3.3 and 3.4, with constant step size, the step size depends not only on $l$, $\tau$, and $L_\max$, but also on $\kappa$. Although Lipschitz constants can generally be estimated, $\kappa$ depends on $\mu$ (strong monotonicity), which is challenging to estimate, potentially weakening the practical applicability of the result.

6) In Theorem 3.3, the contraction factor $(1-\gamma \tau \mu \zeta)^R$ suggests that the step size $\gamma$ is multiplied by $\tau$. However, $\gamma$ is of the order $O(1/\tau)$, which implies that varying the number of local steps does not impact the convergence bound. This indicates that, theoretically, this approach is not more effective than a non-local approach with $\tau = 1$. A similar effect is noted in the stochastic setting. This discrepancy seems to contradict the claim in the abstract: *“We theoretically analyze PEARL-SGD under different step-size selections and prove that it reaches an equilibrium with less communication compared to its non-local counterpart.”* Could you provide clarification on this point?

7) The analysis is conducted under the assumption of quasi-strong monotonicity. Would it be feasible to relax this assumption or perhaps even the convexity assumption? An exploration of less stringent assumptions could broaden the applicability of the findings.

8) In the plots with theoretical parameter settings, the performance lines appear nearly identical. This suggests that the claim that the proposed method outperforms non-local approaches may not hold as strongly as suggested, similar to the issue raised in point 6. Could you comment on this observation?

9) The experiments are conducted primarily on quadratic games. It would be valuable to include additional experiments with more practical objective functions to better represent real-world applications.

10) I am not deeply specialized in this field, but the analysis appears to share similarities with the methods in the following papers:

   - Khaled, Ahmed, Konstantin Mishchenko, and Peter Richtárik. "Tighter theory for local SGD on identical and heterogeneous data." International Conference on Artificial Intelligence and Statistics. PMLR, 2020.
   - Karimireddy, Sai Praneeth, et al. "SCAFFOLD: Stochastic controlled averaging for federated learning." International Conference on Machine Learning. PMLR, 2020.

   Could you please clarify the unique elements of the technical analysis in your proofs compared to these works? Highlighting these distinctions would help underscore the novelty and significance of your theoretical contributions.

**Questions:**

Could you please review the Weaknesses section? I believe some of the points raised may not be entirely accurate. However, since I am not a specialist in this area, I may have overlooked certain details. I am open to reconsidering my score if my concerns can be addressed. As it stands, though, I feel that the paper is not ready for acceptance in its current form.

---

> ### Author Response · Authors · 2024-11-20
>
> We thank Reviewer 8Mga for the constructive and thoughtful feedback. We thank the reviewer for highlighting all the strengths of our work, mentioning that the paper is generally well-written and accessible, the theoretical results are presented in a straightforward and precise way, and the experiments show a comprehensive verification of our findings. Thanks also for pointing out that the topic of our work is timely and compelling.
>
> Below, we address the concerns raised as weaknesses by the reviewer.
>
> > 1. The extended introduction is comprehensive, yet it would benefit from including some key recent contributions…
>
> We have a discussion on client drift, which, in our case, we name player-drift (as the clients are players in our setting) after the results of our main theorems. Even if exploring player drift in more detail is beyond the scope of our work, we can include a discussion of the above papers there.
>
> > 2. While Table 1 is quite informative, it would be even more helpful if it included a direct comparison with previous approaches…
>
> Thank you for the suggestion. This will be a nice idea, but there are no previous works and algorithms that solve the more advanced problem we propose. Introducing the new concept of MpFL is one of the main contributions of our work.
>
> Note that in MpFL, even the dimension $d_i$ of each player’s action $x^i$ could be all different, unlike most FL settings where all local models $x^i$ share the same dimensionality. The majority of existing FL algorithms, which are variants of Local SGD, use averaging of local models $x^i$, **while in our setup, averaging is absurd because the players’ actions have different dimensions and may correspond to different semantic meanings.** In short, it is conceptually invalid to directly compare PEARL-SGD with existing FL algorithms, both theoretically and experimentally, as they are designed for fundamentally different setups.
>
> > 3. The assumptions are clearly presented; however, the problem statement itself could be more rigorously defined. A more precise formulation would enhance the clarity and accessibility of the paper.
>
> Thanks for the suggestion. Let us highlight that in Section 2.1, we provide all the details of the problem setup and precise formulation of what we call MpFL.
>
> The setting is simple as an idea but, to the best of our knowledge, underexplored in the literature (we are not aware of any paper focusing on this). As mentioned in section 2.1, MpFL is the setting where we have an n-player game with the players being clients in a distributed network. In this scenario (which is more realistic than the scenario of prior works focusing on n-player games and assuming all the information is centralized), communication-efficient algorithms are required to reach an equilibrium point. **Such methods currently do not exist in the literature, and this is what our paper focuses on – proposing and analyzing the PEARL-SGD method that precisely achieves this goal (reaching equilibrium with less communication compared to its non-local counterpart).**
>
> > 4. In Algorithm 1, the aggregation step is somewhat underexplained. Furthermore, it’s not emphasized that the server needs to send all client models to each participant…
>
> There is no aggregation step (averaging step) in PEARL-SGD. This is the main difference between the method and MpFL compared to the standard FL setting and methods like Local SGD. We explained the exact details of the method in the first two paragraphs of Section 3 before our main pseudocode. As you can see, the task of the server in our setting is to concatenate the strategies sent to the server by all clients (every few iterations - the local aspect of our approach). **Even in this scenario, communication between clients and servers is the bottleneck, and this is what PEARL-SGD handles, using local computations and communicating only occasionally.**

---

> ### Author Response · Authors · 2024-11-20
> **Response continued (2)**
>
> > 5. In Theorems 3.3 and 3.4, with constant step size, the step size depends not only on $\ell$, $\tau$, and $L_{\mathrm{max}}$ but also on $\kappa$ … potentially weakening the practical applicability of the result.
>
> Indeed, our theorems require the knowledge of $\mu$ to obtain the convergence guarantees.
> While this is a valid point, **theoretical results usually have their value behind the scenes by providing a general understanding of how the algorithm behaves.** We believe that even if the theoretical step-size for PEARL-SGD didn’t have $\kappa$-dependency, in practice, one would rarely select step-sizes based on the theory, e.g., by estimating Lipschitz constants. Rather, one will try out multiple values from a step-size grid and empirically choose the best-performing one. This is why we included the experiments of Figure 2(a,b), 4(a,b) — to simulate such a scenario.
> We speculate that in the future, using potentially different analyses one might be able to remove the dependence on $\mu$ (similar to what was done in the classical FL literature, where one of the earliest papers [1] featured $\mu$ dependence in the step-size, which was later removed).
>
> [1] S. U. Stich. Local SGD converges fast and communicates little. ICLR, 2019.
>
> > 6. In Theorem 3.3, … seems to contradict the claim in the abstract: “We theoretically analyze PEARL-SGD under different step-size selections and prove that it reaches an equilibrium with less communication compared to its non-local counterpart.” Could you provide clarification on this point?
>
> We thank the reviewer for the very careful reading. Stating the conclusion first, the clarification is: **In the deterministic case, indeed, PEARL-SGD is not more effective than its non-local version when using the tight theoretical step-sizes (as Figures 3(c), 4(c) indicate). However, in the stochastic case, PEARL-SGD does achieve the desired accuracy level (distance to equilibrium) within fewer communication rounds.**
> The intuition is as follows. As the reviewer notices, the effect of using $\tau>1$ and $\gamma=O(1/\ell\tau)$ cancel out each other within the term $(1-\gamma\tau\mu\zeta)^R$. However, in the stochastic case, there is an additional “neighborhood term” proportional to $\gamma\sigma^2$. With $\gamma$ being $\tau$ times smaller, this neighborhood term becomes smaller as well. Hence, PEARL-SGD reaches closer to the equilibrium within the same number of communication rounds $R$. In other words, PEARL-SGD requires fewer communication rounds to achieve the same level of accuracy. Corollary 3.5 can be seen as a formalization of this idea.

---

> ### Author Response · Authors · 2024-11-20
> **Response continued (3)**
>
> > 7. The analysis is conducted under the assumption of quasi-strong monotonicity. Would it be feasible to relax this assumption or perhaps even the convexity assumption? An exploration of less stringent assumptions could broaden the applicability of the findings.
>
> Yes, we agree with the reviewer that exploration beyond our current setting of assumption will be valuable, but this could be done in follow-up works.  In our opinion, this should not be seen as a weakness of our work. Not all machine learning papers should primarily focus on general non-convex scenarios. With our work, we provide the setting of MpFL for the first time, and we propose PEARL-SGD as the first algorithm for solving it. Yes, indeed, our analysis focuses on convex scenarios, but this was by design. A new concept should be understood well in favorable scenarios first before moving to a more complicated (e.g., non-convex, non-smooth) setting for which it is not clear how even the non-distributed methods work yet.
>
> Having said the above, let us emphasize the following two points:
>
> **Equilibrium search in games (which our work focuses on) is considered a much more difficult problem class compared to minimization problems (which most classical FL works deal with).** Unlike in minimization problems, in game optimization, gradient descent does not even converge to the local optimum without (quasi-)strong monotonicity. Removing or weakening our quasi-strong monotonicity and playerwise convexity assumptions would indeed be an interesting and important future direction, but it will require additional non-trivial techniques (e.g., smoothed GDA [2], two-timescale extragradient [3,4]).
>
> **In addition, a number of existing theoretical works on classical FL and Local SGD have been done with the strong convexity assumption.** [1,5,6] We believe providing the analysis under seemingly strong assumptions is a meaningful first step, providing intuition that could help the development of future results.
>
> **Please note that the primary purpose of this work is to provide an initial theory of MpFL and set up the conceptual foundations rather than expanding the scope of the theory to the most general level.**
>
> > 8. In the plots with theoretical parameter settings, the performance lines appear nearly identical. This suggests that the claim that the proposed method outperforms non-local approaches may not hold as strongly as suggested …
>
> Our response to Question 6 (the one that questioned our claim on less communication) answers this question as well. While we demonstrated the deterministic case separately to provide a clearer picture, we believe the main application scenarios for MpFL and PEARL-SGD will be the stochastic setups. For such setups, PEARL-SGD has both theoretical/practical gains in terms of the communication cost.
>
> > 9. The experiments are conducted primarily on quadratic games. It would be valuable to include additional experiments with more practical objective functions to better represent real-world applications.
>
> We have run an additional experiment on robot control setups (presented in the general response), which has been previously considered in distributed game theory literature. If the reviewer believes that this could enhance the paper, we will update the manuscript with it.
>
> > 10. I am not deeply specialized in this field, but the analysis appears to share similarities with the methods in the following papers … Could you please clarify the unique elements of the technical analysis in your proofs ...
>
> There are some high-level similarities between these works and our work in the sense that they are all related to client/player drift or the error terms caused by local updates. However, as we explained in our response to Question 2 (on including comparison with previous approaches into Table 1), **none of the previous FL works can handle the general game-theoretical setup that our MpFL addresses. In particular, the PEARL-SGD algorithm and our analysis do not use averaging, and consequently, the techniques for bounding the player drift significantly differ from prior work.** For instance, Lemma 3.8 is one of the key ingredients of our analysis, and we are not aware of the case where similar arguments were used in the FL literature.
>
> [2] J. Yang, A. Orvieto, A. Lucchi and N. He. Faster single-loop algorithms for minimax optimization without strong concavity. AISTATS, 2022.
>
> [3] J. Diakonikolas, C. Daskalakis and M. I. Jordan. Efficient methods for structured nonconvex-nonconcave min-max optimization. AISTATS, 2021.
>
> [4] S. Lee and D. Kim. Fast extra gradient methods for smooth structured nonconvex-nonconcave minimax problems, NeurIPS, 2021.
>
> [5] A. Khaled, K. Mishchenko and P. Richtarik. Tighter theory for local SGD on identical and heterogeneous data. AISTATS, 2020.
>
> [6] K. Mishchenko, G. Malinovsky, S. Stich and P. Richtarik. ProxSkip: Yes! Local gradient steps provably lead to communication acceleration! Finally! ICML, 2022.

---

> ### Author Response · Authors · 2024-11-20
> **Response continued (4, end)**
>
> Thanks again for the review and time invested in evaluating our work. We believe all weaknesses the reviewer mentioned are related to clarifying our setting, algorithm and analysis, which we do above.
>
> **If you agree that we managed to address all issues, please consider raising your mark. If you believe this is not the case, please let us know so that we have a chance to respond.**

---

> ### Author Response · Authors · 2024-11-26
> **Request for response**
>
> Dear Reviewer 8Mga,
>
> We would like to provide a reminder that the pdf update deadline is approaching, and ask the reviewer whether they will have any changes to the original evaluation given our rebuttal.
> In our rebuttal, we provided clarifications on **Weakness 2 (Why existing FL techniques are inapplicable to MpFL setups), Weakness 3 (MpFL problem definition), Weakness 4 (Communication step in PEARL-SGD), Weakness 5 (Step-size dependence on $\mu$), Weaknesses 6&8 (Communication cost reduction in theory/experiments), Weakness 7 (Theoretical assumptions), Weakness 9 (Additional experiments) and Weakness 10 (Unique elements of our theoretical contributions).**
> We also update our pdf file to reflect the suggested changes: **Weakness 1 (Additional references) and Weakness 9** (please see the parts highlighted in red).
>
> Please let us know if you have any other questions.
>
> If we have properly successfully addressed your concerns, please consider raising the rating to provide support for our work.
>
> Thank you,
>
> The authors.

---

> > ### Comment · Reviewer_8Mga · 2024-11-27
> > **Response**
> >
> > Thank you, authors, for the detailed rebuttal.
> >
> > I appreciate the effort you put into addressing several aspects of the work. However, in some cases, your responses seem to focus on related but tangential issues instead of the core concerns I raised. To ensure clarity, I would like to revisit these points one by one:
> >
> > 1. Regarding the player drift aspect, I appreciate your discussion on this topic. However, my concern was more specific. I was asking for a detailed exploration of the relationship between *client drift* and *player drift*. These concepts are related but not identical, and a more in-depth discussion of their connection would help provide a clearer understanding of the proposed method. Thank you for agreeing to expand on this discussion.
> >
> > 2. I understand that previous results do not cover the game theory setting you consider. My suggestion, however, was to examine how the proposed method performs in earlier settings as special cases. This analysis would help illustrate how well the method generalizes and whether the generalization maintains its tightness. Specifically, it would be useful to see that the convergence bounds remain tight without losing precision in simpler or previously studied cases.
> >
> > 3. I acknowledge the novelty of the setting you are addressing. However, I believe that generalizations should aim to preserve tight results, retaining important constants and dependencies in the convergence rates. This is essential to ensure that the theoretical guarantees remain rigorous and meaningful across various scenarios. I appreciate your willingness to clarify this point further in the final version.
> >
> > 4. On the topic of the server-side operation, I understand that your method does not involve explicit averaging. However, I must respectfully disagree with the assertion that there is no aggregation step at all. The server concatenates the models received from all clients and then sends the concatenated set of models back to the clients. This process is, in essence, an aggregation step.
> >
> > More importantly, the proposed method requires the server to send a vector of size $d_1 + \ldots + d_n$ to $n$ clients. This introduces significant communication overhead, especially as the number of clients or the model dimensionality increases. I believe this is a major scalability issue that was not sufficiently highlighted in the paper or addressed in the rebuttal. I encourage you to discuss this limitation more explicitly.
> >
> > 5. Thank you for your comment regarding the tightness of the theoretical bound. If I understand correctly, you acknowledge that the bound is not tight and suggest that improving it is a direction for future work. However, I believe that when generalizing a method to a new setting, it is important to aim for bounds that are reasonably tight in the current work. Loose results may reduce the impact of the contribution and make it harder to compare the method to existing approaches.

---

> > > ### Author Response · Authors · 2024-12-02
> > >
> > > Dear Reviewer 8Mga,
> > >
> > > We would like to provide a kind reminder that we have responded to the reviewer's follow-up comments.
> > > For summary, we provide the list of important changes made to the pdf manuscript based on the reviewer's points:
> > > 1. We added a new detailed discussion on how player drift differs from client drift **(lines 932-948)**.
> > > 2. We reflected the reviewer’s comment on dimensionality and clearly explained this aspect of the synchronization step **(lines 246-249)**.
> > > 3. We updated the abstract and introduction to clearly express the full name of PEARL-SGD to emphasize its stochastic nature **(line 19)**, highlight again that the communication complexity improvement occurs in the sense of reaching a neighborhood of equilibrium with less communication, in the stochastic setup **(line 22)**, explicitly state that our analysis does not achieve communication gain in the deterministic setup **(line 83)**, and clarify that Corollary 3.5 shows communication gain when
> > >  is sufficiently large **(line 91)**.
> > >
> > > We have also provided detailed clarifications on reviewer's comments, including:
> > > 1. We explained that MpFL is not a direct generalization of existing FL setups and therefore it is unclear how the reviewer's comment on comparison with prior work is relevant.
> > > 2. We explained that our analysis is performed tightly, and that we never acknowledged the looseness of our analysis.
> > > 3. We clarify the technical misunderstandings within the review, which led to the inaccurate conclusion that the communication gain by PEARL-SGD is minor and had been oversold. We believe that our work, even in the initial version, had the abstract & introduction well-aligned with actual theoretical contributions. Nevertheless, we tried to further clarify in the writing precisely when, and in what sense, our analysis provides communication gain.
> > > 4. We explained the specific elements of Lemma 3.8 distinguishing our analysis from prior works in classical FL. As a side note, as our Section 3.3 provides a good overview of techniques we use, and technical novelty within the proofs will be clear to readers who are familiar with classical FL.
> > >
> > > **We believe that we have addressed all points raised within the initial review and the follow-up discussion.**
> > > We request the reviewer to let us know if anything is still unclear.
> > >
> > > Thank you,
> > >
> > > Authors.

---

> > > > ### Comment · Reviewer_8Mga · 2024-12-02
> > > > **Response to reminder**
> > > >
> > > > Dear Authors,
> > > >
> > > > Thank you for your reminder. I apologize for not being able to respond as quickly as you may have hoped. I strive to carefully read the text and your responses, as well as to prepare my questions, which requires some time and attention to detail.
> > > >
> > > > I have now added my responses. Thank you for your understanding and for engaging in this productive discussion. Please know that I dedicate significant time and effort to reviewing the material and participating in these discussions.
> > > >
> > > > Best regards,
> > > > Reviewer 8Mga

---

> > ### Comment · Reviewer_8Mga · 2024-11-27
> > **Response (Part 2)**
> >
> > 6. First of all, I find the abstract to be somewhat unclear and potentially misleading. The authors explicitly confirm that the proposed method does not offer any advantage in terms of utilizing local steps in the deterministic case. Presenting the results in a way that might confuse readers or oversell the contributions raises concerns about scientific integrity and clarity. It would be more appropriate to clearly emphasize the limitations of the approach in the abstract to provide an accurate overview of the findings.
> >
> > **Regarding the stochastic case:**
> >
> > I carefully examined Theorem 3.4, which is stated as follows:
> >
> > **Theorem 3.4.** Assume (CVX), (SM), (BV), $(\mathbf{Q S M})$, and $(\mathbf{S C O})$ hold. Let
> > $0 < \gamma_k \equiv \gamma \leq \frac{1}{\ell \tau + 2(\tau - 1) L_{\max} \sqrt{\kappa}}$  and denote  $q = \frac{L_{\max}}{\sqrt{\ell \mu}}.$
> > Then, PEARL-SGD exhibits the rate:
> >
> > $
> > \mathbb{E}\left[\left\Vert\mathbf{x}\_{\tau R}-\mathbf{x}\_{\star}\right\Vert^2\right] \leq(1-\gamma \tau \mu \zeta)^R\left\Vert\mathbf{x}\_0-\mathbf{x}\_{\star}\right\Vert^2+\left(1+(\tau-1)\left((4+\sqrt{3} q) \gamma \tau L\_{\max }+\frac{q}{2 \tau}\right)\right) \frac{\gamma \sigma^2}{\mu \zeta}
> > $ where  $\sigma^2 = \sum_{i=1}^n \sigma_i^2$  and  $\zeta = 2 - \gamma \ell \tau - 2(\tau - 1) \gamma L_{\max} \sqrt{\kappa / 3} > 0$  is guaranteed by the choice of $\gamma$.
> >
> > As mentioned in the paper, let us first consider the case where $\tau = 1$. In this situation, we have $\gamma \leq 1 / \ell$, and the rate reduces to:
> >
> > $$
> > \mathbb{E}\left[\left\Vert\mathbf{x}\_R - \mathbf{x}\_{\star}\right\Vert^2\right] \leq (1 - \gamma \mu)^R \left\Vert\mathbf{x}\_0 - \mathbf{x}\_{\star}\right\Vert^2 + \frac{\gamma \sigma^2}{\mu},
> > $$
> >
> > which is consistent with the standard analysis of the SGDA method.
> >
> > Now, let us turn to the case where $\tau > 1$. In this case, expanding the variance term in the rate gives:
> >
> > $$
> > \mathbb{E}\left[\left\Vert\mathbf{x}\_{\tau R}-\mathbf{x}\_{\star}\right\Vert^2\right] \leq(1-\gamma \tau \mu \zeta)^R\left\Vert\mathbf{x}\_0-\mathbf{x}\_{\star}\right\Vert^2+(\tau-1)\left((4+\sqrt{3} q) \gamma \tau L\_{\max }+\frac{q}{2 \tau}\right) \frac{\gamma \sigma^2}{\mu \zeta} + \frac{\gamma \sigma^2}{\mu \zeta}
> > $$
> >
> > ### Observations on the variance terms:
> >
> > 1. **First term:**
> >    Since $\gamma = \mathcal{O}\left(\frac{1}{\tau}\right)$, the first term, which is responsible for linear convergence, shows very limited sensitivity to the number of local steps $\tau$.
> >
> > 2. **Second term (extra variance term):**
> >    This term is proportional to $(\tau - 1)$ and vanishes when $\tau = 1$. For $\tau > 1$, the term grows as $\mathcal{O}(\tau)$. However, because the stepsize $\gamma$ decreases as $\mathcal{O}\left(\frac{1}{\tau}\right)$, this dependence cancels out, making the second term largely unaffected by variations in the stepsize.
> >
> > 3. **Third term (variance term common to non-local methods):**
> >    This term is identical to the variance term in non-local methods but includes an additional division by
> >    $$
> >    \zeta = 2 - \gamma \ell \tau - 2(\tau - 1) \gamma L_{\max} \sqrt{\kappa / 3} > 0.
> >    $$
> >    Since $\zeta$ cannot exceed 2, this factor provides at most a constant improvement.
> >
> > Overall, the variance term contains an additional summand, but any potential improvement from $\tau > 1$ is limited to a constant factor.

---

> > > ### Author Response · Authors · 2024-11-29
> > > **Author response (Part 2)**
> > >
> > > ## 6. Communication gain in the stochastic setup
> > >
> > > We appreciate the reviewer’s careful look on the theoretical details. We acknowledge that the initial explanation in the rebuttal was not fully informative; we intentionally provided a simplified response to quickly convey the intuition. Now that we know the reviewer is interested in deeper aspects of the analysis, we will provide the exact details below.
> > >
> > > ### 6-1. Local steps are not unnecessary
> > > **First, we point out the reviewer’s following comment is not accurate:**
> > > > For standard SGDA, the stepsize condition $\gamma \le 1/\ell$ can be generalized to $\gamma \le 1 / (C_1 \ell)$ … so the same rate improvement can be achieved without introducing local steps ($\tau > 1$).
> > >
> > > **Simply reducing the step-size in SGDA (without local steps) does not achieve the same effect as PEARL-SGD.**
> > > If one reduces $\gamma$ by the factor of $C_1$ in the non-local SGDA method which has the rate
> > > > $\mathbb{E} \left[ \\| \mathbf{x_R} - \mathbf{x_\star} \\|^2 \right] \le (1-\gamma\mu)^R \\| \mathbf{x_0} - \mathbf{x_\star} \\|^2 + \frac{\gamma\sigma^2}{\mu} $,
> > >
> > > then it does reduce the second term, but the first term $(1-\gamma\mu)^R$ grows larger (which is undesirable).
> > > This is different from the case of local method with $\tau>1$, where reducing $\gamma$ by the factor $\tau$ is canceled out by the additional $\tau$ factor in $(1-\gamma\tau\mu\zeta)^R$, which is essentially left unaffected.
> > >
> > > ### 6-2. On the analysis of variance terms
> > > **It seems the reviewer concluded that the improvement due to local updates in the rate of Theorem 3.4 is at most by a factor of 2 (upper bound on $\zeta$). This is not the case.** The reviewer claimed that the third term $\frac{\gamma\sigma^2}{\mu\zeta}$ is essentially the same for local and non-local methods (except for division by $\zeta$), but this overlooks the fact that PEARL-SGD uses $\gamma$ reduced by the factor of $\mathcal{O}(\tau)$ and therefore this term becomes $\mathcal{O}(\tau)$ times smaller with local updates.
> > > To precisely understand what happens with the second term, we need to dive a bit more into details, which we explain below.
> > >
> > > With the choice $\gamma = \frac{1}{\ell}$, the variance term is $\frac{\sigma^2}{\mu\ell}$.
> > > With $\tau > 1$ and $\gamma = \mathcal{O}\left( \frac{1}{\ell\tau} \right)$, the second and third terms together become
> > > > $\mathcal{O} \left( (1+q)\frac{\sigma^2}{\mu\ell} \left( \frac{1}{\tau} + \frac{L_{\mathrm{max}}}{\ell} \right) \right)$.
> > >
> > > Therefore, the reduction in the variance term is by the factor $(1+q) \left( \frac{1}{\tau} + \frac{L_{\mathrm{max}}}{\ell} \right)$.
> > > Now, in virtually all theory/experiment setups we consider, the factor $\frac{L_{\mathrm{max}}}{\ell}$ has the scale of $\frac{1}{\sqrt{\kappa}} = \sqrt{\frac{\mu}{\ell}}$, or even smaller (for detailed explanation on this, please see **Appendix E, lines 1666-1703** of the updated pdf).
> > > When this is the case, we have $q \le 1$, and therefore, the variance term is reduced by the order of $\frac{1}{\tau} + \frac{1}{\sqrt{\kappa}}$.
> > > This indicates that one can achieve speedup by the order up to $\Theta(\sqrt{\kappa})$, provided that one uses $\tau = \Omega(\sqrt{\kappa})$.
> > >
> > > We plan to appropriately include these ideas into the paper once we are allowed with additional space.
> > > On the other hand, we point out that the above discussion is auxiliary to the flow of the paper, because our main results regarding the communication gain are Corollary 3.5 and Theorem 3.6, which are not directly dependent on the above discussion.
> > >
> > > ### 6-3. Regarding convergence results in terms of $T$
> > > **In Corollary 3.5, we do not tune the total iteration number $T$. Therefore, the reviewer’s claim that this is inapplicable for arbitrary iteration count is not accurate.**
> > >
> > > To elaborate, Corollary 3.5 means the following: Given $T$ (any multiple of $\tau$), we take the (unique) value of $\eta > 0$ such that $T = 2(1+2q) \eta \log \eta$ holds. Then, provided that such $\eta$ is greater than $\kappa \tau$, with step-size $\gamma = \frac{1}{\mu\eta (1+2q)}$, the stated bound holds for PEARL-SGD. That is, **we are choosing $\gamma$ depending on (arbitrary) $T$, not the other way around.** The condition $\eta > \kappa\tau$ requires $R$ to be sufficiently large, so this result considers the regime where $T$ grows large.
> > > We understand that this could have been not crystal clear without being familiar with the FL literature, but note that similar theorem statements have been used in prior work (see, e.g., Corollary 1 in [1]). Once we have more space, we will add comments clarifying this point.
> > >
> > > As a side note, we point out that in Theorem 3.6, we provide the essentially same rate in terms of $T$ (holding for any $T$), which even eliminates the need to tune $\gamma$.
> > > **Corollary 3.5 and Theorem 3.6, together with lines 350–357 of the paper, clearly show the asymptotic $\Theta(1/\sqrt{T})$ improvement on communication complexity over the non-local SGDA (as $T$ grows large).**

---

> > ### Comment · Reviewer_8Mga · 2024-11-27
> > **Response (Part 3)**
> >
> > 6. (Continue)
> >
> > ### Analysis of Corollary 3.5:
> >
> > The authors also provide Corollary 3.5, which states:
> >
> > **Corollary 3.5.** Under the assumptions of Theorem 3.4, let $q = \frac{L_{\max}}{\sqrt{\ell \mu}}$, $\gamma = \frac{1}{\mu \eta(1 + 2q)}$, and $T = \tau R = 2(1 + 2q) \eta \log \eta$, where $\eta > \kappa \tau$ is chosen so that $T$ is a multiple of $\tau$. Then:
> >
> > $$
> > \mathbb{E}\left[\left\|\mathbf{x}\_T-\mathbf{x}\_{\star}\right\|^2\right]=\tilde{\mathcal{O}}\left(\frac{(1+q)^2\left\|\mathbf{x}\_0-\mathbf{x}\_{\star}\right\|^2}{T^2}+\frac{(1+q) \sigma^2}{\mu^2 T}+\frac{(1+q) \tau^2 L\_{\max } \sigma^2}{\mu^3 T^2}\right)
> > $$
> >
> > where $\tilde{\mathcal{O}}$-notation suppresses polylogarithmic factors in $T$.
> >
> > While the corollary demonstrates a theoretical rate, the choice of total iterations, $T = \tau R$, is specifically tuned. This limitation makes the result inapplicable for arbitrary iteration counts, significantly restricting its practical relevance.
> >
> > ### Intuition on local steps:
> >
> > The authors claim:
> >
> > > "In the stochastic case, there is an additional neighborhood term proportional to $\tau$. With $\gamma$ being $\tau$ times smaller, this neighborhood term becomes smaller as well. Hence, PEARL-SGD reaches closer to the equilibrium within the same number of communication rounds."
> >
> > This explanation is somewhat misleading. For standard SGDA, the stepsize condition $\gamma \leq \frac{1}{\ell}$ can be generalized to $\gamma \leq \frac{1}{C_1 \ell}$ with $C_1 > 1$. If $C_1>0$ the stepsize can be scaled the same way as in local method, so the same rate improvement can be achieved without introducing local steps ($\tau > 1$). Thus, increasing the number of local steps is unnecessary to achieve the stated effect.
> >
> > ### Conclusion:
> >
> > Even in the stochastic case, the theoretical advantages of using $\tau > 1$ remain questionable. The statements in the abstract and introduction appear to overstate the benefits, which is concerning from the perspective of scientific rigor. I strongly encourage the authors to revise their presentation to better align with the actual results and to provide a balanced discussion of the method's strengths and limitations.
> >
> > 7. I appreciate that the primary objective of your paper is to explore a new setup. However, I believe that conducting an analysis under a more relaxed setting could significantly enhance the impact and versatility of the work. While I understand your point about leaving such extensions as future work, addressing additional settings within this paper would further strengthen its contributions and broaden its applicability.
> >
> > 8. Kindly refer to my detailed feedback in response to point 6. I remain firmly of the opinion that the results presented in the paper are not fully aligned with the descriptions provided in the abstract and introduction. This disconnect may lead to misinterpretations of the actual contributions and should be carefully addressed to ensure scientific accuracy and transparency.
> >
> > 9. Thank you for incorporating additional experiments. I will review them thoroughly to better understand their implications and relevance to the claims made in the paper. I appreciate your efforts to provide this supplementary evidence, as it adds valuable context to the discussion.
> >
> > 10. Could you please clarify the specifics regarding Lemma 3.8? I suspect that the setup and proof may differ significantly, and I am eager to understand these distinctions in greater detail.
> >
> > While I appreciate the detailed response provided by the authors, I am unable to increase my score as some critical issues remain unresolved.

---

> > > ### Author Response · Authors · 2024-11-29
> > > **Author response (Part 1)**
> > >
> > > We thank the reviewer for the extremely detailed feedback, through which we could understand the reviewer’s intention more precisely. Based on the comments we have made several updates to the paper (please refer to the revised pdf). Below, we provide a list of point-by-point rebuttals to the reviewer’s response.
> > >
> > > ## 1. On the player drift and client drift
> > > We thank the reviewer for clarifying the suggestion. Indeed, player drift is the MpFL-version analogue of client drift, but the two concepts are different. We include the discussion into Appendix A of the revised paper (please see **lines 932-948**), as the discussion is technical and lengthy.
> > >
> > > ## 2. On running PEARL-SGD on “earlier settings”
> > >
> > > While we appreciate the reviewer’s comment, **even if we constrain ourselves to special cases, it is unclear how PEARL-SGD should be evaluated on existing setups from FL. The MpFL setup comes with many conceptual differences from prior FL setups and therefore, is not a straightforward generalization of existing formulations.**
> > >
> > > Below, we explain this in further detail. Suppose we try to compare PEARL-SGD with minimization FL algorithms. Then we could take two simplifications of the MpFL setting:
> > > 1) Set $n=1$, or
> > > 2) Set $f_1=\dots=f_n=f$ (i.e., identical objectives among players).
> > >
> > > If $n=1$, this is no longer a federated learning setup (no client/player to communicate between). If we take the second option, then the MpFL problem can be transformed as $\min_{x^1,\dots,x^n}~f(x^1,\dots,x^n)$ (provided that $f$ is convex). But **this is still totally different from what classical FL algorithms address, because each client/player only controls fixed partial coordinates of $\mathbf{x} = (x^1,\dots,x^n)$.** Prior FL settings do not deal with general optimization problems of such type. There are some loosely relevant works, using specific forms of $f$ designed for specific contexts, but we believe digging into this direction in further detail is really out of the scope of current work.
> > >
> > > ## 3, 5. Tightness of the analysis
> > > **First of all, we clarify that we never acknowledged that our analysis is loose (not tight), neither in the paper nor in the rebuttal. We believe that reviewer’s points 3 and 5 are based on misinterpretation of our intent.** Rather, as the reviewer acknowledges in Point 6, plugging $\tau=1$ into Theorem 3.4 gives the standard analysis of SGDA, indicating the tightness of our analysis. This precisely corresponds to what reviewer mentions; generalization should preserve tightness.
> > >
> > > In the rebuttal we did mention, for example, that $\mu$-dependency of step-size may be resolved in the future.
> > > However, this does not mean that we believe that our analysis is loose. We meant that such improvement might be done using algorithmic adjustments and distinct proof techniques, or by considering additional structure on the n-player game. **We assert that our analysis, at least with the setup and ideas presented in the paper, is performed tightly.**
> > >
> > > ## 4. Clarification on the cost of communication
> > >
> > > The reviewer’s point on the dimensionality is correct, and we explicitly state this aspect of the synchronization step in the revised version (please refer to **lines 246-249** of the updated pdf). We thank the reviewer for the suggestion.
> > >
> > > On the other hand, we clarify once more that **this as a fundamental limitation of the distributed game setup itself** (where each player has their own local action with different dimensions and different semantic meanings, which cannot be condensed into their average as in the classical FL setup), **not as a particular limitation of PEARL-SGD.** The non-local version of PEARL-SGD (distributed per-player SGD; sometimes referred to as SGDA for simplicity) is a standard, baseline algorithm for solving distributed n-player games, and it requires this heavy synchronization at every iteration. PEARL-SGD rather aims to address this limitation of the baseline method by communicating less often.

---

> > > ### Author Response · Authors · 2024-11-29
> > > **Author response (Part 3)**
> > >
> > > ### 6-4. Clarity of the abstract
> > > **We firmly assert that we neither made false statements nor deliberately violated scientific rigor or integrity, at any point.** The name of our algorithm, *Per-Player Local Stochastic Gradient Descent (PEARL-SGD)* already clearly features the ‘stochastic’ aspect, which we believe, clearly indicates that our work is mainly focused on the stochastic setup. However, we included the deterministic analysis to provide the readers with a more complete, transparent picture of the theory. If we had any intention of misleading the readers by concealing information, it is unlikely that we would have included the separate section on the deterministic case in the first place.
> > >
> > > We, however, do see from the reviewer’s comments that it may be worth further emphasis, for the sake of clarity, that the communication gain only occurs in the stochastic setting. In the revised abstract, we clearly express the full name of PEARL-SGD to emphasize its stochastic nature **(line 19)**, and even highlight once again that the communication complexity improvement occurs in the sense of reaching a neighborhood of equilibrium with less communication, *in the stochastic setup* **(line 22)**.
> > > Also, in the revised introduction, we explicitly state that our analysis does not achieve communication gain in the deterministic setup **(line 83)**, and clarify that Corollary 3.5 shows communication gain when $T$ is sufficiently large **(line 91).**
> > >
> > > ## 7. Theoretical assumptions
> > >
> > > We understand and agree with the comment (as it is almost tautologically true that relaxation of assumptions is beneficial). In our case, however, it is far from trivial to modify the components of our analysis to work without one of the assumptions. We strongly believe that it requires novel techniques and ideas (as explained in the original rebuttal), and if successful, it will be a contribution that could be developed into a separate paper.
> > >
> > > ## 8. Please refer to our response to Point 6
> > >
> > > ## 10. Details of technical elements in Lemma 3.8
> > >
> > > We thank the reviewer for their interest. We provide a more detailed explanation as below:
> > >
> > > Lemma 3.8 bounds the distance between the iterate $x_{\tau p}^i$ (from the time of synchronization) and the subsequent iterates $x_{\tau p + t}^i$ within the SGD loop ($0\le t \le \tau$).
> > > Looking at the deterministic case first, the difference $x_{\tau p}^i - x_{\tau p + t}^i$ is the sum of local gradients $\gamma \nabla f_i (x_k^i; x_{\tau p}^{-i})$ for $k=\tau p,\dots,\tau p + t - 1$.
> > > The squared norm of this sum is upper-bounded, using Jensen’s inequality, by the sum of squared norms (up to a constant factor). Then, each squared norm $\gamma^2 \left\\| \nabla f_i (x_k^i; x_{\tau p}^{-i}) \right\\|^2$ is again upper-bounded by $\gamma^2 \left\\| \nabla f_i (x_{\tau p}^i; x_{\tau p}^{-i}) \right\\|^2$ **using Lemma B.1, which (in the deterministic case) states that the gradient norm is nonincreasing along gradient descent for a smooth convex function.** This idea is distinct from existing bounding techniques for FL and Local SGD, which usually establishes inequalities involving the local objective functions’ values $f_i(\cdot)$. In contrast, the function values $f_i(\cdot)$ are never explicitly used in our convergence proofs.
> > >
> > > The proof for the general stochastic case shares the similar spirit at a high level, but requires a more careful handling of stochastic quantities in order to keep the variance term (proportional to $\sigma^2$) in the final bound as tight as possible.
> > >
> > > Another noticeable difference is that in the classical FL setting, each players’ model $x_k^i$ ($i=1,\dots,n$) can be averaged into $\overline{x_k} := \frac{1}{n} \sum_{i=1}^n x_k^i$. Usual analyses of FL put control over the local (client-wise) SGD trajectories based on careful bounding of the model variance $\frac{1}{n} \sum_{i=1}^n \\| x_k^i - \overline{x_k} \\|^2$.
> > > However, such averaging of local models (actions) does not make sense in the MpFL scenario, so we instead control the player-wise SGD trajectories by bounding the quantity $\\| x_{\tau p}^i - x_{\tau p + t}^i \\|^2$ for each $i$ (which is why Lemma 3.8 is needed).
> > >
> > > ### **References**
> > >
> > > [1] A. Khaled, K. Mishchenko and P. Richtarik. Tighter theory for local SGD on identical and heterogeneous data. AISTATS, 2020.
> > >
> > > ### **Concluding message**
> > >
> > > Overall, we greatly appreciate the reviewer’s careful reading and the active discussion with a number of constructive suggestions.
> > > We hope that our response and the improvements made to the manuscript provides a better understanding of our work, both to the reviewer and to the future readers.
> > > We will look forward to hearing back from the reviewer.

---

> > > > ### Comment · Reviewer_8Mga · 2024-12-02
> > > > **Response (Part 1)**
> > > >
> > > > Thank you very much for providing such detailed and thoughtful responses! I appreciate your time and effort. I have outlined some comments, questions, and remarks below that I hope will help clarify certain aspects and offer additional perspectives for consideration.
> > > >
> > > > 1. Thank you for providing this clarification and for the thoughtful explanation.
> > > >
> > > > 2.
> > > > a. I see no issue with considering the option $ n = 1 $, as it naturally represents a simplified special case of the distributed setting. Many distributed methods inherently recover their standard results when $ n = 1 $, making this inclusion both reasonable and useful. Such cases often serve as a baseline for theoretical insights or as a starting point for more complex analyses.
> > > >
> > > > b. I would also like to inquire whether it is possible to consider the following type of problem:
> > > >
> > > > $$
> > > > \min\_{x_1, \ldots, x_n} f(x_1, \ldots, x_n) \quad \text{s.t.} \quad x_1 = x_2 = \ldots = x_n,
> > > > $$
> > > > where $f = \frac{1}{n}\sum_{i=1}^n f_i(x_i)$.
> > > >
> > > > In the homogeneous case, $ f_1 = f_2 = \ldots = f_n $. This reformulation has been addressed in works like the *ProxSkip* paper.
> > > >
> > > > Mishchenko, Konstantin, et al. "Proxskip: Yes! local gradient steps provably lead to communication acceleration! finally!." International Conference on Machine Learning. PMLR, 2022.
> > > >
> > > > Also your approach has some connections to personalization approaches. While personalization might not be the primary focus of your work, it feels somewhat related, especially since such formulations often examine trade-offs between global consensus and local adaptability.
> > > >
> > > > Hanzely, Filip, Boxin Zhao, and Mladen Kolar. "Personalized federated learning: A unified framework and universal optimization techniques." arXiv preprint arXiv:2102.09743 (2021).
> > > >
> > > > Including a brief discussion on this topic could add depth to the paper by connecting it to other areas in distributed optimization or federated learning. It would also highlight the broader applicability of the proposed methods.
> > > >
> > > > I am particularly interested in the formulation at the beginning of this section and would greatly appreciate it if you could share your insights on whether and how this type of problem could fit within your framework. Thank you again for your attention to this matter.
> > > >
> > > > 3.-5. I have carefully reviewed the authors' analysis and appreciate the detailed work and effort put into deriving the results.
> > > >
> > > > Lines 997–1001 discuss the application of Young's inequality, which is a pivotal step in the analysis as it directly influences the contraction factor and error term derived in later sections, particularly in lines 1329–1332. This, in turn, creates a dependency of the step size on the constant $ \mu $ in line 1342. Once the problem in line 1324 is resolved and the optimal $ \alpha $ is substituted into line 1326, the condition in line 1349 naturally follows.
> > > >
> > > > However, the use of Young's inequality, especially in the early stages of the analysis where it is applied to bound the inner product, raises concerns about tightness. While Young's inequality is a standard tool, it is known to introduce looseness in such cases, particularly affecting contraction factors and overall convergence bounds. Here, its application leads to bounds that are not as tight as they could potentially be. While its use in bounding the error (variance) term later in the analysis is less critical, the initial application to the inner product has a compounding negative effect on both the contraction term and the error term. This ultimately makes the overall bounds less precise.
> > > >
> > > > To provide additional context, a similar issue is discussed in Remark 7 of this paper (Ajalloeian, Ahmad, and Sebastian U. Stich. "On the convergence of SGD with biased gradients." arXiv preprint arXiv:2008.00051 (2020)), where the use of Young's inequality in the analysis of biased gradients in a strongly convex setting led to significant challenges in achieving tight bounds. This example illustrates the potential drawbacks of relying on Young's inequality in such scenarios.
> > > >
> > > > In addition to the concerns with Young's inequality, there are some other approximations and simplifications in the analysis. For example, in lines 1234, 1293, and 1338, the authors have used approximations such as $ t+1 \leq 2t $ (line 1338). While these simplifications contribute to improved readability, they come at the cost of slightly looser bounds. That said, these particular approximations are less critical and are acceptable trade-offs for the sake of clarity and accessibility.
> > > >
> > > > Overall, I appreciate the effort and depth of the analysis presented by the authors. However, I believe that addressing the issue with Young's inequality, particularly its early application in bounding the inner product, is essential for improving the tightness of the results. I understand that resolving this issue poses significant technical challenges, but it is a necessary step to enhance the rigor and precision of the derived bounds.

---

> > > > > ### Author Response · Authors · 2024-12-04
> > > > > **Final author response (Part 1)**
> > > > >
> > > > > We acknowledge that the reviewer is spending significant time and effort in reviewing our work and providing constructive suggestions.
> > > > > We sincerely appreciate and respect the reviewer’s deep interest in our paper and the discussion.
> > > > >
> > > > > On the other hand, we would like to note that the reviewer has brought up new technical questions at every cycle—something never mentioned in previous rebuttals/responses. While we understand that the reviewer is being careful and diligent (and we appreciate the effort), this has made it challenging for us to respond to every single question within the constrained time range of the ICLR review process. Having said that, we have done our best to respond to each question the reviewer asked, having in mind we only have 36 hours to respond (since the last update) before the end of the discussion phase.
> > > > >
> > > > > As now we are at the end of the discussion period, **we kindly ask the reviewer to reevaluate our work based on 1) whether we have addressed the reviewer’s major concerns raised so far (in all rounds of discussion), and 2) whether our work will add value to the community.**
> > > > >
> > > > > Below we provide clarifications to the final set of questions:
> > > > >
> > > > > ## 2a. On the comment about the case $n=1$
> > > > >
> > > > > If $n=1$, we have a single player, who does not have any other player to potentially compete with, and the problem is just the (convex) minimization of $f_1 (x^1)$. There is no n-player game structure in it, and this completely changes the dynamics of the optimization. Our algorithm and analysis is designed for n-player games (which is well-known to be more difficult to optimize than simple minimization), and when specialized to the minimization setting, the bounds will naturally be more pessimistic compared to those explicitly designed for convex minimization. **This is clearly not a fair comparison**—we do not expect or require, e.g., a minimax optimization result to maintain tightness for minimization setups—and **we do not feel that this is a meaningful baseline that we should compare with.**
> > > > >
> > > > > ## 2b. On related prior FL works
> > > > >
> > > > > ### **<The consensus formulation and ProxSkip>**
> > > > >
> > > > > The problem the reviewer shared is the classical consensus reformulation in distributed optimization and indeed, was used in [1] (the work that proposed ProxSkip). By incorporating an additional mechanism of enforcing convergence to the consensus set $\{x_1=\dots=x_n\}$, our MpFL setup may recover this result as a special case. However we note that the the formulation uses a very specific, *separable* objective $f(x_1,\dots,x_n) = \frac{1}{n} \sum_{i=1}^n f(x_i)$ where there is no interaction between players (special case of our setting but not relevant from the game-theory viewpoint). We could easily add such comments in the final version of our work.
> > > > >
> > > > > ### **<The personalized FL formulation>**
> > > > > For this, we agree with the reviewer’s comment that the personalized FL is relevant to MpFL.
> > > > > Previously we have only briefly mentioned personalized FL (in lines 951-952) because, as the reviewer says, personalization was not the primary focus of our work. However, given the reviewer’s comments, we now think that highlighting the detailed connection could be valuable. **Although we cannot directly modify the uploaded pdf now, we will mention the following information in the final version.**
> > > > >
> > > > > Consider the following formulation of personalized FL (equation (9) in the paper [2] the reviewer mentioned, or equation (2) in [3]):
> > > > > > $\min_{x^1,\dots,x^n} \frac{1}{n} \sum_{i=1}^n h_i (x^i) + \frac{\lambda}{2n} \sum_{i=1}^n \\| x^i - \overline{x} \\|^2 $
> > > > >
> > > > > where $h_i(x^i)$ is the $i$-th client’s local loss and $\overline{x} = \frac{1}{n} \sum_{i=1}^n x^i$ is the average model.
> > > > > This can be viewed as an instance of MpFL under reformulation, in the following sense.
> > > > > The optimality condition for the above problem is characterized by
> > > > > > $x^i_\star - \bar{x} + \frac{1}{\lambda} \nabla h_i (x^i_\star) = 0$
> > > > >
> > > > > (as mentioned in [3]), and this is equivalent to the equilibrium condition for the n-player game where each player’s objective function is given by
> > > > > > $f_i (x^i; x^{-i}) = \frac{1}{\lambda} h_i (x^i) + \frac{n}{2(n-1)} \\|x^i - \overline{x}\\|^2 $
> > > > >
> > > > > (we have the $\frac{n}{n-1}$ factor because $\overline{x}$ contains $\frac{1}{n} x^i$).
> > > > > Therefore, by solving MpFL with the above specific objectives, one can also handle personalized FL.
> > > > >
> > > > > [1] K. Mishchenko, G. Malinovsky, S. Stich and P. Richtarik. ProxSkip: Yes! Local Gradient Steps Provably Lead to Communication Acceleration! Finally! ICML, 2022.
> > > > >
> > > > > [2] F. Hanzely, B. Zhao and M. Kolar. Personalized Federated Learning: A Unified Framework and Universal Optimization Techniques. TMLR, 2023.
> > > > >
> > > > > [3] F. Hanzely, S. Hanzely, S. Horváth and P. Richtárik. Lower Bounds and Optimal Algorithms for Personalized Federated Learning. NeurIPS, 2020.

---

> > > > ### Comment · Reviewer_8Mga · 2024-12-02
> > > > **Response (Part 2)**
> > > >
> > > > 4. Thank you very much for bringing up this important point and providing the additional clarification.
> > > >
> > > > I fully understand that this limitation is inherent to the theoretical framework you are working with. However, I would like to emphasize that this setting could present considerable challenges in real-world applications. In particular, in the context of cross-device federated learning, it seems quite complicated—and perhaps even unrealistic—to expect that a single communication round could be feasible. This is because the server needs to send models from all clients to each individual client, which becomes particularly problematic when the number of clients is on the order of millions. In such large-scale, distributed settings, the communication cost and overhead could make it nearly impossible to execute even a single communication round effectively.
> > > >
> > > > Therefore, I would suggest that the paper explicitly describe and emphasize the specific practical settings where this theoretical framework is most applicable. For example, cross-silo (organization level) federated learning, where the number of clients is relatively smaller and more manageable, might be a more suitable context. On the other hand, for cross-device settings, this approach might face substantial limitations, making it important to make these distinctions clear. I strongly believe that such a specification would help the reader understand where the framework is truly applicable and would provide more practical value in real-world federated learning scenarios.
> > > >
> > > > I genuinely appreciate your efforts to minimize the number of costly communication rounds, as reducing communication is a key consideration in federated learning. However, if even a single communication round is difficult to achieve in practice, it might be worth considering communication compression techniques or alternative methods that could alleviate this challenge. I recognize that discussing such methods may be beyond the scope of the current work, but I wanted to highlight that focusing on approaches that require impractical communication steps could reduce the overall practical relevance of the framework. A more realistic approach to communication could significantly enhance the impact and applicability of the work.
> > > >
> > > > 6.-1. Thank you very much for pointing out my mistake. I sincerely appreciate your attention to that detail. I realized that I inadvertently omitted the factor $\tau$ in the linear part of the equation. Your clarification regarding both the first (linear) term and the third (last variance) term was very helpful. I truly value your feedback.
> > > >
> > > > Thank you again for your clarification!
> > > >
> > > > 6.-2. Thank you for providing the explanation of the second variance term. However, I would like to note that this term was not fully explained in the initial rebuttal. It should be emphasized that the variance term is proportional to the number of clients. Unfortunately, it was not sufficiently highlighted that this term does not scale with the number of local steps when using the step size $ \frac{1}{\tau \ell} $, as described in the initial rebuttal. I also appreciate the discussion in Appendix E, lines 1666–1703.
> > > >
> > > > Additionally, I noticed that the entire variance term is not divided by the number of clients, which means there is no linear speed-up effect. While I understand that this may be expected in the context of a game-theory setting, I believe it is still worth mentioning.
> > > >
> > > > 6.-3. I would kindly suggest that we refrain from engaging in discussions about who may be more familiar with the federated learning literature or what may be considered "crystal clear" to different individuals. Such discussions may not be particularly productive and might not contribute constructively to the scientific exchange. I believe it would be more beneficial for us to focus on the technical aspects of the work itself, ensuring clarity and precision in the analysis.
> > > >
> > > > Additionally, I would recommend avoiding the use of an argument from authority, where a methodological choice is justified merely by its presence in another paper. While I am familiar with the paper you referenced and respect its contributions, I do have concerns regarding the introduction of additional assumptions, particularly with respect to the number of iterations in the corollary. My suggestion is that if the original theorem does not rely on any assumptions regarding the number of iterations, it would be more consistent and rigorous for the corollary to avoid introducing such assumptions. This would help maintain a coherent logical flow and ensure that the results are consistent with the assumptions established in the main theorem.
> > > >
> > > > 6.-4. Thank you very much for clarifying the abstract and introduction. I truly appreciate the effort you put into making these sections much clearer. I believe that, with these revisions, these sections now explain the obtained results perfectly and provide an excellent overview of the work.

---

> > > > > ### Author Response · Authors · 2024-12-04
> > > > > **Final author response (Part 2)**
> > > > >
> > > > > ## 3, 5. Comment on tightness and Young’s inequality
> > > > >
> > > > > We appreciate the reviewer’s careful reading and the comments. However, we would like to mention two points regarding it.
> > > > >
> > > > > Firstly, Young’s inequality does introduce looseness in some cases, but **this is only true if one has a clearly better way of handling the inner product. Unless the reviewer has a clear vision of how to improve it or suggests other indications of looseness, the use of Young’s inequality alone does not immediately imply looseness.** In our case, despite some trials, we have not found any obvious ways to replace our use of Young’s inequality with a better bound, so we believe it is unlikely that there are straightforward ways to tighten the bound. Additionally, while running experiments with different parameter values $\mu, L_i, \ell$, we numerically observed that our theory often well predicts the step-size range for which the convergence occurs, which is an indication of tightness.
> > > > >
> > > > > Secondly, **we believe that this is not a valid reason to vote for rejection.** If one requires every theory paper to avoid using Young’s inequality, or to fully check whether such sharpening is possible, the bar for publishing any theoretical work will be extremely high. This is reasonable in cases where a paper’s claimed contribution is to provide the tightest possible refinement of an existing work, or the paper’s analysis has an obvious room for improvement. However, neither of them applies to our case. **We believe we have reasonably well-polished results for a work introducing the first analysis in the new setup.**
> > > > >
> > > > > ## 4. On dimensionality issue and practical applications
> > > > > We fully agree with the reviewer’s point related to cross-device and cross-silo federated learning settings. We will include the discussion, and clarify that MpFL (with our current form) is more suitable for the latter. We also agree that incorporating techniques such as compression/sparsification are important future directions that will expand the applicability of the MpFL framework. As these are not limitations of the theory, we can easily handle it in the camera-ready version.
> > > > >
> > > > > ## 6-2. On Theorem 3.4 and variance terms
> > > > > Thanks for following up on our further clarifications. We will make sure to add a remark on the variance term and the no linear speed-up effect of the result, as we agree with the reviewer’s comment that this is expected in the MpFL setup but still is worth mentioning. We will also clarify that the reduction effect of the second variance term is mainly due to the ratio $L_\mathrm{max} / \ell$, and it does not scale down with $\tau$.
> > > > >
> > > > > ## 6-3. On Corollary 3.5
> > > > > In the final version, we will restate Corollary 3.5 so that it is clearly presented that we are not assuming that $T$ should be a certain fixed number. Thank you for letting us realize that this needs clarification, and for understanding that our result does not rely on special assumptions on $T$.
> > > > >
> > > > > ## Concluding statement
> > > > > **We hope that the above response clarifies/addresses the reviewer’s concerns, and that the reviewer might consider reevaluating our work. We thank you once again, for all the productive discussion and feedback.**

---

> > > > ### Comment · Reviewer_8Mga · 2024-12-02
> > > > **Response (Part 3)**
> > > >
> > > > 7. Thank you very much for your insightful comment! I completely understand your point, and I appreciate you taking the time to clarify this aspect.
> > > >
> > > > 10. Thank you for providing such detailed explanations! They have been incredibly helpful in deepening my understanding of the technical aspects of the work.
> > > >
> > > > You have addressed a significant portion of the issues I raised, and as a result, I have adjusted my score accordingly. Thank you for your efforts in resolving these points.

---

### Official Review · Reviewer_DFsy · 2024-11-11

**Soundness:** 3
**Presentation:** 3
**Contribution:** 2
**Rating:** 6
**Confidence:** 4

**Summary:**

This studied the multiplayer federated learning problem, which means that each client learn the model with the knowledge of other clients' model while the master server does not need to do aggregation. The authors called this algorithm as per-player local SGD (PEARL-SGD) and analyze the convergence of it for mainly convex functions.

**Strengths:**

1. The PEAR-SGD is new and interesting.

2. The authors provided the convergence analysis of PEAR-SGD under different settings including convex and its variations.

**Weaknesses:**

1. Although PEAR-SGD is new, I don't see much novelty of it. It is not surprising that the individual model can converge fast by knowing all others' model unless my understanding is wrong.

2. The analysis only focused on convex and its variations. There is no analysis on general non-convex objective functions.

3. Since the master server needs to distribute all the individual models in each round, I didn't see how this can save communication cost.

**Questions:**

In the experiments, I suggest the authors could compare the proposed algorithm with other federated learning learning algorithm in terms of convergence rate, accuracy and communication cost.

**Details Of Ethics Concerns:**

N/A.

---

> ### Author Response · Authors · 2024-11-20
>
> We thank the reviewer for viewing MpFL as a new and interesting problem. Below, we provide responses to each point mentioned by the reviewer.
>
> > Although PEAR-SGD is new, I don't see much novelty of it. It is not surprising that the individual model can converge fast by knowing all others' model unless my understanding is wrong.
>
> **PEARL-SGD is the first local method for solving multiplayer federated learning (MpFL) problems. Any prior works that have the network's clients as players of a game with the goal of finding an equilibrium do not use local steps. This highlights the importance and novelty of PEARL-SGD.**
>
> **Let us also highlight that it is not the case that PEARL-SGD converges fast because it utilizes others’ models as additional information.** In the MpFL setup, “knowing other’s model” (the communication step where individual models/actions are collected and distributed by the master node) is already necessary even for running the baseline algorithm (non-local method: distributed SGDA). **This should not be viewed as a limitation of PEARL-SGD. Rather, it is the other way around; PEARL-SGD using local steps reduces the communication rounds for reaching an equilibrium, i.e., each player/client requires fewer communications with others’ model information (other players).**
>
> For the novelty part, the usage of local gradient steps itself is a very common and widely used concept in the FL literature. However, please note that **we are introducing the new, game-theoretic FL setup (MpFL), and we are providing, for the first time, the proper way of using local gradient steps to work for this new MpFL setup. In this regard, we believe that PEARL-SGD and its analysis are indeed novel.**
>
> Below, through a sub-response, we provide a detailed explanation of why the baseline algorithm already requires communication between the players in MpFL. **We would like to know whether the reviewer would maintain this evaluation, even when given the explanation.**
>
> > The analysis only focused on convex and its variations. There is no analysis on general non-convex objective functions.
>
> Yes, this is true. However we do not believe that this is a weakness of our work. In our opinion, not all machine learning papers should primarily focus on deep learning scenarios. With our work, we provide for the first time the setting of multiplayer federated learning, and proposed PEARL-SGD as an algorithm for solving it. Yes, indeed, our analysis focuses on convex scenarios, but this was by design. A new concept should be understood well in favorable scenarios first before moving to a more complicated (e.g., non-convex, non-smooth) setting for which it is not clear how even the non-distributed methods work yet.
>
> Having said the above, let us emphasize the following two points:
>
> 1) **Equilibrium search in games (which our work focuses on) is considered a much more difficult problem class compared to minimization problems (which most classical FL works deal with).** Unlike in minimization problems, in game optimization, gradient descent does not even converge to the local optimum without (quasi-)strong monotonicity. Removing or weakening our quasi-strong monotonicity and playerwise convexity assumptions would indeed be an interesting and important future direction, but it will require additional non-trivial techniques (e.g., smoothed GDA [6], two-timescale extragradient [1,2]).
>
> 2) **In addition, a number of existing theoretical work on classical FL and Local SGD have been done with the strong convexity assumption** [3,4,5]. We believe providing the analysis under seemingly strong assumptions is a meaningful first step, providing intuition that could help the development of future results.
>
> **Please note that the primary purpose of this work is to provide an initial theory of MpFL and set up the conceptual foundations rather than expanding the scope of the theory to the most general level.**
>
> [1] J. Diakonikolas, C. Daskalakis, and M. I. Jordan. Efficient methods for structured nonconvex-nonconcave min-max optimization. AISTATS, 2021.
>
> [2] S. Lee and D. Kim. Fast extra gradient methods for smooth structured nonconvex-nonconcave minimax problems, NeurIPS, 2021.
>
> [3] S. U. Stich. Local SGD converges fast and communicates little. ICLR, 2019.
>
> [4] A. Khaled, K. Mishchenko, and P. Richtarik. Tighter theory for local SGD on identical and heterogeneous data. AISTATS, 2020.
>
> [5] K. Mishchenko, G. Malinovsky, S. Stich, and P. Richtarik. ProxSkip: Yes! Local gradient steps provably lead to communication acceleration! Finally! ICML, 2022.
>
> [6] J. Yang, A. Orvieto, A. Lucchi and N. He. Faster single-loop algorithms for minimax optimization without strong concavity. AISTATS, 2022.

---

> > ### Author Response · Authors · 2024-11-20
> > **Detailed explanation of why baseline algorithm requires communications**
> >
> > We deal with the n-player game setup where we search for an equilibrium, and traditional algorithms for game optimization assume **centralized training**. In centralized training, a single agent who has access to all models (actions) $x^i$ and all gradients ($\nabla_{x^i} f_i$) performs all computations. It computes the joint gradient operator $\mathbf{F(x)}$ to run the optimization algorithm. In our work, the baseline algorithm is the (centralized) gradient method: $\mathbf{x_{k+1} = x_k - \gamma F(x_k)}$.
> > In the MpFL setup, there are multiple nodes (players) with different interests. Therefore, **each player $i=1,\dots,n$ can only compute their local objective $f_i$ (representing the interest of player i) and its gradient.** But $f_i$ relies on both $x^i$ and $x^{-i}$, so for computing the gradient of $f_i$, the $i$-th player has to know both $x^i$ and $x^{-i}$. Therefore, in order to run the baseline algorithm (the centralized gradient method), **at every iteration**, each player has to receive the information of other players’ actions $x^{-i}$ (from the master node), compute $\nabla_{x^i} f_i (x^i; x^{-i})$ (the $i$-th component of $\mathbf{F}(\mathbf{x})$), update $x^i$, and then share this information with all the other players (through the master node).

---

> ### Author Response · Authors · 2024-11-20
> **Response continued**
>
> > Since the master server needs to distribute all the individual models in each round, I didn't see how this can save communication cost.
>
> As we mentioned above, the baseline algorithm for our setup (not using local steps) requires the master node to collect and distribute all individual models in every iteration. **PEARL-SGD performs this communication step more rarely (reduced by the factor of $\tau$ times), and reaches the neighborhood of equilibrium using fewer numbers of such communication steps,** as shown in Corollary 3.5 and Theorem 3.6.
>
> In our paper, we precisely explain scenarios where our method improves the complexity of communication for solving n-player games (reaching equilibrium with less communication). This is one of the main contributions of our work.
>
> > In the experiments, I suggest the authors could compare the proposed algorithm with other federated learning learning algorithm in terms of convergence rate, accuracy and communication cost.
>
> We strongly emphasize that, to the best of our knowledge, **no previous FL algorithms can handle the general game-theoretic setup that our MpFL addresses. As such there is no other algorithm to compare with that actually solves our multiplayer game setting.**
>
> Note that in MpFL, even the dimension $d_i$ of each player’s action $x^i$ could be all different, unlike the most FL settings where all local models $x^i$ share the same dimensionality. Majority of existing FL algorithms, which are variants of Local SGD, use averaging of local models $x^i$, **while in our setup, averaging is absurd because the players’ actions have different dimensions and may correspond to different semantic meanings. Therefore, it is conceptually not valid to apply existing FL techniques to our experiment setups** of Section 4.
>
> Thanks again for the review and time invested in evaluating our work. Reading your comments, we believe that all pointed weaknesses could be resolved by clarifying our setting and algorithms, as we do above.
> **If you agree that we managed to address all issues, please consider raising your mark. If you believe this is not the case, please let us know so that we have a chance to respond.**

---

> ### Author Response · Authors · 2024-11-26
> **Request for response**
>
> Dear Reviewer DFsy,
>
> We would like to provide a reminder that the pdf update deadline is approaching, and ask the reviewer whether they will have any changes to the original evaluation given our rebuttal.
> In our rebuttal, we provided clarifications on **(1) Novelty and importance of PEARL-SGD, (2) Theoretical assumptions, (3) How reduction of communication cost occurs, and (4) Why existing FL techniques are inapplicable to MpFL setups.**
>
> Please let us know if you have any other questions.
>
> If we have properly successfully addressed your concerns, please consider raising the rating beyond the borderline scores to provide support for our work.
>
> Thank you,
>
> The authors.

---

> > ### Comment · Reviewer_DFsy · 2024-11-27
> >
> > I would like to thank the authors for the detailed response. I raised my score to 6.

---

### Author Response · Authors · 2024-11-20
**General Response to Reviewers**

We sincerely thank all reviewers for providing constructive and thoughtful feedback.

We are delighted that the reviewers agree that **Multiplayer Federated Learning (MpFL) and our algorithms PEARL-SGD are new and interesting** and that **we offer a solid theory that could serve as a basis for future developments.**

We summarize the strengths of our work recognized by the reviewers.

- Reviewers pJvm, xjkH, and 8Mga appreciated the **presentation of the paper** and found it easy to follow.
- Reviewer DFsy recognizes the **novel algorithm PEARL-SGD** introduced in the paper. Moreover, both reviewers DFsy and xjkH appreciate the **rigorous convergence guarantees** of PEARL-SGD established in our work.
- Reviewers Bqbe and xjkH value the **motivation of the MpFL** setup. Reviewer Bqbe understands the potential practical applications of the MpFL setup when **agents may have individual objective functions.** Moreover, reviewer xjkH thinks this will **open up opportunities for further research in this direction.**
- Reviewer 8Mga acknowledges the experimental setup provided in the paper as it demonstrates the theoretical findings of the work.
- Reviewer 8Mga also identified the conclusion section of the paper as very important, in which we provided a concise summary and possible implications of the work.

Here we highlight once again the contributions of our work:

- **We developed a novel Multiplayer Federated Learning (MpFL) framework,** which models the Federated Learning process as a game among players with individual objective functions.
- We proposed a stochastic algorithm called **PEARL-SGD**, where each player performs independent local SGD updates and periodically (after $\tau$ many local steps) communicates with others via a central server.
- We established **rigorous convergence guarantees** of PEARL-SGD for both deterministic and stochastic settings. In the deterministic setting, we have linear convergence with constant step size. Moreover, we use constant and decreasing step sizes in the stochastic setting to ensure linear convergence to a neighborhood and exact sublinear convergence, respectively.
- We provide extensive experiments to validate our theoretical results, demonstrating the **advantage of using multiple local updates** ($\tau > 1$) in PEARL-SGD.

**We hope that you will engage with us in a back-and-forth discussion and we will be happy to answer any remaining questions.**

Below, we provide an additional experiment within the context of distributed control.

---

> ### Author Response · Authors · 2024-11-20
> **Additional Experiment**
>
> Some reviewers raised concerns about the lack of experiments based on real-world applications. To address this, we have conducted an **additional experiment on a real-world problem, and this has been added to the paper, in Appendix D (page 29).**
>
> Here, we consider a distributed formation control problem of mobile robots from [1]. Each robot in this multi-agent system has its own objective, which depends on the positions $x^i$ (corresponding to action/strategy in the formulation of our paper) of robots. Specifically, the objective (cost) function of robot $i$ is given by $J_{i1} + J_{i2}$, where $J_{i1} = \frac{c_i}{2}\| x^i - x^i_\mathrm{anc}\|^2$ represents the cost penalizing the distance of agent $i$ from some anchor point $x^i_{anc}$ and $J_{i2} = \frac{d_i}{2} \sum_{j = 1}^N \| x^i - x^j - h_{ij}\|^2$ is the cost associated with the relative distances between all the robots. The control problem finds a (Nash) equilibrium $\mathbf{x_\star} = ( x^1_\star, \dots, x^n_\star )$ of the n-player game, which is the concatenation of all robots' position vectors ensuring that each robot stays close to its desired location $x^i_\mathrm{anc}$ while maintaining appropriate relative displacement from other robots. We follow the experiment setup within [1] to choose the values of $c_i, d_i, x^i_\mathrm{anc}, h_{ij}$, with $n=5$. We add Gaussian noise to the gradients to simulate stochasticity. In this setup, all our theoretical assumptions are satisfied. Further details are provided in the updated Appendix D of the paper.
>
> We implement PEARL-SGD for this problem, using synchronization intervals $\tau = 1, 2, 4, 5, 8, 20$ and the theoretical step-size $\gamma = \frac{1}{\ell \tau + L_{\max}(\tau -1) \sqrt{\kappa}}$. The [attached plot](https://ibb.co/V3v0FTg) shows that PEARL-SGD achieves better accuracy within a given number of communication rounds with larger values of $\tau$. **This highlights the benefit of using local update steps in solving real-data problems formulated as multiplayer games.**
>
> We address the remaining concerns through our individual responses to each reviewer.
>
> [1] D. Kalyva and H. E. Psillakis. Distributed control of a mobile robot multi-agent system for Nash equilibrium seeking with sampled neighbor information. Automatica, 2024.

---

### Meta-Review · Area_Chair_8y7a · 2024-12-21

**Metareview:**

a) Summary

The paper introduces Multiplayer Federated Learning (MpFL), which models federated learning as an n-player game where each client independently optimizes their own utility function under a game-theoretic equilibrium framework. It proposes the PEARL-SGD algorithm, which uses local updates and periodic communication to reduce the communication complexity required to approximate the equilibrium. Theoretical analysis proves that PEARL-SGD achieves linear convergence to a neighborhood of equilibrium under strong assumptions, with empirical results supporting these claims in stochastic scenarios.

b) Strengths

- **Promising FL research direction**: equilibrium computation in multiplayer games is becoming important given the increasing integration of ML models and agents into everyday life, interacting with one another. Further, the inherently distributed nature of this problem make techniques from FL potentially useful.
- The authors show that FL ideas like local steps and infrequent communication can be generalized to this setting and the analysis techniques translate as well.
- Empirical Validation: The paper provides experimental results that align with theoretical predictions, demonstrating reduced communication costs in achieving equilibrium in stochastic setups.


c) Weaknesses

- Narrowness of results: While the original framework is quite general, the analysis is done only for a very narrow subset. Specifically, they need (quasi) strong monotonicity and (star)co-coercitivity.
- Comparision with existing works: As numerous reviewers pointed out, given the narrowness of their results, the setting becomes directly comparable to many of the federated min-max algorithms. However, the authors do not sufficiently compare or contrast their results and methods with these papers. In particular, a detailed comparision and improvement over (Zhang et al., 2023) is missing.

d) Reason for **rejection**.

While the proposed framework is general, the analysis is carried out only a small limited setting, limiting its generality. Additionally, the paper does not adequately compare or contrast its methods and findings with existing federated min-max algorithms, and with other distributed variational inequality works like Zhang et al. (2023). Hence, despite the direction being very interesting and impactful to the areas of both federated learning and game theory, I recommend rejection in the current form.

**Additional Comments On Reviewer Discussion:**

During the rebuttal, reviewers raised concerns about the narrow theoretical scope of the results, insufficient comparisons with related works, scalability challenges in the communication model, and unclear distinctions between MpFL and existing federated learning frameworks. The authors addressed some issues by clarifying the problem formulation, adding mathematical definitions, and including a new experiment. However, key points—such as the restrictive assumptions limiting generality and insufficient comparisons with existing federated learning algorithms remained inadequately resolved. While the responses demonstrated effort, the lack of comprehensive improvements on these major concerns led to the decision to reject.

---

### Decision · Program_Chairs · 2025-01-22

Reject